# Patterns in data of extreme droughts/floods and harvest grades derived from historical documents in Eastern China during 801–1910

Zhixin Hao[1,2], Maowei Wu[1], Jingyun Zheng[1,2], Jiewei Chen[1,2], Xuezhen Zhang[1,2], and Shiwei Luo[3]

[1] Key Laboratory of Land Surface Pattern and Simulation, Institute of Geographic Sciences and Natural Resources Research, Chinese Academy of Sciences, Beijing 100101, China

[2] University of Chinese Academy of Sciences, Beijing 100049, China

[3] School of Geography and Tourism Science, Chongqing Normal University, Chongqing 401331, China

*Correspondence to*: Jingyun Zheng (zhengjy@igsnrr.ac.cn) and Shiwei Luo (lsw@cqnu.edu.cn)

**Abstract.** In China, historical documents record a large quantity of information related to climate change and grain harvest. This information can help to explore the impacts of extreme drought or flood on crop production, which can provide implications for the adaptation of agriculture to higher extreme climate probability in the context of global warming. In this paper, reported extreme drought/flood chronologies and reconstructed grain harvest series derived from historical documents were adopted, in order to investigate the association between the reported frequency of extreme drought/flood in eastern China and reconstructed poor harvests during 801–1910. The results show that extreme droughts were reported more often in 801–870, 1031–1230, 1481–1530 and 1581–1650 over whole of eastern China. On regional scale, extreme droughts were reported more often in 1031–1100, 1441–1490, 1601–1650 and 1831–1880 in North China, 801–870, 1031–1120, 1161–1220 and 1471–1530 in Jianghuai, 991–1040, 1091–1150, 1171–1230, 1411–1470 and 1481–1530 in Jiangnan. The grain harvest was reconstructed to be generally poor in 801–940, 1251–1650 and 1841–1910, but bumper 951–1250 and 1651–1840, approximately. During the entire period from 801–1910, the frequency of reporting of extreme droughts in any sub-region of eastern China was significantly associated over the long term with lower reconstructed harvests. The association between the reported frequency of extreme floods and reconstructed low harvests appeared to be much weaker, while reconstructed harvest was much worse when extreme drought and extreme flood in different sub-regions were reported in the same year. The association between reconstructed poor harvests and reported frequency of regional extreme droughts was weak during the warm epoch of 920–1300 but strong during the cold epoch of 1310–1880, which could imply that, a warm climate could weaken the impact of extreme drought on poor harvests, yet other historical factors may also contribute to these different patterns extracted from the two datasets.

## 1 Introduction

Extreme drought is the most damaging climate-related hazard to agriculture, as it leads to crop failure due to a reduction in water supply. Numerous studies, based on observation data, show that many regions of the world (in particular southern

Europe and West Africa) have been experiencing trends toward longer and more intense droughts since the 1950s (IPCC, 2012; Dai, 2011; 2013). The climate in eastern China is dominated by the Asian monsoon, with large precipitation variability, leading to drought and flood at regional, and occasionally larger, scale (Ding et al., 2013). Data show that severe and extreme droughts have become more frequent in Northeast China, North China and the eastern part of Northwest China since

the late 1990s, and in Southwest China between 2006–2013 (Zou et al., 2005; 2010; Zhai et al., 2010; Yu et al., 2014). Assessment reports show that droughts affected, on average, $20.9 \times 10^4$ km$^2$ of cropland annually from 1949 to 2013, accounting for one sixth of China's total arable land. From 1991 to 2013, the loss of grain crop yields as a result of drought was, on average, $26.75 \times 10^9$ kg per year. From 2004 to 2013, direct economic losses caused by drought averaged CNY¥63.67 billion (approximately US$10 billion) annually. In addition, projections suggest that the risk of extreme droughts will

increase in most of China along with future global warming (Qin et al., 2015).

Recently, a number of studies have focused on the long-term drought events reconstruction and extreme droughts identification through the use of high-resolution proxy data, such as historical documents and tree rings, at local, regional, and national scales. These studies have found that historical extreme droughts had tremendous impacts on agriculture, livelihoods, and socio-economic systems (e.g., Hao et al., 2010a, b; Shen et al., 2007; 2008; Zhang, 2000; 2005; Zheng et al.,

2006, 2014a; Gou et al., 2015a, b; Deng et al., 2016; Liu et al., 2017; Yang et al., 2014; Li et al., 2018; Xiao et al., 2017; Zhang et al., 2011; Liang et al., 2006). For example, using Chinese historical documents, Hao et al. (2010b) found that droughts in 1876 and 1877 led to 45% and 50% harvest reduction in each of those years, respectively, caused the price of rice to increase by 5–10 times its normal value, and resulted in the deaths of more than 13 million people, due to famine and plague. Zheng et al. (2014a) found that the severe droughts in 1627–1643 triggered and also significantly promoted

widespread peasant uprising, which finally resulted in the collapse of the Ming dynasty.

Meanwhile, several studies also focused on the correlation between reconstructed climate and grain harvest data over the past 2000 years, expecting to build a possible connection between climate and harvest fluctuations at macro-scale. For example, Su et al. (2014) investigated the fluctuations of reconstructed climate and grain harvest data in China from 206 BCE to 960 CE, and found a high positive correlation between reconstructed temperature and grain yield data. According to

their datasets, bumper harvest decades corresponded to warm-normal or warm-wet climate reconstructions, while poor harvest decades often corresponded to cold and dry climate reconstructions. Yin et al. (2015, 2016) further demonstrated that harvests were mostly reconstructed as poor between 210 BCE and 1910 CE under cold conditions, when the climate changed from warm to cold along with dry or from wet to dry, and that reconstructed grain harvest was better during the warm phase. It should be noticed that these conclusions are based on reconstructed datasets, and do not necessarily reflect actual historical

connections due to several reasons as discussed in the *Discussion* section. These studies focused on the connection between agriculture and long-term climate change, while the effect on harvest induced by short-term extreme events (such as extreme droughts/floods) might be different. Therefore, this study aims to explore the patterns in the data of extreme droughts/floods and harvests in eastern China from 801 to 1910, using reconstructions of regional grain harvest grades and extreme drought/flood events derived from Chinese historical documents. The results from historical datasets could provide

implications to improve understanding in the relationship between poor harvests and extreme drought/flood, and how cold and warm periods, such as the Medieval Climate Anomaly (MCA, 950–1250) and the Little Ice Age (LIA, 1450–1850) (IPCC, 2013), contributed to difference in that relationship.

## 2 Data and Method

### 2.1 Data

Two datasets were used, including a chronology of regional extreme drought and flood, and a series of grain harvest grades. Both were derived from Chinese historical documents.

(1) The chronology of regional extreme drought and flood. This chronology was reconstructed based on the annual drought/flood grade at 63-stations in eastern China since 137 BCE (east of 105 °E; 25 °N–40 °N approximately). This area was divided into three sub-regions of North China Plain, Jianghuai and Jiangnan (Fig. 1) (Hao et al., 2010a, and renewed by Zheng et al., 2014b). Records for the reconstruction of annual drought/flood grade at 63-stations were extracted from such historical documents as *Twenty-Four Histories* and *Qing History Draft* (official histories of each Chinese dynasties), chronicles, miscellaneous historical books, local gazettes and others. For example, in both *The Book of Tang* and the *New Book of Tang* (two of *Twenty-Four Histories*), the documents at the 19th year of emperor Zhenyuan in Tang Dynasty (803 CE) record: "In day 25 of the 6th month (of the Chinese lunar calendar, or July 17, 803 in the solar calendar; same hereinafter), no rain fell in the Guanzhong area (now Xi'an and surrounding areas) from the 1st month (January 27 to February 24) until now. Hundreds of officials and many people are praying for rain. In day 26 of the 7th month (August 18), rain. In day 17 of the 8th month (September 6), heavy rain. From mid-autumn, drought occurred again". Descriptions such as these clearly provide information about the duration, spatial coverage and intensity of drought. The drought of that year was also recorded in *The Corpus of Huangpu Chizheng* (written by Huangpu Chizheng, 777–835) and in *The Complete Prose Works of the Tang Dynasty* (edited by Dong Gao, 1740–1818). These texts provide different sources for cross-validation, thus, increasing the reliability of the records. Fig. 2 shows a copy of a local gazette, which regularly recorded anomalous climate information at the county, prefectural or provincial level. From this sample, the duration and intensity of drought or flood occurred in Yangzhou Prefecture. For example, in 1848, there were three instances of heavy rain, causing flooding from the 6th to the 8th month. In 1856, severe drought from the 5th to the 8th month led to the Grand Canal drying up (see Figure 2 caption). As this sample suggests, historical documents were usually focused on the events due to no or less precipitation than usual and, thus could be regarded as meteorological drought rather than hydrological or agricultural droughts, although some records also report impacts on the hydrosphere (e.g., rivers drying up for river) or on agriculture (e.g. wilting for crops). Historical documents, therefore, appear to report all droughts equally, rather than only those affecting crops. Similarly, floods recorded in the historical documents could be regarded as more rain or heavier rain than usual, rather than in the context of rivers bursting their banks or tsunamis, although some records also report the impacts of overflowing or bursting lakes and rivers due to more or heavier precipitation.

Based on these records, Zhang (1996) reconstructed a dataset of annual drought/flood grades at 63 stations from 137 BCE. Each station consisted of a local area of approximately 20 counties with the same climate. Grades were classified using ideal frequency criteria of 10% (grade 1, severe drought), 20% (grade 2, drought), 40% (grade 3, normal), 20% (grade 4, flood), and 10% (grade 5, heavy flood) for the whole area and all time. These grades were calibrated based on descriptions of duration, intensity, and area of the drought/flood event during the wet season (usually May to September), and its impact (Table S1). Thus, the season of the drought/flood grade data overlaps with critical agricultural activities and phases of crop growth. The drought/flood grade data are unevenly spatially distributed across the 2000-year period. For example, drought/flood grade data for south China (south of 30°N approximately) were limited for the period before CE 760, and there were even fewer data for south of the Huaihe River (approximately 34°N) before CE 300 (Zhang, 1996). However, the coverage of this dataset has extended to south China since 760 CE and, therefore, covered the whole study area. There also existed missing data before 1470, as fewer historical documents have survived from these earlier times (Zhang, 1996; Hao et al, 2016). Statistics show that the mean percentage of available data was 44.1% for 800–1469 and only 20% or lower for periods around 850 and for the 880s–920s, 1230s–1250s, 1360s and 1390s. During the period of 800–1469, the mean percentage of available data reporting "disasters or extremes" (i.e., grade 1, 2, 4, 5) was 41.8% and reporting "normal" (i.e., grade 3) was 2.3% (Fig. S1). Moreover, there was a period of 520 years when no "normal" record existed. This means that most of the available grade data recorded disasters and extremes following the principle of "recording the unusual rather than the normal" in the compilation of Chinese history. Even so, it is reasonable to infer that a considerable proportion of extreme events was recorded in 800-1469, comparing the percentage of records reporting "disasters and extremes" in that period and in the ideal frequency (41.8% compared with 60%, respectively). Compared with other datasets, this dataset has certain spatio-temporal advantages. For example, it spans a longer period than the dataset for yearly dryness/wetness grades (1=very wet, 2=wet, 3=normal, 4=dry, 5=very dry) for 120 stations across China from 1470 to 1979 (Academy of Meteorological Science of China Central Meteorological Administration, 1981). In addition, this dataset also covers more areas with higher spatial resolution for local stations (each consisting of around 20 counties), compared with Zhang et al.'s (1997) series of regional dry/wet grades in six areas (consisted of more than 100 counties) from the Lower Yangtze Valley to the North China Plain during 960–1992. Therefore, this dataset provides a valuable proxy and has already been used to study characteristics of precipitation change in eastern China over the past 2000 years. For example, Zheng et al. (2006) used this dataset to reconstruct a 1500 year regional dry/wet index series for the North China Plain (approximately 34–40°N), the Jiang-Huai area (approximately 31–34°N) and the Jiang-Nan area (approximately 25–31°N). Hao et al. (2016) used the dataset to investigate spatial precipitation anomaly patterns in eastern China during centennial cold and warm periods over the past 2000 years. However, this dataset also has weaknesses. For example, the reconstruction derived from the historical documents relies highly on the accuracy of the compilation, and is only available in a form that retracing of the original sources and reconstruction steps are not same as reconstructions from natural evidences.

Since there existed missing data for each of the sub-regions (i.e., North China Plain, Jianghuai and Jiangnan) shown in Fig. 1, extreme drought and flood years were identified according to both the coherence of drought/flood grade for each individual

station, and the percentage of each drought/flood grade occurrence within the available data in each sub-region. In detail, an extreme drought (or flood) year in any sub-region was defined with two terms: (1) grade 3 occurred in less than 25% of all stations, i.e., disasters or extremes were reported for more than 75% of all stations with available data; and (2) grade 1 and grade 2 (or grade 5 and grade 4) occurred in more than 80% stations with records of disasters or extreme conditions (i.e.,

stations with grade 3 or missing data were excluded), and grade 1 (or grade 5) occurred in two or more stations. Extreme drought or extreme flood years were defined in this way under the assumption that the probabilities for omitting drought and flood events in reporting and transmission of historical records were random and unbiased, despite the greater frequency of missing data in the earlier records. In other words, if one period had a large number of documents, it was expected to be rich in both drought and flood records, and vice versa. Therefore, the amount of missing data should not have a significant effect

on the relative drought-to-flood ratio within the available data. To verify the rationality of this method and criteria, validation was conducted in Hao et al. (2010a), based on 10 extreme events identified from a series of precipitation observations in each sub-region according to a threshold of probabilities of 10% and 90% occurrence. In this validation, all or part of grade 3 stations were deliberately omitted, and only 40% or 60% of stations with disaster or extreme grade were reserved without changing the drought-to-flood ratio within the available data. The results show that, with one exception, years of extreme

drought and extreme flood, identified according to this method and criteria, closely matched those extreme events identified by precipitation data, demonstrating that the method and criteria were reasonable. The reason for the close match is that precipitation variability in eastern China is dominated by the East Asian Summer Monsoon (EASM). Therefore, when extreme drought or flood events occur, the precipitation variation for stations within each sub-region usually share similar relative magnitudes. Moreover, this validation also demonstrates that the criteria used for identifying historical sub-regional

extreme drought or flood years is equivalent to the definition of extreme climate event with occurrence probability of less than 10% from 1951 to 2000, which was suggested by IPCC (2012) and has often been adopted in other research focusing on extreme climate.

Furthermore, extreme drought or flood years for the whole of eastern China were defined by reconstructed extreme drought or flood years from the three sub-regions combined, and from the dry-wet index series over the whole of eastern China

reconstructed and renewed by Zheng et al. (2006, 2014c), which synthesized annual drought/flood grades from all 63 stations. The criteria to define an extreme drought or flood year for the whole eastern China were: extreme drought (or flood) in all three sub-regions; or extreme drought (or flood) in two sub-regions and no extreme flood (or drought) in the other sub-region; or extreme drought (or flood) in only one sub-region with an annual dry-wet index for the whole of eastern China lower (or greater) than the mean of that series by at least 1.282 times the standard deviation (i.e., less than 10% probability of

drought or flood occurrence over the last 2000 years). Moreover, if both reconstructed extreme drought years and extreme flood years for different sub-regions overlapped in the same year, that year was defined as an extreme year with the co-reporting of both conditions. In addition, to illustrate the uncertainty of regional extreme drought/flood reconstructions, the confidence levels of these reconstructions were also assessed, based on the percentage of years with data available at 50 year

intervals, in which extremely high confidence is defined as more than 99%; very high confidence: >90%; high confidence: >80%; medium confidence: >50%; and low confidence: >33.3%.

(2) Grain harvest grade series. This series was reconstructed based on historical records collected from the *Twenty-Four Histories* and *Qing History Draft*, which included descriptions of yearly grain yield estimates (e.g., a golden bumper year,
abundant, plentiful, not bad, slightly poor, poor, very poor, etc.) and related information regarding national food security (e.g., enough, insufficient, starving, famine, beggars everywhere, etc.), features of tax remission induced by agricultural disasters, people's livelihoods, grain prices and grain storage status at the country scale from 206 BCE to 1910 CE (Su et al, 2014; Yin et al, 2015). In Chinese historical documents, the yearly harvest was usually recorded as a relative level compared to an expected maximum yield, rather than crop yield per hectare, although some records also report impacts of harvest
fluctuation on food availability, tax remissions, livelihoods, and so on. Therefore, these harvest records exclude differences in absolute yield between sub-regions with different climates, soil fertility and types, crop varieties, etc., as well as difference between historical periods with changing agricultural centres, farming technologies, staple crops, and so on. (Su et al., 2014). As argued by Marks (1998), although it is unclear how yearly harvest percentages in historical times were estimated and what the referee of harvest yield was, also the ratings were probably impressionistic; these harvest rating
estimates in Chinese history were commonly used for food supply management and the state granary system administration by officials, and also served as public information for tenants who use the ratings to reduce the amount of rent they were expected to pay landlords. Therefore, it is believed that the harvest rating descriptions recorded throughout Chinese history are a good indication of fluctuations in yearly grain yields (Marks, 1998).

In the dataset, yearly harvest was classified into 6 grades: 1-Very poor, 2-Poor, 3-Slightly poor, 4-Average, 5-Near bumper,
6-Bumper. It should be noted that these reconstructed grades do not represent absolute grain yield, but rather the relative percentage in production of staple crops and reflect their inter-annual variability. The criteria and methods for year-by-year grading of the documentary records (i.e., grain yield descriptions and related information) were presented by Su et al. (2014) and summarized by Yin et al. (2015). The classification of the yearly harvest grade and descriptions recorded in historical documents is shown in Table S2. Comparison and validation in their researches (Su et al., 2014; Yin et al., 2015) show that
this grade system corresponds to a percentage harvest rating system defined by the government as a code in *The Collected Statutes of the Qing Dynasty* (the "*Da-Qing Hui-Dian*" in Chinese, was finally compiled in 1899, and the definition of the harvest rating system was recorded in Vol. 21), in which "Very poor" corresponds to "below 40%" of harvest yield, "Poor" corresponds to 40–50%, "Slightly Poor" corresponds to 50–60%, "Average" corresponds to 60–70%, "Near bumper" corresponds to 70–80%, and "Bumper" corresponds to above 80%.
It is worth noting that the farther back in time, the more records are missing. This is especially the case prior to 760 CE, regarding both the dataset of annual drought/flood grades for regional extreme drought/flood chronology and the grain harvest grade series. Thus, the study period chosen for investigating the associated variation of reported extreme droughts/floods and reconstructed harvests in eastern China was 801 to 1910. During this study period, several social factors existed which could have influenced China's total yield, such as changing borders of empires, the expansion of agricultural

area (e.g., uplands in the south and southwest colonized by Han settlers during the late Ming and Qing periods), the updated crop varieties introduced from the New World (e.g., peanuts and sweet potatoes), advanced agricultural management technology, and so on. However, such social factors should have only limited influence on yearly harvest grade dataset, since the harvest in the documents was reported as a relative level rather than the absolute yield, also the main grain product area,

the staple crop, and the cropping system have been relatively stable throughout the study period (Yin et al., 2015).

In addition, a 2000-year temperature series, with decadal resolution for the whole of China (Ge et al., 2013), was also adopted to identify long-term cold and warm period fluctuations. This temperature series was reconstructed based on 28 temperature proxies using principal component (PC) regression and partial least squares (PLS) regression, respectively (Ge et al., 2013). The comparison showed that the pattern of long-term temperature change over China is roughly consistent with

that shown by the IPCC (2013) for the Northern Hemisphere as an overlap of multi-reconstructions since 850.

**2.2 Method**

Four kinds of data processing method were used in this study, including the moving average, the Wilcoxon rank sum test, the two-sampled t-test and the contingency table with the Chi-square test ($\chi^2$).

(1) To illustrate variations in the reported frequency of regional extreme drought and flood from 801 to 1910, the moving-

window frequency of reconstructed extreme drought/flood years in three sub-regions and in the whole of eastern China were first calculated with windows of 50 years and steps of 10 years. For example, a smoothed series was made up of means of 801–850, 811–860, 821–970, and so on. Next, the Wilcoxon rank sum test was applied, to examine which interval had significantly more or fewer drought/flood years compared to all the other intervals. By labelling extreme drought (or flood) years as 1 and non-extreme years as 0, the chronology of extreme drought (or flood) years could be transformed into a rank

series, with the mean of this rank series equivalent to the frequency of drought (or flood) years. Therefore, those intervals with significantly more or fewer reconstructed drought (or flood) years could be recognized through a Wilcoxon rank sum test performed on the rank series.

(2) To examine whether the reported frequency of extreme drought/flood in each sub-region and in the whole of eastern China were associated with reconstructed poor harvest, harvest grade data for 801–1910 were divided into three categories:

extreme drought years, extreme flood years, and non-extreme years, according to the chronology of regional extreme drought and flood for each sub-region or for the whole of eastern China. The two-sampled t-test was then applied to examine whether the means of harvest grade in extreme drought (or flood) years and non-extreme years showed a significant difference. Meanwhile, a contingency table and Chi-square test ($\chi^2$) were adopted to examine the association between reported regional extreme drought/flood and each grade of reconstructed harvest. For example, corresponding to the 720

30  years with available harvest grade data, there were 97 extreme drought years, 82 extreme flood years, and 541 non-extreme years in reconstruction for the North China Plain. To examine whether there was a significantly higher frequency of harvest reconstructed as poor (grade 2) in extreme drought years compared to non-extreme years for this sub-region, a contingency

table was made by counting extreme drought years with grade 2, extreme drought years with other grades, non-extreme years with grade 2, and non-extreme years with other grades. The chi-square test ($\chi^2$) was then adopted to test the significance.

(3) To illustrate whether there were any differences in the frequency of reporting extreme drought/flood between cold and warm periods, the frequency of reconstructed extreme drought/flood years was calculated and the Wilcoxon rank sum test was performed for cold and warm periods in the three sub-regions and the whole of eastern China. Similarly, the contingency table and Chi-square test ($\chi^2$) were adopted to examine if there existed significantly different pattern in the association between the reported frequency of regional extreme drought/flood and reconstructed harvests during cold and warm periods.

## 3 Results

### 3.1 The variation of reported regional extreme drought/flood and reconstructed grain harvest grade during 801–1910

Fig. 3 shows the chronology of extreme drought/flood years and variation for each sub-region (i.e., North China Plain, Jianghuai and Jiangnan area), as well as the whole of eastern China from 801 to 1910. According to this chronology, the North China Plain had 133 extreme drought years and 113 extreme flood years, Jianghuai had 95 extreme drought and 118 extreme flood years, and Jiangnan had 90 extreme drought and 119 extreme flood years (Table 1). The comparison shows that the reporting of extreme droughts was a bit more frequent than extreme flood in the North China Plain due to its sub-humid climate, while both Jianghuai and Jiangnan were reported to have slightly fewer extreme drought than extreme flood years, due to the humid climate in these regions. The whole of eastern China had 126 extreme drought years and 122 extreme flood years, and 20 extreme years when extreme drought and flood were co-reported in different sub-regions.

Moreover, moving-window frequencies of extreme drought/flood years during each 50 years with a decadal step show remarkable multi-decadal variation in the frequency of reporting both extreme drought and flood in each sub-region and the whole of eastern China from 801 to 1910 (Fig. 3). In the North China Plain, extreme droughts were reported more frequently in 1031–1100, 1441–1490, 1601–1650, 1831–1880, but less in 1151–1200, 1381–1430, 1651–1710, and 1721–1780. Extreme floods were reported more frequently in 941–1000, 1021–1070, 1721–1790, 1801–1830, 1841–1900, but less in 821–880, 1191–1260, and 1461–1550. In Jianghuai, extreme droughts were reported more frequently in 801–870, 1031–1120, 1161–1220, 1471–1530, but less in 881–960, 1221–1290, 1301–1350, 1381–1430, 1531–1580, and 1671–1760. Extreme floods were reported more frequently in 1551–1600, 1641–1680, 1691–1710, 1721–1770, 1811–1890, but less in 1261–1310, 1321–1370, and 1461–1550. In Jiangnan, extreme droughts were reported more frequently in 991–1040, 1091–1150, 1171–1230, 1411–1470 and 1481–1530, but less in 1231–1320, 1361–1420, 1691–1750 and 1841–1890. Extreme floods were reported more frequently in 991–1040, 1151–1230, 1241–1300, 1400–1430 and 1821–1880, but less in 851–970 and 1300–1370. For the whole of eastern China, extreme droughts were reported more frequently in 801–870, 1031–1230, 1481–1530 and 1581–1650, but less in 891–1000, 1231–1320, 1381–1430, 1531–1580, 1651–1780, 1791–1850 and 1881–1910. Extreme floods were reported more frequently in 811–840, 951–990, 1051–1070, 1381–1430, 1641–1670, 1721–1770

and 1810–1870, but less in 841–900, 1321–1380, 1431–1560, 1581–1640 and 1671–1700. Moreover, there was significantly more frequent co-reported extreme drought and flood years in 1291–1300, 1481–1500 (both with 2 years) and 1581–1600 with 3-years, respectively.

Figure 4a shows the series of reconstructed grain harvest grades from 801 to 1910, with 390 years (35.1%) of missing data. Percentages for each harvest grade were 4.9% (grade 1), 16.0% (grade 2), 18.4% (grade 3), 14.4% (grade 4), 6.9% (grade 5), and 4.2% (grade 6) (Table 2). In total, the percentage of all poor harvest years (grades 1, 2 and 3) was 39.3%, while the percentage of bumper harvest years (grades 5 and 6) was only 11.2%. Moreover, the moving-window percentage for each harvest grade and data missing for each 50 years with a decadal step (Fig. 4b) show an evident jump around the 1640s, when there was an increase in grade 4 years and a decrease in missing data years. This is mainly because the *Qing History Draft* recorded more "average" harvest years, while the *Twenty-Four Histories* excluded most "average" years, according to the principle of "recording unusual rather than common events", as argued in Yin et al. (2015). Therefore, to avoid the effect of missing data and to maintain the homogeneity in statistics, data missing years and "average (grade 4)" years were excluded in the analysis, as adopted in Zheng et al. (2006). Relative percentages for years with each anomalous harvest grade (i.e., 1, 2, 3, 5, and 6) compared with total anomalous years were calculated for each 50-year moving-window (Fig. 4c) to illustrate variation in reconstructed grain harvest during 801–1910. By excluding grade 4 and missing data years, the relative percentages for each harvest grade for the whole period were 9.6% (grade 1), 31.8% (grade 2), 36.4% (grade 3), 13.8% (grade 5), and 8.4% (grade 6) (Table 2). Noted that, in Fig. 4c, grades 1 and 2 (hereafter, "grade 1+2") were combined as the poor group, and grades 5 and 6 (hereafter, "grade 5+6") as the bumper group, as grade 1 and grade 6 were only 4.9% and 4.2%, respectively, in all years. For the entire period, the relative percentages of grade 1+2 and grade 5+6 were 41.4% and 22.2%, respectively.

According to the variation in the frequency of reconstructed poor and bumper harvests (Fig. 4c), the grain harvest for 801–1910 can be roughly divided into five periods. From 801 to 940, the reconstructed grain harvest was generally poor with fewer bumper harvests (only 7.1% for grade 5+6) but more slightly poor harvests (51.4% for grade 3). From 951 to 1250, there were more reconstructed bumper harvests (31.9% for grade 5+6) with relatively fewer poor harvests (37.7% for grade 1+2, 30.4% for grade 3), with the exception of 1121–1170, when the frequencies of harvest grades were similar to that of the 9th Century. From 1251 to 1650, there were noticeably more reconstructed poor harvests (63.1% for grade 1+2), but very few bumper harvests (9.2% for grade 5+6). In 1651–1840, there were more reconstructed bumper harvests (53.5% for grade 5+6) and fewer poor harvests (9.3% for grade 1+2). After 1841, the reconstructed grain harvest became generally poor again with fewer grade 5+6 (9.1%) but more slightly poor (63.6% for grade 3) harvest grades. On the whole, the reconstructed harvest was generally better in 951–1250, corresponding to a warmer climate, and lower in 1450–1650, corresponding to a colder climate, as argued in Yin et al. (2015, 2016). However, the reconstructed harvests were again higher in 1651–1840, corresponding to a cold climate. This can also be confirmed by other datasets from independent sources. For example, according to harvest reports in the Archives of the Qing Dynasty, the mean harvest percentage over eastern China for 1730–

1820 was even greater than 70% (i.e. near bumper) (Ge and Wang, 1995). In Guangdong Province, southern China, this was over 75% for 1707–1800 (Marks, 1998).

## 3.2 Association between reported frequency of regional extreme drought/flood and reconstructed harvest grades during 801–1910

Table 3 shows the mean harvest grade for all reconstructed extreme drought/flood years and non-extreme years for the three sub-regions and the whole of eastern China from 801 to 1910. The mean was 3.3 for all non-extreme years over the whole of eastern China, and similar in the three sub-regions. The two-sampled t-test showed the mean harvest grade for all extreme drought years was significantly lower than that for non-extreme years, which was 2.95 for all extreme drought years in the whole of eastern China, 2.84 for the North China Plain, 2.89 for Jiangnan, and 2.99 for Jianghuai. However, no significant

differences were detected for extreme flood years in any sub-region. Although the mean harvest grade in years of co-reporting of extreme drought and flood for the whole of eastern China was a mere 2.53 (Table 3), which is much lower than the mean of 3.30 for non-extreme years.

To illustrate the association between reconstructed harvest and reported frequency of regional extreme drought/flood, Table 4 shows the frequency of each harvest grade (i.e., the percentage of years with each harvest grade, accounting for all years

without missing data) for extreme drought/flood years and non-extreme years at each sub-region and for the whole of eastern China. It was found that the frequency of poor harvests (grade 1+2) for extreme drought years in the North China Plain significantly increased from 29.2% (for non-extreme years, hereinafter) to 47.4%, with frequencies of grade 1 and grade 2 significantly increasing from 6.3% and 22.9% to 13.4% and 34.0%, respectively. Compared with non-extreme years, although extreme drought in Jianghuai was not accompanied by significant increases in frequency of grade 1 or grade 2,

there was a significant increase in the frequency of grade 1+2 (poor harvest) from 31.5% to 41.9%, and a significant decrease in the frequency of grade 5 (near bumper harvest) from 12.2% to 5.4%. For extreme drought years in Jiangnan, the frequency of grade 1+2 (poor harvest) significantly increased from 30.3% to 51.5%, with the frequency of grade 2 significantly increasing from 22.9% to 42.4% and the frequency of grade 4 ("average" harvest) significantly decreasing from 23.8% to 12.1%. When extreme droughts occurred in the whole of eastern China, it was found that, not only did the

frequencies of grade 1 and grade 2 increase from 6.8% and 22.4% to 11.7% and 36.2%, respectively, but also the frequency of grade 4 significantly decreased from 24.6% to 13.8%. Moreover, it was also found that the co-reporting of extreme drought and flood in the whole of eastern China, though only in limited number of years, was associated with the greatest shift in reconstructed harvest grades, in which the frequency of grade 2 significantly increased from 22.4% to 60.0%. In summary, reported extreme drought and reconstructed poor harvests (grade 1+2) tend to occur simultaneously, and this

pattern could be found in any sub-region of eastern China.

However, the association between reconstructed harvest and reported frequency of extreme floods seems to be much weaker. No significant difference was found between the frequency of poor harvests in extreme flood years and non-extreme years, although extreme floods were accompanied by significantly decreased bumper harvest frequency in some cases. Extreme

floods in Jianghuai and Jiangnan, for instance, were both associated with significantly decreased frequency of grade 5 (near bumper) harvests from 12.2% and 11.9% to 5.7% and 5.6%, respectively. It is possible that rainfall in these two humid areas was already sufficient for crop growth, thus, extreme floods had a negative impact on harvest potential (Zheng and Huang, 1998).

## 3.3 The different patterns in association between reported frequency of regional extreme drought and reconstructed grain harvest between warm and cold periods

To explore whether the reported frequency of regional extreme drought/flood was differently associated with reconstructed grain harvests in warm and cold periods, a multi-proxies-based temperature reconstruction with a decadal resolution for all of China is presented in Fig. 3e. It is shown that temperature across China from 801 to 1910 experienced two long-term anomalous epochs, both of which contained several multi-decadal fluctuations. These two anomalous epochs were the warm epoch during 920–1300 and the cold epoch during 1310–1880 (Fig.4), in which the first one covered MCA (950–1250) and the latter covered LIA (1450–1850).

The comparison in the reported frequencies of extreme drought/flood between the warm (920–1300) and cold (1310–1880) epochs (Table 5) shows only a small difference in the reported frequency of extreme droughts between warm and cold epochs for the North China Plain and Jianghuai. However, in Jiangnan, reconstructed extreme drought years was more frequent in 920–1300 (10.8%) than in 1310–1880 (6.5%). Therefore, the reported extreme droughts for the whole of eastern China in 920–1300 (13.9%) was significantly more frequent than in 1310–1880 (9.6%). As for the reported frequency of extreme floods, significant differences were only found in the south. Compared with 1310–1880, there were significantly fewer extreme floods in Jianghuai but more in Jiangnan from 920 to 1300. After the composition of their opposite tendency, no significant differences in the frequency of extreme floods between 920–1300 and 1310–1880 were found for the whole of eastern China. Thus, there were slightly more reported extreme droughts during the warm period than the cold period, while no differences were found in the frequency of reported extreme floods over eastern China.

As found in section 3.2, higher frequency of reporting of extreme droughts in eastern China was significantly associated with reconstructed poor harvests (grade 1+2) when compared with non-extreme years. Since there were more extreme droughts over eastern China in 920–1300 than in 1310–1880 in the reconstructed drought/flood dataset, the reconstructed harvest in the warm epoch could be expected to be worse than in the cold epoch. However, as Yin et al. (2015, 2016) found, the reconstructed harvest in warm epoch was better than that in cold epoch. This suggests that the patterns in association between reported frequency of regional extreme drought and reconstructed grain harvest differed between warm and cold epochs. Table 6 illustrates this difference, by showing the frequency of poor harvests (grade 1+2) in extreme drought/flood years and non-extreme years for warm and cold epochs, i.e., 920–1300 and 1310–1880. Noted that there exists a shift in the distribution of each harvest grade after the 1640s due to an increase in grade 4 years and a decrease in missing data years. Meanwhile, there existed an evidently high frequency of bumper harvests (grade 5+6) from the 1650s to the 1810s.

Therefore, the patterns in association between reconstructed poor harvests (grade 1+2) and reported frequency of regional extreme droughts for 1650–1880 were also investigated.

The results show that, during the warm epoch of 920–1300, the reconstructed poor harvest was not significantly associated with regional reported frequency of extreme droughts, although the frequency of poor harvest in extreme drought years was slightly higher than in non-extreme years for each sub-region. In contrast, during the cold epoch of 1310–1880, the frequency of poor harvest in extreme drought years was significantly higher than in non-extreme years, which indicates a significant association in reconstructed poor harvest and reported frequency of extreme droughts. Moreover, similar characteristics were found for the latter half of the cold period from 1650 to 1880, which indicates that the shift of harvest grade distribution did not affect the association between reconstructed poor harvest and reported frequency of extreme droughts/floods during the cold epoch.

## 4 Discussion

Chinese historical documents provide a unique proxy for reconstructions of climate change and annual harvest grades, which helps to explore the impacts of climate change on grain yields up to thousand years ago. Compared with previous research (e.g., Su et al., 2014; Yin et al., 2015, 2016), which focused on the correlation between reconstructed temperature variation and grain harvest fluctuations at decadal resolution during historical times, this study presented an examination of the association between reported frequency of extreme drought/flood and reconstructed grain harvests and its different pattern between warm and cold periods year by year. The statistical results from section 3.2 indicate that the reported frequency of regional extreme droughts is closely associated with poor harvests in reconstruction, while no significant pattern is found between the reported frequency of extreme floods and reconstructed poor harvests. This phenomenon might be explained by the fact that extreme droughts usually cover an immense area dominated by a single large-scale air mass, leading to significant and extensive impacts on agriculture, economics, and many other social factors. On the other hand, extreme floods were mostly caused by rainstorms induced by the confrontation of air masses, which usually occur across a relatively narrow belt. Meanwhile, rainstorms could irrigate agricultural land in areas surrounding the extreme floods and, thus, improve grain yields, leading to limited impacts of extreme floods on harvests over an immense area (Zhang, 1982). Although it should be noted that these meteorological interpretations only apply for short-term extreme flood events. When it comes to long-term wet periods, it could as well be responsible for bad harvest, due to the low-temperature and short sunlight.

Section 3.3 showed the statistical results for the different pattern in association between reported frequency of regional extreme droughts and reconstructed grain harvests between warm and cold periods. These results suggest that the simultaneously occurrence of reported extreme droughts and reconstructed poor harvests tends to be weakened in a warm climate background. Possible cause might be that, there was low and limited adaptability during that period, and the warm climate provided more thermal resources and extended the growing season, thus increasing the multiple cropping index and providing more thermal-limited lands for growing crops. This gave people more options to adapt to climatic variation, and

mitigated the impacts of extreme droughts on harvest yield. As assessed by Zhang (1982), the harvest may change by approximately 10% if the temperature changed by 1 ℃ on national scale based on the data from 1909 to 1979, in which the harvest increased significantly in 7 out of 8 warm years. However, a cold climate could limit multiple cropping and shrink the area of arable land, leading to the harvest becoming more vulnerable to extreme drought. Moreover, as reported by Zhang et al. (2007), limited resources could also cause social turbulence, such as famine, peasant uprising, the outbreak of war, and population decline, all of which may further increase agricultural vulnerability. Therefore, even though the frequency of reporting of extreme droughts was slightly higher in the warm period of 920–1300, the frequency of reconstructed poor harvests did not increase significantly.

However, these inferences are purely based on those two reconstructed datasets, and insufficient to reveal actual historical connections. One of the reasons is that both datasets used *Twenty-Four Histories* and *Qing History Draft* as their record resources in the reconstruction, which might induce artefact in databases and lead to spurious correlations between extreme drought/flood and harvest. Another reason is that there existed major shifts in many social factors (such as political, economic, demographic or cultural change) in the study period, which means that the different pattern in association between reported extreme drought and reconstructed poor harvests between warm and cold periods might be co-created by those confounding factors along with climate factor. Although the dataset for extreme droughts/floods used more historical documents and were not limited to the *Twenty-Four Histories* and *Qing History Draft* as resources, and both datasets have been validated by other proxy data in previous studies, the independence and consistency problems could not be eliminated entirely. The solution for these two problems requires further research using more independent datasets from multi-proxies and many other study fields, and could be of great importance in improving the understanding of the climatic impacts on agriculture and adaptation to future global warming along with higher extreme climate probability.

Meanwhile, several issues still exist that could lead to uncertainty in the statistics. First, due to missing records before 1400, the reconstructed chronology of regional extreme drought and flood may not include all events. For example, as shown in Fig. 3, the reconstructed chronologies of regional extreme drought and flood for several periods (e.g., 850–950, 1350–1400 in both Jianghuai and Jiangnan, and 1250–1300 in Jianghuai) had low confidence levels. Moreover, the chronologies for 1150–1200 in the North China Plain, 1200–1250 in Jianghuai, and 800–850, 950–1100 in Jiangnan also had medium confidence levels. Therefore, most of the reconstructed chronology of extreme drought and flood for the whole of eastern China before 1400 was only in medium or low confidence. Second, there may also be limitations in the grading of annual harvests for some years. For example, in 1181, the harvest was assessed as grade 6, according to the description, "It is a plentiful year after sufficient rainfall killed locusts" in the *History of Song* (one of the *Twenty-Four Histories*), which usually recorded country-wide events. Thus, this description was regarded as a record at national scale. However, the *Compilation of Essential Regulations of the Song Dynasty* (*Song Hui-Yao Ji-Gao* in Chinese, edited by Xu Song, Vol. 66, Vol. 69), reported widespread "drought" in southern China that year, which led to poor or slightly poor harvests at the sub-regional scale. Similar inconsistencies in the historical records have also been found in 1032, 1073, 1353, and 1638. These cases suggest that the grain harvest descriptions recorded in *Twenty-Four Histories* may not indicate anomalous harvests for all years at the

national scale. In addition, around the 1650–60s, there was a clear jump to grade 5+6 (bumper harvest), yet this period is commonly recognized in previous literatures as having poor harvests and famine in this coldest interval of the Little Ice Age. Also, the periods 1130–1150 and 1210–1270 (the early and later Southern Song Dynasty), 880–980 (later Tang Dynasty and Five Kingdoms) and around 1400, have remarkably more missing harvest grade data. These limitations introduce uncertainty

by affecting the statistical relationship between harvest and extreme drought/flood in certain periods. Therefore, further research is required, especially in the search for more historical records from the other scattered documents to supplement the evidence recorded in the *Twenty-Four Histories*.

**5 Conclusion**

Variation in the reported frequency of regional extreme drought/flood and reconstructed grain harvest, and possible

association between the two, were investigated for 801–1910, using a reconstructed chronology of regional extreme drought/flood and a series of grain harvest grades. The results reveal significantly higher reported frequency of extreme droughts in 801–870, 1031–1230, 1481–1530 and 1581–1650, and higher reported frequency of extreme floods in 811–840, 951–990, 1051–1070, 1381–1430, 1641–1670, 1721–1770 and 1810–1870 over the whole of eastern China. Regionally, more extreme droughts were reported in different intervals, including 1031–1100, 1441–1490, 1601–1650 and 1831–1880

for North China, 801–870, 1031–1120, 1161–1220 and 1471–1530 for Jianghuai, 991–1040, 1091–1150, 1171–1230, 1411– 1470 and 1481–1530 for Jiangnan. The grain harvest was reconstructed to be generally poor in, approximately, 801–940, 1251–1650 and 1841–1910, but bumper in 951–1250 and 1651–1840. Both t-test and Chi-square test ($\chi^2$) demonstrated that higher frequency of reporting of extreme droughts in any sub-region of eastern China are significantly associated with reconstructed poor harvests. Meanwhile, the harvest was reconstructed to be much worse when extreme drought and extreme

flood were reported in the same year in different sub-regions of eastern China. However, the association between reported frequency of extreme flood and reconstructed harvest yield was not detected in the long-term average. Moreover, the comparison showed the reported frequency of extreme droughts over the whole of eastern China was higher in the warm epoch of 920–1300 than that in the cold epoch of 1310–1380. However, during the warm epoch, the association between reconstructed poor harvests and reported frequency of regional extreme droughts was weak, though the frequency of poor

harvests in extreme drought years was still slightly higher than in non-extreme years for each sub-region. This could be explained by the provision of more thermal resources and the extended growing season in the warm climate, which increased the multiple cropping index and created more thermal-limited lands for agricultural crops, allowing for more options to mitigate the impacts of extreme drought on the harvest. Thus, the study provides further historical evidence to help explore the implications for agricultural adaptation to future global warming along with higher extreme climate probability.

**Data availability.** All data used in the paper are available from the corresponding authors on request.

**The Supplement related to this article:** Table S1, Table S2 and Fig. S1

**Author contribution.** JZ, ZH & SL contributed the idea and design the structure of manuscript; JZ & SL collected the data; ZH, MW, JC & XZ analyzed the data; ZH, JZ & MW wrote the manuscript.

**Competing interests.** The authors declare that they have no conflict of interest.

**Acknowledgements.** This study was supported by the National Key R&D Program of China (2017YFA0603300), the National Natural Science Foundation of China (41671036, 41831174) and the Strategic Priority Research Program of the Chinese Academy of Sciences (XDA19040101).

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

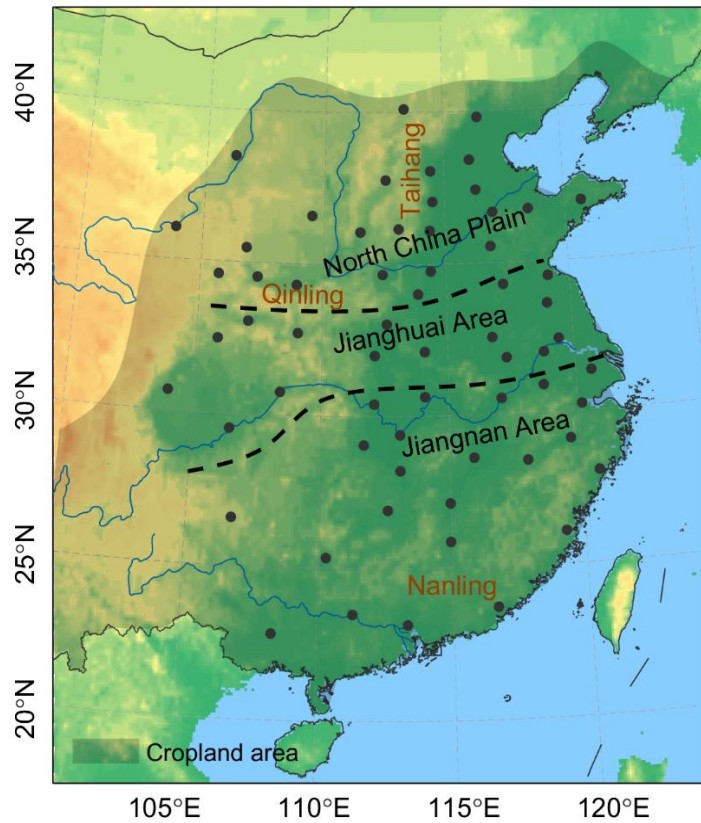

**Figure 1: Location of 63 drought/flood grade data stations and sub-region divisions. The shaded area indicates approximate cropland distributed approximately from 801 to 1910.**

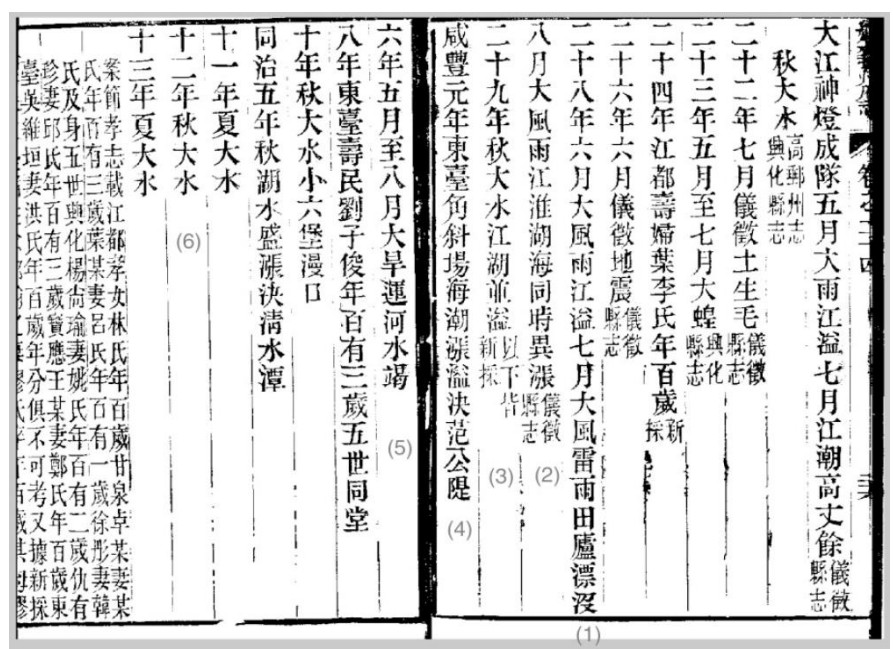

Figure 2: An example of anomalous climate information recorded in local gazettes (Quoted from Gazettes of Yangzhou Prefecture, compiled in 1874). The two pages list anomalous climate information in that region from 1842 to 1874, read from right to left, and dated according to the Chinese lunar calendar. Numbers in brackets are added by the authors to show meaning: (1) The 28th year (of Daoguang emperor, 1848), the 6th month, due to the strong wind and heavy rain, the Yangtze River overflowed; the 7th month, because of strong wind and thunder storms, fields and houses were submerged. (2) The 8th month, strong wind and heavy rain, the level of the Yangtze River, the Huaihe River, the (Gaoyou) Lake, and the sea level abnormally rose at the same time, citation from Gazettes of Yizheng County (a county in the prefecture), where records of "strong wind" and "the sea level abnormally rose" also suggested heavy rain induced by a typhoon. (3) There was a flood in the autumn of the 29th year (1849), the Yangtze River and the (Gaoyou) Lake overflowed at the same time. Notes of "Below are newly collected", suggested that these records were quoted from earlier local gazettes. (4) The 1st year of Emperor Xianfeng (1851), Jiaoxiechang (the name of a salt field) of Dongtai County (a county in the prefecture), the tide from the sea overflowed and broke the Sea Wall of Mr. Fan. (5) The 6th year (1856), the 5th to 8th months, there was severe drought and the Grand Canal dried up. (6) Autumn of the 12th year of Emperor Tongzhi (1873), there was a flood.

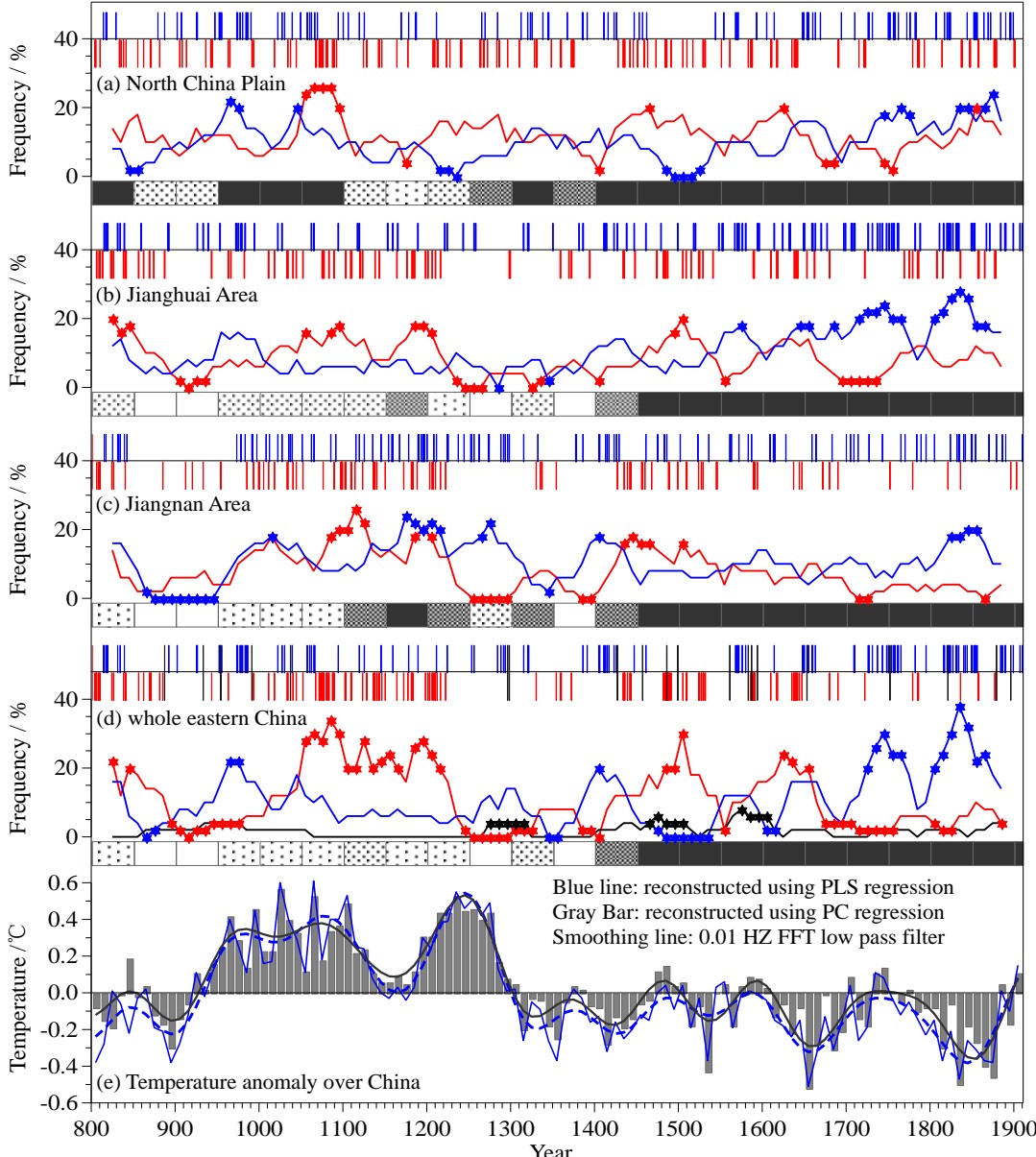

**Figure 3: The chronology (vertical line in the top row of each plate) and frequency (star-line, 50-year moving-window with decadal time step) of extreme drought (red) and flood (blue) the North China Plain (a), Jianghuai (b), Jiangnan area (c), and the whole of eastern China (d) from 801 to 1910. Black line in plate (d) indicates the same frequency for the co-reporting of extreme drought and flood for the whole of eastern China. The stars represent intervals with significantly more or less extreme drought and flood compared with all other intervals, based on the Wilcoxon rank sum test. The bars on the bottom row of each plate illustrate confidence levels (probability in being correct, PBC) for each reconstructions at 50 years intervals: extremely high confidence (PBC>99%): dark; very high confidence (PBC>90%): 50% shaded dark; high confidence (PBC>80%): 25% shaded dark; medium confidence (PBC>50%): 12.5% shaded dark; and low confidence (PBC>33.3%): blank. (e) The temperature series over China (at decadal resolution) for comparison, which was reconstructed based on 28 temperature proxies using principal component (PC) regression and partial least squares (PLS) regression (Ge et al., 2013).**

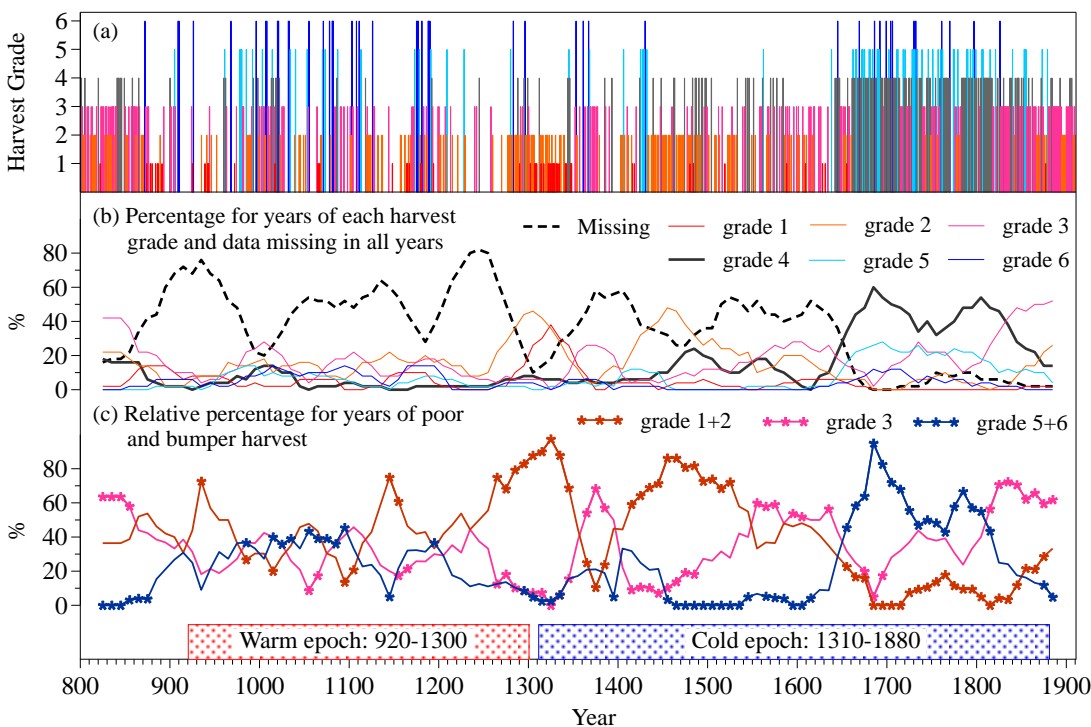

**Figure 4: (a)** The reconstructed yearly harvest grade from 801–1910, 1-Very poor (red), 2-Poor (orange), 3-Slightly poor (pink), 4-Average (black), 5-Near bumper (light blue), 6-Bumper (blue). **(b)** The variation of percentage for the years with each harvest grade and missing data, using a moving-window of 50 years by decadal step. **(c)** Relative percentage for years of poor (grade 1+2), slightly poor (grade 3) and bumper (grade 5+6) harvest compared with total years, excluding "average" harvest and missing data years, using a moving-window of 50 years by decadal step. Stars: intervals with significantly higher or lower percentage compared with all other intervals, using the Wilcoxon rank sum test. Box: the warm and cold epochs identified from the temperature reconstruction for all of China, shown in Figure 2e.

**Table 1: Frequency of reported extreme drought and flood in each sub-region and the whole of eastern China, 801–1910.**

| Sub-region | Extreme event | Number of years | Frequency (%) |
|---|---|---|---|
| North China Plain | Drought | 133 | 12.0 |
| | Flood | 113 | 10.2 |
| Jianghuai area | Drought | 95 | 8.6 |
| | Flood | 118 | 10.6 |
| Jiangnan area | Drought | 90 | 8.1 |
| | Flood | 119 | 10.7 |
| Whole eastern China | Drought | 126 | 11.4 |
| | Flood | 122 | 11.0 |
| | Co-reporting of Drought and Flood | 20 | 1.8 |

**Table 2: Number of years and percentage for each harvest grade and missing data, 801-1910.**

| Harvest grade | 1 | 2 | 3 | 4 | 5 | 6 | Data missing |
|---|---|---|---|---|---|---|---|
| Number of years | 54 | 178 | 204 | 160 | 77 | 47 | 390 |
| % in all years | 4.9 | 16.0 | 18.4 | 14.4 | 6.9 | 4.2 | 35.1 |
| % in years excluding grade 4 and missing data | 9.6 | 31.8 | 36.4 | - | 13.8 | 8.4 | - |

**Table 3: Comparison between mean harvest grades in extreme drought/flood years and non-extreme years for the three sub-regions and the whole of eastern China.**

| | Non-extreme | Extreme drought | Extreme Flood | Co-reporting [#] |
|---|---|---|---|---|
| North China Plain | 3.31 | **2.84**[***] | 3.20 | -- |
| Jianghuai Area | 3.27 | **2.99**[*] | 3.23 | -- |
| Jiangnan Area | 3.28 | **2.89**[**] | 3.21 | -- |
| Whole of eastern China | 3.30 | **2.95**[**] | 3.27 | **2.53**[**] |

Note: Significance level: [***], $p<0.01$; [**], $p<0.05$; [*], $p<0.05$. [#]: The co-reporting of extreme drought and extreme flood in different sub-

5   regions.

**Table 4: Comparison of the frequency of each harvest grade between extreme drought/flood years and non-extreme years, 801–1910.**

| Region | Harvest grade | | 1 | 2 | 3 | 4 | 5 | 6 | 1+2[a] | 5+6[b] | Total |
|---|---|---|---|---|---|---|---|---|---|---|---|
| North China Plain | Non-Extreme | Years | 34 | 124 | 158 | 129 | 57 | 39 | 158 | 96 | 541 |
| | | % | 6.3 | 22.9 | 29.2 | 23.8 | 10.5 | 7.2 | 29.2 | 17.7 | |
| | Extreme drought | Years | 13 | 33 | 22 | 17 | 9 | 3 | 46 | 12 | 97 |
| | | % | **13.4**** | **34.0**** | 22.7 | 17.5 | 9.3 | 3.1 | **47.4**\*** | 12.4 | |
| | Extreme flood | Years | 7 | 21 | 24 | 14 | 11 | 5 | 28 | 16 | 82 |
| | | % | 8.5 | 25.6 | 29.3 | 17.1 | 13.4 | 6.1 | 34.1 | 19.5 | |
| Jianghuai Area | Non-Extreme | Years | 39 | 137 | 153 | 127 | 68 | 34 | 176 | 102 | 558 |
| | | % | 7.0 | 24.6 | 27.4 | 22.8 | 12.2 | 6.1 | 31.5 | 18.3 | |
| | Extreme drought | Years | 8 | 23 | 21 | 12 | 4 | 6 | 31 | 10 | 74 |
| | | % | 10.8 | 31.1 | 28.4 | 16.2 | *5.4** | 8.1 | **41.9*** | 13.5 | |
| | Extreme flood | Years | 7 | 18 | 30 | 21 | 5 | 7 | 25 | 12 | 88 |
| | | % | 8.0 | 20.5 | 34.1 | 23.9 | *5.7** | 8.0 | 28.4 | 13.6 | |
| Jiangnan Area | Non-Extreme | Years | 42 | 129 | 158 | 134 | 67 | 34 | 171 | 101 | 564 |
| | | % | 7.4 | 22.9 | 28.0 | 23.8 | 11.9 | 6.0 | 30.3 | 17.9 | |
| | Extreme drought | Years | 6 | 28 | 14 | 8 | 5 | 5 | 34 | 10 | 66 |
| | | % | 9.1 | **42.4**\*** | 21.2 | *12.1**** | 7.6 | 7.6 | **51.5**\*** | 15.2 | |
| | Extreme flood | Years | 6 | 21 | 32 | 18 | 5 | 8 | 27 | 13 | 90 |
| | | % | 6.7 | 23.3 | 35.6 | 20.0 | *5.6** | 8.9 | 30.0 | 14.4 | |
| Whole eastern China | Non-Extreme | Years | 35 | 115 | 146 | 126 | 59 | 32 | 150 | 91 | 513 |
| | | % | 6.8 | 22.4 | 28.5 | 24.6 | 11.5 | 6.2 | 29.2 | 17.7 | |
| | Extreme drought | Years | 11 | 34 | 21 | 13 | 7 | 8 | 45 | 15 | 94 |
| | | % | **11.7*** | **36.2**\*** | 22.3 | *13.8**** | 7.4 | 8.5 | **47.9**\*** | 16.0 | |
| | Extreme flood | Years | 7 | 20 | 35 | 19 | 10 | 7 | 27 | 17 | 98 |
| | | % | 7.1 | 20.4 | 35.7 | 19.4 | 10.2 | 7.1 | 27.6 | 17.3 | |
| | Co-occurrence[#] | Years | 1 | 9 | 2 | 2 | 1 | 0 | 10 | 1 | 15 |
| | | % | 6.7 | **60.0**\*** | 13.3 | 13.3 | 6.7 | 0.0 | **66.7**\*** | 6.7 | |

Values in bold with stars denote significant more or less (also in italic) reported extremes, using Chi-test ($\chi^2$) at level of: ***, $p<0.01$; **, $p<0.05$; *, $p<0.1$. [a]: for grade 1 and grade 2 in total. [b]: for grade 5 and grade 6 in total. [#]: for the co-reporting of extreme drought and extreme flood in different sub-regions.

**Table 5: Comparison of frequency of reported extreme drought/flood between warm and cold epochs.**

| Sub-Region | | Frequency (%) | |
|---|---|---|---|
| | | Warm epoch 920–1300 | Cold epoch 1310–1880 |
| North China Plain | extreme drought | 12.6 | 11.7 |
| | extreme flood | 9.4 | 11.2 |
| Jianghuai Area | extreme drought | 9.2 | 7.9 |
| | extreme flood | **7.1**[***] | **13.3**[***] |
| Jiangnan Area | extreme drought | **10.8**[**] | **6.5**[**] |
| | extreme flood | **13.6**[*] | **9.8**[*] |
| whole eastern China | extreme drought | **13.9**[**] | **9.6**[**] |
| | extreme flood | 9.7 | 11.9 |

Values in bold with stars denote significantly more or less (also in italic) frequent reported extreme droughts/floods in 920–1300 compared with 1310–1880, using rank sum test at level of: [***], $p<0.01$; [**], $p<0.05$; [*], $p<0.1$.

**Table 6: Comparison of frequency of reconstructed poor harvest (grade 1+2) for extreme drought years and non-extreme years for warm and cold epochs.**

| Region | Anomalous temperature epochs | | Warm epoch 920–1300 | | Cold epoch 1310–1880 | | in which 1650–1880 | |
|---|---|---|---|---|---|---|---|---|
| | Harvest grade | | 1+2 | others | 1+2 | others | 1+2 | others |
| North China Plain | Non-Extreme | Years | 53 | 90 | 78 | 233 | 6 | 159 |
| | | % | 37.1 | 62.9 | 25.1 | 74.9 | 3.6 | 96.4 |
| | Extreme drought | Years | 13 | 13 | 24 | 31 | 5 | 17 |
| | | % | 50.0 | 50.0 | **43.6**$^{***}$ | 56.4 | **22.7**$^{***}$ | 77.3 |
| Jianghuai Area | Non-Extreme | Years | 57 | 90 | 86 | 237 | 10 | 154 |
| | | % | 38.8 | 61.2 | 26.6 | 73.7 | 6.1 | 93.9 |
| | Extreme drought | Years | 12 | 12 | 16 | 20 | 4 | 12 |
| | | % | 50.0 | 50.0 | **44.4**$^{**}$ | 55.6 | **25.0**$^{***}$ | 75.0 |
| Jiangnan Area | Non-Extreme | Years | 49 | 79 | 86 | 254 | 15 | 173 |
| | | % | 38.3 | 61.7 | 25.3 | 74.7 | 8.0 | 92.0 |
| | Extreme drought | Years | 13 | 14 | 19 | 11 | 1 | 5 |
| | | % | 48.2 | 51.8 | **63.3**$^{***}$ | 36.7 | 16.7 | 83.3 |
| Whole eastern China | Non-Extreme | Years | 48 | 79 | 76 | 229 | 8 | 156 |
| | | % | 37.8 | 62.2 | 24.9 | 75.1 | 4.9 | 95.4 |
| | Extreme drought | Years | 15 | 20 | 23 | 20 | 3 | 8 |
| | | % | 42.9 | 57.1 | **53.5**$^{***}$ | 46.5 | **27.3**$^{**}$ | 72.7 |

Values in bold with stars denote more reported droughts/floods significantly, using Chi-test ($\chi^2$) at level of: $^{***}$, p<0.01; $^{**}$, p<0.05; $^{*}$, p<0.1.

