# Peer review of "Patterns in data of extreme droughts/floods and harvest grades derived from historical documents in Eastern China during 801– 1910"

_Climate of the Past, 2019_

## Referee Comment (RC1) · Anonymous Referee #1 · 20 Feb 2019

Using databases of Song, Ming, and Qing documents, this paper finds that the frequency of reports of extreme droughts—but not always floods—correlates with reductions in harvests as reported in historical sources. On this basis, the authors conclude that there are clear historical periods when droughts reduced harvests, and therefore that these events had significant societal impacts. The sources and methods used in this manuscript appear to be standard in other publications on Chinese historical climatology. In this instance, however, I am not convinced they are adequate to prove the authors' conclusion. The problems concern, first, the author's use of their historical databases; second, the large temporal and spatial scale of the study; and third, the interpretation given to the pattern of correlations found.
Problems in use of databases:

The authors' use of databases of flood and drought events and harvest grades raises numerous questions which must be answered before it is clear whether or not the correlations identified are valid: 1) What kinds of droughts are recorded in the historical sources: meteorological drought? hydrological drought? agricultural drought? or some combination of these? Were observers more likely to report precisely those droughts that affected crops, or did they report all droughts equally? 2) What kinds of floods are recorded in the historical sources: heavy rains? tsunamis? rivers that burst their banks? Does the database control for ongoing problems related to river hydrology? How do major events such as course changes in the Yellow River figure into the measure of flood frequency: as one flood? as many? 3) What is the seasonality of the meteorological events recorded in the historical sources? Does the seasonality of floods or droughts necessarily overlap with the seasonality of critical agricultural activities or phases of crop growth? 4) What is being measured by "harvest"? Yield per seed? Total yield per hectare? Food availability? 5) Are degrees of flood, drought, and harvest based entirely on narrative descriptions, or are there objective phenological or quantitative measures to help define them?

Regarding the temporal and spatial scale of the study, I am concerned that it relies on improbable assumptions of continuity and homogeneity in Chinese population, land use, and record keeping. In order to accept as valid any long-term correlations between reported drought or flood frequency and "Chinese" or even "regional" "harvests" I would need the authors to address the following issues: 1) How do the data control for the changing borders of Chinese empires? A priori, I would expect vastly different vulnerabilities and patterns of reporting between the Northern Song and Southern Song periods, simply based on the major geographical shifts in population and wealth between those two dynasties. 2) How do data on "harvests" control for changes in staple crops, introduction of New World crops including peanuts and sweet potatoes, changing cropping patterns, and the increasing commercial orientation of agriculture?

3) How do the data deal with the changing vulnerabilities to climate variability based on changing settlement patterns even within regions (e.g., uplands in the south and southwest colonized by Han settlers during the late Ming and Qing periods)? 4) Given the very long time period examined here, wouldn't we expect new adaptations to reduce vulnerabilities to predictable climate variability and disasters? 5) Most importantly, how can we make up for the fact there are simply more records from the Qing period than earlier periods? I don't see that the methods used in this manuscript avoid the problem that more records will create a misimpression of a greater frequency of floods and droughts. The authors propose to ignore reports of "average" conditions in Qing records to make them more comparable to Song and Ming records. However, that would only work if the Song and Ming records still reliably reported all disasters and extremes and only left out "average" conditions. I don't see any reason to make that assumption. Perhaps the authors could experiment with methods of introducing "noise" into the data in order to reflect the events missing from the reports. Or else they could employ a Bayesian method to indicate that the presence or absence of certain descriptions in the records may be used to obtain updated posterior probabilities of actual conditions, without ever assuming that the records provide a complete account of events. In any case, the authors must come up with a way to handle these changes in the documentary record over time if they are to make a convincing case for stable long-term correlations between floods and droughts and harvests.

Third, even if the correlations found in the study are valid, there is a problem with the authors' historical interpretation of them. The correlations discovered here are not between climate and harvests, but rather reports of floods and droughts and reported harvests. The authors assume that the correlations mean that floods and droughts reduced harvests. However, there are a number of potentially confounding variables, which indicate other potential pathways of causality and therefore other historical possibilities: 1. Drought and/or flood might have correlated with other climate variables (such as temperature) that caused harvest failure. 2. Drought and/or flood might have increased the likelihood that officials reported problems such as poor harvests and

other disasters 3. Harvest failures might have increased the likelihood that officials reported disasters such as droughts and/or floods. 4. Droughts and/or floods might have harmed human and animal health, reducing labor for harvests. 5. Droughts and/or floods might have damaged infrastructure and transportation, leading to food availability decline. 6. Droughts and/or floods might have driven migrations, creating regional shortages both where agricultural labor emigrated and where people arrived seeking food. 7. Periods of drought and/or flood might have reduced public revenue and/or increased public expenses, thus increasing the political and economic instability and decreasing food availability. (For instance, it's not clear how much the figures overall are influenced by the very high frequency of disasters and widespread famine during the political turbulence and violence accompanying the collapse of the Ming dynasty.) I am not arguing that any of these scenarios is necessarily the case. Nevertheless, each of these may be influencing the observed correlations.

In summary, I do not believe that the authors' database and methods currently prove a valid correlation between flood and drought frequency and harvests in imperial China, nor that such a correlation would prove that drought or flood reduced harvest yields. The problem is not that the authors' hypothesis is unreasonable. It is simply that the conditions and data are too heterogeneous over such a large spatial and temporal scale. Any correlations found on such a scale are likely to have arisen from some artefact of the record-keeping or through the influence of some confounding variable, rather than to reflect a real and consistent climatic impact on agriculture.

Nevertheless, I would not like to dismiss this study out of hand. These datasets still have tremendous potential for historical climatology research. Better statistical methods could be devised to deal with changes in the frequency of historical reporting. By bringing trained historians onto such a project, the authors might find ways to handle problems related to historical changes in Chinese population, politics, land use, and economy. I would like to see the authors successfully address such problems in their research

Technical notes: The paper variously sometimes to geographical parts of the country (e.g., “Northeast China”) and sometimes to regional designations (e.g., “Jiangnan”). The paper would be clearer if it stuck with regional designations and names of provinces only. The paper also needs extensive editing for English language grammar, spelling, and correct syntax. This is not merely a stylistic issue. The meaning of several passages is unclear due to lack of clear and correct English usage.

---

## Referee Comment (RC2) · Anonymous Referee #2 · 7 May 2019

This is a nice piece of paper, well-structured with clear scoping and good delineation. It investigates the time evolution of extreme drought/flood events and the correlations of those extreme events with crop harvest in cold/warm epochs. In general, this paper provides very knowledgeable information on the drought/flood and harvest reconstruction method derived from documentary records, very comprehensive literature review in sections 1 and 2.1. However, there are still some points that I would suggest for authors to further improve the scientific quality and literacy of the paper.

Data used for analysis in this study is based on the previous analysis. The rationale for deciding the drought/flood (Zheng et al. 2006 and Hao et al. 2016) and harvest

grades (Su et al, 2014; Yin et al, 2015) seemed promising however readers need to trace back those papers for more information on the source and profiles of data. There thus exists an ambiguity about how those records were collected for building the data sets, the characteristics and amounts of records, and data reliability evaluation. At this point, I can only assume that the data are reliable for the following analysis. It can be helpful if the authors provide some basic statistics (as appendix maybe, e.g. number of records per year, min, max etc for variables) of the data profile.

Based on the data, methods used for analysis is relatively simple. The authors used 50-years moving average (they term it as moving window) to smoothen extreme drought/flood trend, used Wilcoxon rank test to examine/compare median values of every intervals, and used contingency table to examine the effects between extreme drought/flood and harvest and between them and cold/warm periods. For this section, I suggest the authors to add a short paragraph giving readers some concepts about the method structure before going into details. Also there are some unclear parts: what do you mean by saying 'moving-window of 50 years and step of 10 years (line 24-25, page 5, and figure 3 caption, page 18)'? Please provide explanations. For the same figure 3 caption, please use real number to replace full confidence, high confidence, medium or low confidence. It is unclear what those mean. Also, I don't quite understand the sentence in line 26-28 page 5 "This is because the mean of rank series in an interval was equivalent to the frequency of ....by labeling the extreme drought/flood years as 1 and non-extreme years as 0". Please provide more explanations.

The research results are clear and straightforward. The drought, flood and harvest trends and their descriptions are clear. However some trends are inconsistent with previous studies. For example, the authors mentioned that "there was an evident jump around 1640s with increase of years of (harvest) grade 4..." (line 12-13 page 7). I admire the following sentences on the discussions of the records in Qing and previous dynasties to clarify the discrepancies of the historical books. However, after removing grade 4 (average harvest), there still existed an obvious jump of grade 5&6 (bumper)

after 1640 which was commonly recognized in previous literatures as having poor harvest and famine in this coldest interval of little ice age. There might be some reasons including suddenly increasing number of local chronicles in Qing dynasty that could dilute the drought magnitude based on your grading method or the 50-years moving average can further smoothen the trend. In a word, it will be extremely valuable if the authors can compare the present analysis with previous studies and provide explanations or new perspectives.

Overall, this paper provides new and important insights into the correlations among extreme event, harvest and cold/warm climate. Data statistic is suggested to provide, and since missing data especially for harvest is prominent (35%), it is suggested that authors are more careful to claim their conclusions. Some inconsistency is also found between text and tables, e.g. 49.4% (line 23) and 24.0% (line 24) on page 8 are different from those shown in table 4. Further English editing is strongly suggested to improve high quality writing style of this nice paper.

---

## Author Comment (AC1) · 7 Jun 2019

**Reply to the referee comments on "Extreme droughts/floods and their impacts on harvest derived from historical documents in Eastern China during 801–1910" by Zhixin Hao et al**

Dear editors and reviewers,

Thank you for your valuable comments and thoughtful suggestions on our manuscript. Following your comments on the manuscript, we made careful revisions, and the point-to-point response of the comments is listed below. We hope these revisions would make this manuscript more acceptable for publication. Please feel free to contact me if you have any questions.

Many thanks again. With best wishes.

Sincerely yours,

Jingyun Zheng

**Anonymous Referee #1**

**Using databases of Song, Ming, and Qing documents, this paper finds that the frequency of reports of extreme droughts (but not always floods) correlates with reductions in harvests as reported in historical sources. On this basis, the authors conclude that there are clear historical periods when droughts reduced harvests, and therefore that these events had significant societal impacts. The sources and methods used in this manuscript appear to be standard in other publications on Chinese historical climatology. In this instance, however, I am not convinced they are adequate to prove the authors' conclusion. The problems concern, first, the author's use of their historical databases; second, the large temporal and spatial scale of the study; and third, the interpretation given to the pattern of correlations found.**

**Problems in use of databases:**

**The authors' use of databases of flood and drought events and harvest grades raises numerous questions which must be answered before it is clear whether or not the correlations identified are valid:**

**1) What kinds of droughts are recorded in the historical sources: meteorological drought? hydrological drought? agricultural drought? or some combination of these? Were observers more likely to report precisely those droughts that affected crops, or did they report all droughts equally?**

Accepted and revised. Most of droughts recorded in historical documents were events due to no or less precipitation, and could be regarded as meteorological droughts. Therefore it's reasonable to suggest that they reported all droughts equally rather than those affected crops. (P3, L23-28)

*From these samples, it's easy to know that the droughts recorded in the historical documents were usually focused on the events due to no or less precipitation in a period at first, thus could be regarded as meteorological drought rather than hydrological drought or agricultural drought, although some records also reported the impacts on hydrosphere (e.g., dry up for river) or agriculture (e.g. wilting for crops). Therefore, the droughts recorded in the historical documents appear to report all droughts equally rather than those affected crops only.*

**2) What kinds of floods are recorded in the historical sources: heavy rains? tsunamis? rivers that burst their banks? Does the database control for ongoing problems related to river hydrology? How do major events such as course changes in the Yellow River figure into the measure of flood frequency: as one flood? as many?**

Accepted and revised. Similar to the records on drought, flood recorded in historical documents were mostly about more rains or heavy rains. River hydrology events were not taken into account unless it was resulted from more precipitation or heavy rain. (P3, L28-30)

*Similarly, the floods recorded in the historical documents could be regarded as more rains or heavy rains rather than the river that burst their banks or tsunamis, although some records also reported the impacts on the overflowing or bursting for lakes or river due to more precipitation or heavy rain.*

**3) What is the seasonality of the meteorological events recorded in the historical sources? Does the seasonality of floods or droughts necessarily overlap with the seasonality of critical agricultural activities or phases of crop growth?**

Accepted and revised. The 63-stations annual drought/flood grades was reconstructed and calibrated with descriptions on duration, intensity, and area of the drought/flood events in wet season, which was usually May to September in the study area and overlaps with the critical agricultural activities and phases of crop growth. (P3, L32-P4, L3)

*The grades were classified using the ideal frequency criteria of 10% (grade 1, severe drought), 20% (grade 2, drought), 40% (grade 3, normal), 20% (grade 4, flood), and 10% (grade 5, heavy flood) for whole area and all time, which were calibrated with the descriptions on duration, intensity, and area of the drought/flood events in wet season (usually May to September), and its impact (Table*

*S1). Thus, the season for the drought/flood grade data overlaps with the critical agricultural activities and phases of crop growth.*

**4) What is being measured by "harvest"? Yield per seed? Total yield per hectare? Food availability?**

Accepted and revised. Harvest in records was a relative concept and represented the ratio of actual yield compared to the possible maximum yield. (P6, L3-5)

*In Chinese historical documents, the yearly harvest was usually recorded as a relative level compared to expected maximum yield rather than the individual crop production per hectare, although some records also reported the impacts on food availability, tax remissions, people's livelihoods, etc.*

**5) Are degrees of flood, drought, and harvest based entirely on narrative descriptions, or are there objective phenological or quantitative measures to help define them?**

Accepted and revised. The degrees of flood, drought, and harvest was not based entirely on narrative descriptions. The duration, intensity, and area of these events and their impacts were adopted in calibration, as well. We added two supplementary tables (Table S1, S2) to explain the detailed criteria in grading drought/flood (P3, L31-P4, L2; Table S1) and harvest (P6, L14-17; Table S2). The supplementary materials are attached at the end of this text.

*Based on these records, Zhang (1996) reconstructed the dataset of 63-stations annual drought/flood grades from 137 BCE, in which each station was set as a local area consisted of about 20 counties with the same climate. The grades were classified using the ideal frequency criteria of 10% (grade 1, severe drought), 20% (grade 2, drought), 40% (grade 3, normal), 20% (grade 4, flood), and 10% (grade 5, heavy flood) for whole area and all time, which were calibrated with the descriptions on duration, intensity, and area of the drought/flood events in wet season (usually May to September), and its impact (Table S1).*

*In the dataset, the levels of yearly harvest were classified into 6 grades: 1-Very poor, 2-Poor, 3-Slightly poor, 4-Average, 5-Near bumper, 6-Bumper. The criteria and methods for grading the documentary records (i.e., grain yield descriptions and the related information) year by year were presented by Su et al. (2014) and summarized by Yin et al. (2015), in which the classification of the yearly harvest grade and descriptions recorded in historical documents was shown in Table S2.*

**Regarding the temporal and spatial scale of the study, I am concerned that it relies on improbable assumptions of continuity and homogeneity in Chinese population, land use, and record keeping. In order to accept as valid any long-term correlations between reported drought or flood frequency and "Chinese" or even "regional" "harvests" I would need the authors to address the following issues:**

**1) How do the data control for the changing borders of Chinese empires? A priori, I would expect vastly different vulnerabilities and patterns of reporting between the Northern Song and Southern Song periods, simply based on the major geographical shifts in population and wealth between those two dynasties.**

Accepted and revised. For droughts and floods, the historical records was transformed into graded data based on 63-stations, each of which was set as a local area consisted of about 20 counties and does not change in different dynasties (P3, L31-32). Although the available graded data was unevenly distributed spatially for different dynasties, it had been proved in the paper of Hao et al. (2010a) that the extreme drought/flood years recognized were mostly robust despite of the percentage of data-missing stations (P5, L5-12). As for harvest, the impact of changing borders on harvest grade should also be limited since the main grain product area had been relatively stable in the study period and the records in documents was about relative harvest rather than absolute yield as suggested by Yin et al. (2015) (P6, L30-33).

*Based on these records, Zhang (1996) reconstructed the dataset of 63-stations annual drought/flood grades from 137 BCE, in which each station was set as a local area consisted of about 20 counties with the same climate.*

*To verify the rationality of this method and criteria, the validation was conducted in the paper of Hao et al. (2010a) based on 10 extreme events identified according to the threshold with probabilities of 10% and 90% occurrences which was based on the series of precipitation observation in each sub-regions. In the validation, all or part of grade 3 stations were omitted deliberately, and only 40% or 60% stations with disaster or extreme grade was reserved without changing the drought-to-flood ratio within the available data. The results showed that the years of extreme drought and extreme flood identified by this method and criteria using grade data well matched with those extreme events identified by the precipitation data, except for one mismatched event of extreme flood that occurred in 1958 in North China Plain, which demonstrated that the method and criteria was reasonable.*

*However, these social factors should have limited influence on the yearly harvest documents since that the main grain product area, the staple crop, and the cropping system have been relatively*

*stable in the study period, and besides the records in documents was relative harvest rather than absolute yield (Yin et al., 2015).*

**2) How do data on "harvests" control for changes in staple crops, introduction of New World crops including peanuts and sweet potatoes, changing cropping patterns, and the increasing commercial orientation of agriculture?**

Accepted and revised. The records for harvests were usually about relative percentage compared to expected maximum yield rather than absolute yield, and thus it should not be influenced by these factors. (P6, L3-7)

*In Chinese historical documents, the yearly harvest was usually recorded as a relative level compared to expected maximum yield rather than the individual crop production per hectare, although some records also reported the impacts on food availability, tax remissions, people's livelihoods, etc. Therefore these harvest records excluded the differences on absolute yield between sub-regions with different climate, soil fertility, crop variety, etc., and the difference between historical periods with changing agricultural centre, farming technology, staple crops (Su et al., 2014).*

**3) How do the data deal with the changing vulnerabilities to climate variability based on changing settlement patterns even within regions (e.g., uplands in the south and southwest colonized by Han settlers during the late Ming and Qing periods)?**

Accepted and revised. The graded harvest data represents a nationwide status and since the main grain product area had been relatively stable in the study period, the expansion of agricultural area should have limited influence on the yearly harvest documents. (P6, L27-33)

*During the study period, there did exist several social factors that could influence the total yield in China, such as the changing borders of empires, the expansion of agricultural area (e.g., uplands in the south and southwest colonized by Han settlers during the late Ming and Qing periods), the updated crop varieties due to the introduction of New World crops (e.g., peanuts and sweet potatoes), the advanced agricultural management technology, and so on. However, these social factors should have limited influence on the yearly harvest documents since that the main grain product area, the staple crop, and the cropping system have been relatively stable in the study period, and besides the records in documents was relative harvest rather than absolute yield (Yin et al., 2015).*

**4) Given the very long time period examined here, wouldn't we expect new adaptations to reduce vulnerabilities to predictable climate variability and disasters?**

Accepted and revised. This question has also been addressed in the revisions responding to above questions. (P6, L3-7, L27-33).

*In Chinese historical documents, the yearly harvest was usually recorded as a relative level compared to expected maximum yield rather than the individual crop production per hectare, although some records also reported the impacts on food availability, tax remissions, people's livelihoods, etc. Therefore these harvest records excluded the differences on absolute yield between sub-regions with different climate, soil fertility, crop variety, etc., and the difference between historical periods with changing agricultural centre, farming technology, staple crops (Su et al., 2014).*

*During the study period, there did exist several social factors that could influence the total yield in China, such as the changing borders of empires, the expansion of agricultural area (e.g., uplands in the south and southwest colonized by Han settlers during the late Ming and Qing periods), the updated crop varieties due to the introduction of New World crops (e.g., peanuts and sweet potatoes), the advanced agricultural management technology, and so on. However, these social factors should have limited influence on the yearly harvest documents since that the main grain product area, the staple crop, and the cropping system have been relatively stable in the study period, and besides the records in documents was relative harvest rather than absolute yield (Yin et al., 2015).*

**5) Most importantly, how can we make up for the fact there are simply more records from the Qing period than earlier periods? I don't see that the methods used in this manuscript avoid the problem that more records will create a misimpression of a greater frequency of floods and droughts. The authors propose to ignore reports of "average" conditions in Qing records to make them more comparable to Song and Ming records. However, that would only work if the Song and Ming records still reliably reported all disasters and extremes and only left out "average" conditions. I don't see any reason to make that assumption. Perhaps the authors could experiment with methods of introducing "noise" into the data in order to reflect the events missing from the reports. Or else they could employ a Bayesian method to indicate that the presence or absence of certain descriptions in the records may be used to obtain updated posterior probabilities of actual conditions, without ever assuming that the records provide a complete account of events. In any case, the authors must come up with a way to handle these changes in the documentary record over time if they are to make a convincing case for stable**

**long-term correlations between floods and droughts and harvests.**

Accepted and revised. The method of ignoring "average" conditions is based on the hypothesis that the records on droughts and floods were omitted randomly and unbiased, which suggests that the relative drought-to-flood ratio in the available data would be close to that in actual history. As in the abovementioned revisions, the recognition for extreme drought and flood years was still effective even if 40% or 60% of the available data with disasters was omitted deliberately. (P5, L1-15)

*This is because that the probabilities for omitting drought and flood records were random and unbiased, despite of the increasing occurrences of missing data back in time; i.e. if there's one period with a large number of documents, it would be rich in both drought and flood records, and vice versa. Therefore the amount of missing data should not have a significant effect on the relative drought-to-flood ratio within the available data compared with that in actual history. To verify the rationality of this method and criteria, the validation was conducted in the paper of Hao et al. (2010a) based on 10 extreme events identified according to the threshold with probabilities of 10% and 90% occurrences which was based on the series of precipitation observation in each sub-regions. In the validation, all or part of grade 3 stations were omitted deliberately, and only 40% or 60% stations with disaster or extreme grade was reserved without changing the drought-to-flood ratio within the available data. The results showed that the years of extreme drought and extreme flood identified by this method and criteria using grade data well matched with those extreme events identified by the precipitation data, except for one mismatched event of extreme flood that occurred in 1958 in North China Plain, which demonstrated that the method and criteria was reasonable. This is because that the precipitation variability in eastern China is dominated by the east Asian summer monsoon (EASM), thus when an extreme drought or flood event occurs, the precipitation variation for stations within each sub-region usually share a similar relative magnitude.*

**Third, even if the correlations found in the study are valid, there is a problem with the authors' historical interpretation of them. The correlations discovered here are not between climate and harvests, but rather reports of floods and droughts and reported harvests. The authors assume that the correlations mean that floods and droughts reduced harvests. However, there are a number of potentially confounding variables, which indicate other potential pathways of causality and therefore other historical possibilities:**

**1. Drought and/or flood might have correlated with other climate variables (such as temperature) that caused harvest failure.**

Accepted. As elaborated in previous study, the relationship between temperature and harvest had been investigated by Yin et al. (2015, 2016), which suggested that there would be better harvest in warm climate. And our study, in section 3.2 of the original manuscript, found that more occurrence of extreme drought in eastern China could lead to significant increase of frequency of poor harvest (grade 1+2) compared with non-extreme years. To further examine whether the drought and/or flood are correlated with temperature change, and if so, how the drought and/or flood are correlated with harvest failure under different temperature backgrounds, we presented a study in section 3.3, and found that there were slightly more extreme droughts in the warm period. However, the connection between extreme droughts and poor harvest was not significantly close in the warm epoch, while it was more significant in the cold epoch. These results suggested that warm period could weaken the impact of extreme drought on poor harvest during historical times. (P11, L21-25; P12, L3-10)

*As found in section 3.2, more occurrence of extreme drought in eastern China could lead to significant increase of frequency of poor harvest (grade 1+2) compared with non-extreme years. Since there were more extreme droughts occurred over eastern China in 920–1300 compared to 1310–1880, the harvest in warm epoch should be worse than that in cold epoch. However, as found by Yin et al. (2015, 2016), the harvest in warm epoch was better than that in cold epoch. This implicated that there should be different effects of regional extreme drought on grain harvest between warm and cold epochs.*

*The results showed that, during the warm epoch of 920–1300, the connection between the occurrence of poor harvest and regional extreme drought was not significantly close, though the frequency of poor harvest in extreme drought years was still slightly higher than that in non-extreme years for each sub-region. By contrast, during the cold epoch of 1310–1880, the frequency of poor harvest in extreme drought years was significantly higher than that in non-extreme years, which indicated that the connection between the occurrence of poor harvest and extreme drought was still significant. Moreover, similar characteristics were also found for the latter half of cold period in 1650–1880, which indicated that the shift of harvest grade distribution didn't affect the connection between poor harvest and extreme drought/flood in cold epoch. These results suggested that warm period could weaken the impact of extreme drought on poor harvest during historical times.*

**2. Drought and/or flood might have increased the likelihood that officials reported problems such as poor harvests and other disasters**

Accepted. As expressed in abovementioned revisions (P6, L3-7; Table S2), the records on harvests in historical documents was a relative level and focused directly on cropping in most cases, therefore

it is reasonable to suggest that there was no tendency in harvest records.

*In Chinese historical documents, the yearly harvest was usually recorded as a relative level compared to expected maximum yield rather than the individual crop production per hectare, although some records also reported the impacts on food availability, tax remissions, people's livelihoods, etc. Therefore these harvest records excluded the differences on absolute yield between sub-regions with different climate, soil fertility, crop variety, etc., and the difference between historical periods with changing agricultural centre, farming technology, staple crops (Su et al., 2014).*

**3. Harvest failures might have increased the likelihood that officials reported disasters such as droughts and/or floods.**

Accepted. As mentioned before, the droughts and floods records in historical documents were usually focused on abnormal precipitation, and appeared to report all extremes equally. (P3, L23-30)

*From these samples, it's easy to know that the droughts recorded in the historical documents were usually focused on the events due to no or less precipitation in a period at first, thus could be regarded as meteorological drought rather than hydrological drought and agricultural drought, although some records also reported the impacts on hydrosphere (e.g., dry up for river) or agriculture (e.g. wilting for crops). Therefore, the droughts recorded in the historical documents appear to report all droughts equally rather than those affected crops only. Similarly, the floods recorded in the historical documents could be regarded as more rains or heavy rains rather than the river that burst their banks or tsunamis, although some records also reported the impacts on the overflowing or bursting for lakes or river due to more precipitation or heavy rain.*

**4. Droughts and/or floods might have harmed human and animal health, reducing labor for harvests.**

Accepted and revised. We added these possible pathways for the connection between extreme events and poor harvest in the revised manuscript. (P10, L27-32)

*This is because that the relationship between poor harvest and extreme drought might be caused by not only the reductions in water supply, but also other possible indirect pathways. For example, the droughts might harm human and domestic animal health or cause migrations which reduced labor for farming. Moreover, the extreme events might reduce public revenue or increase public expenses which increase the political and economic instability, and further affect the agricultural activities*

*(Zheng et al., 2014a).*

**5. Droughts and/or floods might have damaged infrastructure and transportation, leading to food availability decline.**

Accepted. Most of the yearly harvest records were direct wording on assessment of crop yield, which could not be influenced by damaged grain transportation. (Table S2)

**6. Droughts and/or floods might have driven migrations, creating regional shortages both where agricultural labor emigrated and where people arrived seeking food.**

Accepted and revised. This possible pathway has been addressed in revised manuscript along with pathway 4. (P10, L27-32)
*This is because that the relationship between poor harvest and extreme drought might be caused by not only the reductions in water supply, but also other possible indirect pathways. For example, the droughts might harm human and domestic animal health or cause migrations which reduced labour for farming. Moreover, the extreme events might reduce public revenue or increase public expenses which increase the political and economic instability, and further affect the agricultural activities (Zheng et al., 2014a).*

**7. Periods of drought and/or flood might have reduced public revenue and/or increased public expenses, thus increasing the political and economic instability and decreasing food availability. (For instance, it's not clear how much the figures overall are influenced by the very high frequency of disasters and widespread famine during the political turbulence and violence accompanying the collapse of the Ming dynasty.) I am not arguing that any of these scenarios is necessarily the case. Nevertheless, each of these may be influencing the observed correlations.**

Accepted and revised. This possible pathway has also been addressed in revised manuscript along with pathway 4 and 6. (P10, L27-32)
*This is because that the relationship between poor harvest and extreme drought might be caused by not only the reductions in water supply, but also other possible indirect pathways. For example, the droughts might harm human and domestic animal health or cause migrations which reduced labour for farming. Moreover, the extreme events might reduce public revenue or increase public expenses which increase the political and economic instability, and further affect the agricultural activities (Zheng et al., 2014a)..*

**In summary, I do not believe that the authors' database and methods currently prove a valid correlation between flood and drought frequency and harvests in imperial China, nor that such a correlation would prove that drought or flood reduced harvest yields. The problem is not that the authors' hypothesis is unreasonable. It is simply that the conditions and data are too heterogeneous over such a large spatial and temporal scale. Any correlations found on such a scale are likely to have arisen from some artefact of the record-keeping or through the influence of some confounding variable, rather than to reflect a real and consistent climatic impact on agriculture.**

**Nevertheless, I would not like to dismiss this study out of hand. These datasets still have tremendous potential for historical climatology research. Better statistical methods could be devised to deal with changes in the frequency of historical reporting. By bringing trained historians onto such a project, the authors might find ways to handle problems related to historical changes in Chinese population, politics, land use, and economy. I would like to see the authors successfully address such problems in their research**

**Technical notes:**

**1). The paper variously sometimes to geographical parts of the country (e.g., "Northeast China") and sometimes to regional designations (e.g., "Jiangnan"). The paper would be clearer if it stuck with regional designations and names of provinces only.**

Accepted and revised. The related manuscript has been revised to make sure that same expression were used referring to each sub-regions in the study area (i.e. the North China Plain, the Jiang-Huai area, and the Jiang-Nan area). Although in certain sentences expressions such as "southern China" should still be used for accuracy when referring to specific orientation of China.

**2). The paper also needs extensive editing for English language grammar, spelling, and correct syntax. This is not merely a stylistic issue. The meaning of several passages is unclear due to lack of clear and correct English usage.**

Accepted and Revised. The revised manuscript is under processing by professional translation company for better English writing. Although it would take some time before it is accomplished. The polished manuscript could be available in about 10 days.

**Anonymous Referee #2**

**This is a nice piece of paper, well-structured with clear scoping and good delineation. It investigates the time evolution of extreme drought/flood events and the correlations of those extreme events with crop harvest in cold/warm epochs. In general, this paper provides very knowledgeable information on the drought/flood and harvest reconstruction method derived from documentary records, very comprehensive literature review in sections 1 and 2.1. However, there are still some points that I would suggest for authors to further improve the scientific quality and literacy of the paper.**

**Data used for analysis in this study is based on the previous analysis. The rationale for deciding the drought/flood (Zheng et al. 2006 and Hao et al. 2016) and harvest grades (Su et al, 2014; Yin et al, 2015) seemed promising however readers need to trace back those papers for more information on the source and profiles of data. There thus exists an ambiguity about how those records were collected for building the data sets, the characteristics and amounts of records, and data reliability evaluation. At this point, I can only assume that the data are reliable for the following analysis. It can be helpful if the authors provide some basic statistics (as appendix maybe, e.g. number of records per year, min, max etc for variables) of the data profile.**

Accepted and revised. Detailed statistics of the data profile had been given in revision (P3, L32-P4, L16; P6, L15-17). In addition, two tables on the methods for building the datasets and one figure on the statistics of data-missing status for extreme events are provided as supplementary materials (Table S1, S2; Fig. S1). These datasets have been commonly used in former studies and proved to be valid (P4, L22-26). The supplementary materials are attached at the end of this text.

*The grades were classified using the ideal frequency criteria of 10% (grade 1, severe drought), 20% (grade 2, drought), 40% (grade 3, normal), 20% (grade 4, flood), and 10% (grade 5, heavy flood) for whole area and all time, which were calibrated with the descriptions on duration, intensity, and area of the drought/flood events in wet season (usually May to September), and its impact (Table S1). Thus, the season for the drought/flood grade data overlaps with the critical agricultural activities and phases of crop growth. Although the data for the grade of drought/flood were unevenly distributed spatially for the whole 2000 year, i.e. there were few available data for grade of drought/flood in south China (south of 30$^o$N approximately) before CE 760 and even fewer data in south of the Huaihe River (approximately 34$^o$N) before CE 300 (Zhang, 1996), the coverage of this dataset has extended to south China since 760 CE and therefore covered the whole study area. There also existed missing data before 1470 as fewer historical documents survived from these earlier times (Zhang, 1996; Hao et al, 2016). The statistics shows that the mean percentage of available*

*data was 44.1% for 800-1469 and only 20% or even lower in periods around 850 and in 880s-920s, 1230s-1250s, 1360s and 1390s, in which the mean percentage of available data reporting "disasters or extremes" (i.e., grade 1, 2, 4, 5) and "normal" (i.e., grade 3) was 41.8% and 2.3%, respectively (Fig. S1). Moreover, there was no "normal" record in a period with length of 520 years. This means that the most of available grade data recorded disasters or extremes due to the principle of "recording the unusual rather than the normal" in the compilation on Chinese history. In consideration of the ideal frequency criteria in which 40% of all records were defined as "normal" and the other 60% were defined as "disasters and extremes", it could be implicated that about 70% of "disasters and extremes" that actually happened in that period was recorded (41.8% compared with 60% for whole area and all time).*

*The criteria and methods for grading the documentary records (i.e., grain yield descriptions and the related information) year by year were presented by Su et al. (2014) and summarized by Yin et al. (2015), in which the classification of the yearly harvest grade and descriptions recorded in historical documents was shown in Table S2.*

*Therefore, this dataset provides a valuable proxy and has been well used to study characteristics of precipitation changes over eastern China back to the past 2000 years. For example, by using this data set, Zheng et al. (2006) reconstructed a 1500 year regional dry/wet index series for the North China Plain (approximately 34–40˚N), the Jiang-Huai area (approximately 31–34˚N) and the Jiang-Nan area (approximately 25–31˚N).*

**Based on the data, methods used for analysis is relatively simple. The authors used 50-years moving average (they term it as moving window) to smoothen extreme drought/flood trend, used Wilcoxon rank test to examine/compare median values of every intervals, and used contingency table to examine the effects between extreme drought/flood and harvest and between them and cold/warm periods. For this section, I suggest the authors to add a short paragraph giving readers some concepts about the method structure before going into details.**

Accepted and revised. A new paragraph has been added to introduce the method structure in this section. (P7, L7-11)

*Four kinds of data processing methods were used in this study, including the moving average method, the Wilcoxon rank sum test, the two-sampled t-test and the contingency table. The first two methods were used to explore the characteristics of occurrences of regional extreme drought/flood and the grain harvest grade from 801 to 1910, while the latter two methods were used to examine the relationship between poor harvest and extreme drought/flood for the whole study period or for specific warm/cold periods.*

**Also there are some unclear parts:**

**1).what do you mean by saying 'moving-window of 50 years and step of 10 years (line 24-25, page 5, and figure 3 caption, page 18)'? Please provide explanations.**

Accepted and revised. (P7, L14-15)

*For example, the smoothed series is made up of means of 801-850, 811-860, 821-870, etc.*

**2). For the same figure 3 caption, please use real number to replace full confidence, high confidence, medium or low confidence. It is unclear what those mean.**

Accepted and revised. (P20, L6-9)

*Bars at bottom row of each plate illustrated the confidence levels (probability in being correct, PBC): very high confidence (PBC>90%): dark; high confidence (PBC: 66.7%~90%): 50% shaded dark; medium confidence (PBC: 50%~66.7%): 25% shaded dark; low confidence: 12.5% shaded dark (PBC: 33.3%~50%); and very low confidence: blank (PBC: <33.3%) for each reconstructions at per 50 years.*

**3). Also, I don't quite understand the sentence in line 26-28 page 5 "This is because the mean of rank series in an interval was equivalent to the frequency of ……by labeling the extreme drought/flood years as 1 and non-extreme years as 0". Please provide more explanations.**

Accepted and revised. (P7, L16-19)

*By labelling the extreme drought (or flood) years as 1 and non-extreme years as 0, the chronology of extreme drought and flood years would be transformed into a rank series, and the mean of this rank series is equivalent to the frequency of drought (or flood) years. Therefore the intervals with significant more or less drought (or flood) years could be recognized through Wilcoxon rank sum test performed on the rank series.*

**The research results are clear and straightforward. The drought, flood and harvest trends and their descriptions are clear. However some trends are inconsistent with previous studies. For example, the authors mentioned that "there was an evident jump around 1640s with increase of years of (harvest) grade 4......" (line 12-13 page 7). I admire the following sentences on the discussions of the records in Qing and previous dynasties to clarify the discrepancies of the historical books. However, after removing grade 4 (average harvest), there still existed an**

**obvious jump of grade 5&6 (bumper) after 1640 which was commonly recognized in previous literatures as having poor harvest and famine in this coldest interval of little ice age. There might be some reasons including suddenly increasing number of local chronicles in Qing dynasty that could dilute the drought magnitude based on your grading method or the 50-years moving average can further smoothen the trend. In a word, it will be extremely valuable if the authors can compare the present analysis with previous studies and provide explanations or new perspectives.**

Accepted and revised. Detailed discussion about the abrupt change in frequency of grade 5+6 in harvest has been applied. The jump around 1650s might be resulted from the uncertainty of source data, we added it in the discussion (P13, L15-18). However, the harvest in the whole Qing Dynasty was significantly high indeed, which could also be confirmed by other datasets from independent sources (P11, L29-P12, L1).

*In addition, there existed an obvious jump of grade 5+6 (bumper harvest) around 1650s -1660s, yet this period was commonly recognized in previous literatures as having poor harvest and famine in this coldest interval of little ice age. Also, in the periods of 1130~1150, 1210-1270 (i.e., the early and later Southern Song Dynasty), 880-980 (i.e., the later Tang Dynasty and Five Kingdom) and around 1400, there were remarkably more data missing on harvest grade data.*

*Meanwhile, there existed an evidently high frequency of bumper harvest (grade 5+6) from 1650s to 1810s. This jump could also be confirmed by other datasets from independent sources. For example, according to the harvest reports in Archives of Qing Dynasty, the mean harvest percentage over eastern China for 1730-1820 was even over 70% (i.e. near bumper) (Ge and Wang, 1995), in which it was over 75% in Guangdong Province, southern China for 1707-1800 (Marks, 1998).*

*Ge, Q., and Wang, W.-C.: Population Pressure, climate change and Taiping Rebellion, Geogr. Res., 14(4), 32–42, 1995. (in Chinese)*

**Overall, this paper provides new and important insights into the correlations among extreme event, harvest and cold/warm climate. Data statistic is suggested to provide, and since missing data especially for harvest is prominent (35%), it is suggested that authors are more careful to claim their conclusions. Some inconsistency is also found between text and tables, e.g. 49.4% (line 23) and 24.0% (line 24) on page 8 are different from those shown in table 4. Further English editing is strongly suggested to improve high quality writing style of this nice paper.**

Accepted and revised. The inconsistency between text and tables has been revised (P10, L15-16)

and the revised manuscript has been handed over to professional translation company for better English writing. Although it would take some time before it is accomplished. The polished manuscript could be available in about 10 days.

**Table S1: Criteria for calibrating the drought/flood grade and descriptions of drought/flood disasters recorded in historical documents (Zhang, 1996)**

| Grade | Frequency distribution (%) | Descriptions with meaning in historical documents |
|---|---|---|
| 1 | 10 | Continuous drought lasting two or more months in wet season (usually May to September) or crossing two seasons with severe intensity and impact over a broad area, such as "villages for hundreds of miles were abandoned" |
| 2 | 20 | Drought lasting two months or more than one month in wet season with visible impacts |
| 3 | 40 | Usual case (such as "rain blended well in seasons") or nothing special to be recorded |
| 4 | 20 | More rains lasting less than two months or more heavy rains in wet season with evident impact |
| 5 | 10 | Continuous more rains lasting two or more months, or extraordinary heavy rains in wet season with severe impacts, such as "driving boats over land" |

**Table S2: The classification of yearly harvests level derived from the wording recorded in historical documents (Su et al., 2014; Yin et al., 2015)**

| Levels of yearly harvests | 1 | 2 | 3 | 4 | 5 | 6 |
|---|---|---|---|---|---|---|
| | Very poor | Poor | Slightly poor | Average | Near bumper | Bumper |
| Harvest rating in percentage | <40% | 40%-50% | 50%-60% | 60-70% | 70-80% | >80% |
| Direct wording on assessment of crop yield | Very poor harvest; Very bad year; Nothing was reaped; No harvest in vast areas | Poor harvest; Poor grain harvest; Bad year; No prospect for harvest; No any bumper crops | Slightly poor harvest; No enough crop yield; No good harvest/year | Normal harvest; Unusual year; Not bad year; Common year | Bumper harvest; Good harvest; Bumper year; Favorable year | Large bumper harvest; Large harvest; A golden year; Bumper harvest of all "five" crops |
| Impact wording on assessment of food security related to crop harvest directly | People became cannibal; Large famine and no food; Deadly famine year and beggars were everywhere, many people died of hunger | People were starving; No enough food for people; People had pale and anemic complexions; Production on alcohol and related drink were prohibited | | | Surplus grain stored in the fields; Foods and clothes more than enough | |
| Related features, e.g., tax remission, people's livelihoods, grain prices and grain storage status | All taxes were remitted; People fled everywhere; Pirates and beggars were rampant; Food storage was exhausted | Taxes were partly remitted; People were not engaged in agriculture; People lived in poverty; People could not support themselves; Most of the people were poor; People were impoverished and had many complaints; The granaries were nearly empty; The whole country was poor and weak; The grain price was five or ten thousand Qian per Dan; The grain price was soaring; Grain became more and more expensive | Farms were laid waste; People had no surplus food storage | | People lived a prosperous and contented life; There was enough grain for store | Every family had adequate supplies of food and clothes; People were well-off; The whole country was in safety and good order; The granaries were all full; The grain price was so low that peasants could not earn enough money |

[Figure]

**Figure S1: The percentage of available data in the reconstructed 63-stations annual drought/flood grades from 801 to 1470 CE. Pink line indicates the yearly variation for all grades in total and red bold line shows its 10-yr moving average. Black bars indicates the yearly variation for "normal" conditions (i.e. grade 3).**

---

## Author Comment (AC2) · 14 Jun 2019

**Correction to the Author's Comments on "Extreme droughts/floods and their impacts on harvest derived from historical documents in Eastern China during 801–1910" by Zhixin Hao et al**

Dear editors and reviewers,

A correction to the caption of figure 3 has been made. Please refer to the revised caption since it is related to question 2 of the "unclear parts" given by anonymous referee #2. Please feel free to contact me if you have any further questions.

Many thanks again. With best wishes.

Sincerely yours,

Jingyun Zheng

**Anonymous Referee #2**

**2). For the same figure 3 caption, please use real number to replace full confidence, high confidence, medium or low confidence. It is unclear what those mean.**

Accepted and revised. There was an error in the description of this caption in the previous author's comments. Corrected caption is listed below (P20, L6-9). Same correction has been made in the manuscript (P5, L28-31).

*Bars at bottom row of each plate illustrated the confidence levels (probability in being correct, PBC) for each reconstructions at per 50 years: extremely high confidence (PBC>99%): dark; very high confidence (PBC>90%): 50% shaded dark; high confidence (PBC>80%): 25% shaded dark; medium confidence (PBC>50%): 12.5% shaded dark; low confidence (PBC>33.3%): blank.*

*In addition, to illustrate the uncertainty of reconstructions for regional extreme drought/flood, the confidence levels of them were also assessed based on the percentage of years with data available at per 50 years interval, in which the extremely high confidence is defined as the percentage more than 99%; very high confidence: >90%; high confidence: >80%; medium confidence: >50%; and low confidence: >33.3%.*

---

## Author Response (AR1)

**Reply to the referee comments on "Extreme droughts/floods and their impacts on harvest derived from historical documents in Eastern China during 801–1910" by Zhixin Hao et al**

Dear editors and reviewers,

Thank you for your valuable comments and thoughtful suggestions on our manuscript. Following your comments on the manuscript, we made careful revisions, and the point-to-point response of the comments is listed below. We hope these revisions would make this manuscript more acceptable for publication. Please feel free to contact me if you have any questions.

Many thanks again. With best wishes.

Sincerely yours,

Jingyun Zheng

**Anonymous Referee #1**

**Using databases of Song, Ming, and Qing documents, this paper finds that the frequency of reports of extreme droughts (but not always floods) correlates with reductions in harvests as reported in historical sources. On this basis, the authors conclude that there are clear historical periods when droughts reduced harvests, and therefore that these events had significant societal impacts. The sources and methods used in this manuscript appear to be standard in other publications on Chinese historical climatology. In this instance, however, I am not convinced they are adequate to prove the authors' conclusion. The problems concern, first, the author's use of their historical databases; second, the large temporal and spatial scale of the study; and third, the interpretation given to the pattern of correlations found.**

**Problems in use of databases:**

**The authors' use of databases of flood and drought events and harvest grades raises numerous questions which must be answered before it is clear whether or not the correlations identified are valid:**

**1) What kinds of droughts are recorded in the historical sources: meteorological drought? hydrological drought? agricultural drought? or some combination of these? Were observers more likely to report precisely those droughts that affected crops, or did they report all droughts equally?**

Accepted and revised. Most of droughts recorded in historical documents were events due to no or less precipitation, and could be regarded as meteorological droughts. Therefore it's reasonable to suggest that they reported all droughts equally rather than those affected crops. (P3, L23-27)

*As this sample suggests, historical documents were usually focused on the events due to no or less precipitation than usual and, thus could be regarded as meteorological drought rather than hydrological or agricultural droughts, although some records also report impacts on the hydrosphere (e.g., rivers drying up for river) or on agriculture (e.g. wilting for crops). Historical documents, therefore, appear to report all droughts equally, rather than only those affecting crops.*

**2) What kinds of floods are recorded in the historical sources: heavy rains? tsunamis? rivers that burst their banks? Does the database control for ongoing problems related to river hydrology? How do major events such as course changes in the Yellow River figure into the measure of flood frequency: as one flood? as many?**

Accepted and revised. Similar to the records on drought, flood recorded in historical documents were mostly about more rains or heavy rains. River hydrology events were not taken into account unless it was resulted from more precipitation or heavy rain. (P3, L27-29)

*Similarly, floods recorded in the historical documents could be regarded as more rain or heavier rain than usual, rather than in the context of rivers bursting their banks or tsunamis, although some records also report the impacts of overflowing or bursting lakes and rivers due to more or heavier precipitation.*

**3) What is the seasonality of the meteorological events recorded in the historical sources? Does the seasonality of floods or droughts necessarily overlap with the seasonality of critical agricultural activities or phases of crop growth?**

Accepted and revised. The 63-stations annual drought/flood grades was reconstructed and calibrated with descriptions on duration, intensity, and area of the drought/flood events in wet season, which was usually May to September in the study area and overlaps with the critical agricultural activities and phases of crop growth. (P3, L31-P4, L3)

*Grades were classified using ideal frequency criteria of 10% (grade 1, severe drought), 20% (grade 2, drought), 40% (grade 3, normal), 20% (grade 4, flood), and 10% (grade 5, heavy flood) for the whole area and all time. These grades were calibrated based on descriptions of duration, intensity, and area of the drought/flood event during the wet season (usually May to September), and its*

*impact (Table S1). Thus, the season of the drought/flood grade data overlaps with critical agricultural activities and phases of crop growth.*

**4) What is being measured by "harvest"? Yield per seed? Total yield per hectare? Food availability?**

Accepted and revised. Harvest in records was a relative concept and represented the ratio of actual yield compared to the possible maximum yield. (P6, L1-3)

*In Chinese historical documents, the yearly harvest was usually recorded as a relative level compared to an expected maximum yield, rather than crop yield per hectare, although some records also report impacts of harvest fluctuation on food availability, tax remissions, livelihoods, and so on.*

**5) Are degrees of flood, drought, and harvest based entirely on narrative descriptions, or are there objective phenological or quantitative measures to help define them?**

Accepted and revised. The degrees of flood, drought, and harvest was not based entirely on narrative descriptions. The duration, intensity, and area of these events and their impacts were adopted in calibration, as well. We added two supplementary tables (Table S1, S2) to explain the detailed criteria in grading drought/flood (P3, L30-P4, L2; Table S1) and harvest (P6, L12-15; Table S2). Please refer to the supplementary material which is uploaded additionally as an independent file.

*Based on these records, Zhang (1996) reconstructed a dataset of annual drought/flood grades at 63 stations from 137 BCE. Each station consisted of a local area of approximately 20 counties with the same climate. Grades were classified using ideal frequency criteria of 10% (grade 1, severe drought), 20% (grade 2, drought), 40% (grade 3, normal), 20% (grade 4, flood), and 10% (grade 5, heavy flood) for the whole area and all time. These grades were calibrated based on descriptions of duration, intensity, and area of the drought/flood event during the wet season (usually May to September), and its impact (Table S1).*

*In the dataset, yearly harvest levels were classified into 6 grades: 1-Very poor, 2-Poor, 3-Slightly poor, 4-Average, 5-Near bumper, 6-Bumper. The criteria and methods for year-by-year grading of the documentary records (i.e., grain yield descriptions and related information) were presented by Su et al. (2014) and summarized by Yin et al. (2015). The classification of the yearly harvest grade and descriptions recorded in historical documents is shown in Table S2.*

**Regarding the temporal and spatial scale of the study, I am concerned that it relies on improbable assumptions of continuity and homogeneity in Chinese population, land use, and record keeping. In order to accept as valid any long-term correlations between reported drought or flood frequency and "Chinese" or even "regional" "harvests" I would need the authors to address the following issues:**

**1) How do the data control for the changing borders of Chinese empires? A priori, I would expect vastly different vulnerabilities and patterns of reporting between the Northern Song and Southern Song periods, simply based on the major geographical shifts in population and wealth between those two dynasties.**

Accepted and revised. For droughts and floods, the historical records was transformed into graded data based on 63-stations, each of which was set as a local area consisted of about 20 counties and does not change in different dynasties (P3, L30-31). Although the available graded data was unevenly distributed spatially for different dynasties, it had been proved in the paper of Hao et al. (2010a) that the extreme drought/flood years recognized were mostly robust despite of the percentage of data-missing stations (P5, L5-11). As for harvest, the impact of changing borders on harvest grade should also be limited since the main grain product area had been relatively stable in the study period and the records in documents was about relative harvest rather than absolute yield as suggested by Yin et al. (2015) (P6, L28-31).

*Based on these records, Zhang (1996) reconstructed a dataset of annual drought/flood grades at 63 stations from 137 BCE. Each station consisted of a local area of approximately 20 counties with the same climate.*

*To verify the rationality of this method and criteria, validation was conducted in Hao et al. (2010a), based on 10 extreme events identified from a series of precipitation observations in each sub-region according to a threshold of probabilities of 10% and 90% occurrence. In this validation, all or part of grade 3 stations were deliberately omitted, and only 40% or 60% of stations with disaster or extreme grade were reserved without changing the drought-to-flood ratio within the available data. The results show that, with one exception, years of extreme drought and extreme flood, identified according to this method and criteria, closely matched those extreme events identified by precipitation data, demonstrating that the method and criteria were reasonable.*

*However, such social factors should have only limited influence on yearly harvest grade dataset, since the harvest in the documents was reported as a relative level rather than the absolute yield, also the main grain product area, the staple crop, and the cropping system have been relatively stable throughout the study period (Yin et al., 2015).*

**2) How do data on "harvests" control for changes in staple crops, introduction of New World crops including peanuts and sweet potatoes, changing cropping patterns, and the increasing commercial orientation of agriculture?**

Accepted and revised. The records for harvests were usually about relative percentage compared to expected maximum yield rather than absolute yield, and thus it should not be influenced by these factors. (P6, L1-5)

*In Chinese historical documents, the yearly harvest was usually recorded as a relative level compared to an expected maximum yield, rather than crop yield per hectare, although some records also report impacts of harvest fluctuation on food availability, tax remissions, livelihoods, and so on. Therefore these harvest records exclude differences in absolute yield between sub-regions with different climates, soil fertility and types, crop varieties, etc., as well as difference between historical periods with changing agricultural centres, farming technologies, staple crops, and so on. (Su et al., 2014).*

**3) How do the data deal with the changing vulnerabilities to climate variability based on changing settlement patterns even within regions (e.g., uplands in the south and southwest colonized by Han settlers during the late Ming and Qing periods)?**

Accepted and revised. The graded harvest data represents a nationwide status and since the main grain product area had been relatively stable in the study period, the expansion of agricultural area should have limited influence on the yearly harvest documents. (P6, L25-31)

*During this study period, several social factors existed which could have influenced China's total yield, such as changing borders of empires, the expansion of agricultural area (e.g., uplands in the south and southwest colonized by Han settlers during the late Ming and Qing periods), the updated crop varieties introduced from the New World (e.g., peanuts and sweet potatoes), advanced agricultural management technology, and so on. However, such social factors should have only limited influence on yearly harvest grade dataset, since the harvest in the documents was reported as a relative level rather than the absolute yield, also the main grain product area, the staple crop, and the cropping system have been relatively stable throughout the study period (Yin et al., 2015).*

**4) Given the very long time period examined here, wouldn't we expect new adaptations to reduce vulnerabilities to predictable climate variability and disasters?**

Accepted and revised. This question has also been addressed in the revisions responding to above questions. (P6, L1-5, L25-31).

*In Chinese historical documents, the yearly harvest was usually recorded as a relative level compared to an expected maximum yield, rather than crop yield per hectare, although some records also report impacts of harvest fluctuation on food availability, tax remissions, livelihoods, and so on. Therefore these harvest records exclude differences in absolute yield between sub-regions with different climates, soil fertility and types, crop varieties, etc., as well as difference between historical periods with changing agricultural centres, farming technologies, staple crops, and so on. (Su et al., 2014).*

*During this study period, several social factors existed which could have influenced China's total yield, such as changing borders of empires, the expansion of agricultural area (e.g., uplands in the south and southwest colonized by Han settlers during the late Ming and Qing periods), the updated crop varieties introduced from the New World (e.g., peanuts and sweet potatoes), advanced agricultural management technology, and so on. However, such social factors should have only limited influence on yearly harvest grade dataset, since the harvest in the documents was reported as a relative level rather than the absolute yield, also the main grain product area, the staple crop, and the cropping system have been relatively stable throughout the study period (Yin et al., 2015).*

**5) Most importantly, how can we make up for the fact there are simply more records from the Qing period than earlier periods? I don't see that the methods used in this manuscript avoid the problem that more records will create a misimpression of a greater frequency of floods and droughts. The authors propose to ignore reports of "average" conditions in Qing records to make them more comparable to Song and Ming records. However, that would only work if the Song and Ming records still reliably reported all disasters and extremes and only left out "average" conditions. I don't see any reason to make that assumption. Perhaps the authors could experiment with methods of introducing "noise" into the data in order to reflect the events missing from the reports. Or else they could employ a Bayesian method to indicate that the presence or absence of certain descriptions in the records may be used to obtain updated posterior probabilities of actual conditions, without ever assuming that the records provide a complete account of events. In any case, the authors must come up with a way to handle these changes in the documentary record over time if they are to make a convincing case for stable long-term correlations between floods and droughts and harvests.**

Accepted and revised. The method of ignoring "average" conditions is based on the hypothesis that the records on droughts and floods were omitted randomly and unbiased, which suggests that the relative drought-to-flood ratio in the available data would be close to that in actual history. As in the abovementioned revisions, the recognition for extreme drought and flood years was still effective even if 40% or 60% of the available data with disasters was omitted deliberately. (P4, L34-P5, L13)

*Extreme drought or extreme flood years were defined in this way, as the probabilities for omitting drought and flood records were random and unbiased, despite the greater frequency of missing data in the older records. In other words, if one period had a large number of documents, it was expected to be rich in both drought and flood records, and vice versa. Therefore, the amount of missing data should not have a significant effect on the relative drought-to-flood ratio within the available data. To verify the rationality of this method and criteria, validation was conducted in Hao et al. (2010a), based on 10 extreme events identified from a series of precipitation observations in each sub-region according to a threshold of probabilities of 10% and 90% occurrence. In this validation, all or part of grade 3 stations were deliberately omitted, and only 40% or 60% of stations with disaster or extreme grade were reserved without changing the drought-to-flood ratio within the available data. The results show that, with one exception, years of extreme drought and extreme flood, identified according to this method and criteria, closely matched those extreme events identified by precipitation data, demonstrating that the method and criteria were reasonable. The reason for the close match is that precipitation variability in eastern China is dominated by the East Asian Summer Monsoon (EASM). Therefore, when extreme drought or flood events occur, the precipitation variation for stations within each sub-region usually share similar relative magnitudes.*

**Third, even if the correlations found in the study are valid, there is a problem with the authors' historical interpretation of them. The correlations discovered here are not between climate and harvests, but rather reports of floods and droughts and reported harvests. The authors assume that the correlations mean that floods and droughts reduced harvests. However, there are a number of potentially confounding variables, which indicate other potential pathways of causality and therefore other historical possibilities:**
**1. Drought and/or flood might have correlated with other climate variables (such as temperature) that caused harvest failure.**

Accepted. As elaborated in previous study, the relationship between temperature and harvest had been investigated by Yin et al. (2015, 2016), which suggested that there would be better harvest in warm climate. And our study, in section 3.2 of the original manuscript, found that more occurrence

of extreme drought in eastern China could lead to significant increase of frequency of poor harvest (grade 1+2) compared with non-extreme years. To further examine whether the drought and/or flood are correlated with temperature change, and if so, how the drought and/or flood are correlated with harvest failure under different temperature backgrounds, we presented a study in section 3.3, and found that there were slightly more extreme droughts in the warm period. However, the connection between extreme droughts and poor harvest was not significantly close in the warm epoch, while it was more significant in the cold epoch. These results suggested that warm period could weaken the impact of extreme drought on poor harvest during historical times. (P11, L18-22; P11, L28-P12, L2)

*As found in section 3.2, more occurrence of extreme drought in eastern China led to a significant increase in the frequency of poor harvests (grade 1+2) when compared with non-extreme years. Since more extreme droughts occurred over eastern China in 920–1300 than in 1310–1880, the harvest in the warm epoch could be expected to be worse than in the cold epoch. However, as Yin et al. (2015, 2016) found, the harvest in warm epoch was better than that in cold epoch. This suggests that the effects of regional extreme drought on the grain harvest differed between warm and cold epochs.*

*The results show that, during the warm epoch of 920–1300, there was no significant connection between the occurrence of poor harvest and regional extreme drought, although the frequency of poor harvest in extreme drought years was slightly higher than in non-extreme years for each sub-region. In contrast, during the cold epoch of 1310–1880, the frequency of poor harvest in extreme drought years was significantly higher than in non-extreme years, which indicates that the connection between the occurrence of poor harvest and extreme drought was still significant. Moreover, similar characteristics were found for the latter half of the cold period from 1650 to 1880, which indicates that the shift of harvest grade distribution did not affect the connection between poor harvest and extreme drought/flood during the cold epoch. These results suggest that the warm period could weaken the impact of extreme drought on poor harvests in historical times.*

**2. Drought and/or flood might have increased the likelihood that officials reported problems such as poor harvests and other disasters**

Accepted. As expressed in abovementioned revisions (P6, L1-5; Table S2), the records on harvests in historical documents was a relative level and focused directly on cropping in most cases, therefore it is reasonable to suggest that there was no tendency in harvest records.

*In Chinese historical documents, the yearly harvest was usually recorded as a relative level compared to an expected maximum yield, rather than crop yield per hectare, although some records*

*also report impacts of harvest fluctuation on food availability, tax remissions, livelihoods, and so on. Therefore these harvest records exclude differences in absolute yield between sub-regions with different climates, soil fertility and types, crop varieties, etc., as well as difference between historical periods with changing agricultural centres, farming technologies, staple crops, and so on. (Su et al., 2014).*

**3. Harvest failures might have increased the likelihood that officials reported disasters such as droughts and/or floods.**

Accepted. As mentioned before, the droughts and floods records in historical documents were usually focused on abnormal precipitation, and appeared to report all extremes equally. (P3, L23-29)

*As this sample suggests, historical documents were usually focused on the events due to no or less precipitation than usual and, thus could be regarded as meteorological drought rather than hydrological or agricultural droughts, although some records also report impacts on the hydrosphere (e.g., rivers drying up for river) or on agriculture (e.g. wilting for crops). Historical documents, therefore, appear to report all droughts equally, rather than only those affecting crops. Similarly, floods recorded in the historical documents could be regarded as more rain or heavier rain than usual, rather than in the context of rivers bursting their banks or tsunamis, although some records also report the impacts of overflowing or bursting lakes and rivers due to more or heavier precipitation.*

**4. Droughts and/or floods might have harmed human and animal health, reducing labor for harvests.**

Accepted and revised. We added these possible pathways for the connection between extreme events and poor harvest in the revised manuscript. (P10, L25-28)

*This relationship may have been caused by both reductions in water supply and other indirect pathways. For example, droughts might harm human and domestic animal health or result in migration, leading to a reduced agricultural labour force. In addition, extreme events might reduce public revenue or increase public expenses, thus increasing political and economic instability, and further affecting agricultural activities (Zheng et al., 2014a).*

**5. Droughts and/or floods might have damaged infrastructure and transportation, leading to food availability decline.**

Accepted. Most of the yearly harvest records were direct wording on assessment of crop yield, which could not be influenced by damaged grain transportation. (Table S2)

**6. Droughts and/or floods might have driven migrations, creating regional shortages both where agricultural labor emigrated and where people arrived seeking food.**

Accepted and revised. This possible pathway has been addressed in revised manuscript along with pathway 4. (P10, L25-28)

*This relationship may have been caused by both reductions in water supply and other indirect pathways. For example, droughts might harm human and domestic animal health or result in migration, leading to a reduced agricultural labour force. In addition, extreme events might reduce public revenue or increase public expenses, thus increasing political and economic instability, and further affecting agricultural activities (Zheng et al., 2014a).*

**7. Periods of drought and/or flood might have reduced public revenue and/or increased public expenses, thus increasing the political and economic instability and decreasing food availability. (For instance, it's not clear how much the figures overall are influenced by the very high frequency of disasters and widespread famine during the political turbulence and violence accompanying the collapse of the Ming dynasty.) I am not arguing that any of these scenarios is necessarily the case. Nevertheless, each of these may be influencing the observed correlations.**

Accepted and revised. This possible pathway has also been addressed in revised manuscript along with pathway 4 and 6. (P10, L25-28)

*This relationship may have been caused by both reductions in water supply and other indirect pathways. For example, droughts might harm human and domestic animal health or result in migration, leading to a reduced agricultural labour force. In addition, extreme events might reduce public revenue or increase public expenses, thus increasing political and economic instability, and further affecting agricultural activities (Zheng et al., 2014a).*

**In summary, I do not believe that the authors' database and methods currently prove a valid**

**correlation between flood and drought frequency and harvests in imperial China, nor that such a correlation would prove that drought or flood reduced harvest yields. The problem is not that the authors' hypothesis is unreasonable. It is simply that the conditions and data are too heterogeneous over such a large spatial and temporal scale. Any correlations found on such a scale are likely to have arisen from some artefact of the record-keeping or through the influence of some confounding variable, rather than to reflect a real and consistent climatic impact on agriculture.**

**Nevertheless, I would not like to dismiss this study out of hand. These datasets still have tremendous potential for historical climatology research. Better statistical methods could be devised to deal with changes in the frequency of historical reporting. By bringing trained historians onto such a project, the authors might find ways to handle problems related to historical changes in Chinese population, politics, land use, and economy. I would like to see the authors successfully address such problems in their research**

**Technical notes:**

**1). The paper variously sometimes to geographical parts of the country (e.g., "Northeast China") and sometimes to regional designations (e.g., "Jiangnan"). The paper would be clearer if it stuck with regional designations and names of provinces only.**

Accepted and revised. The related manuscript has been revised to make sure that same expression were used referring to each sub-regions in the study area (i.e. the North China Plain, the Jiang-Huai area, and the Jiang-Nan area). Although in certain sentences expressions such as "southern China" should still be used for accuracy when referring to specific orientation of China.

**2). The paper also needs extensive editing for English language grammar, spelling, and correct syntax. This is not merely a stylistic issue. The meaning of several passages is unclear due to lack of clear and correct English usage.**

Accepted and Revised. The revised manuscript has been edited for proper English language by LetPub. Certificate of English language editing provided by LetPub is attached below in this response file.

**Anonymous Referee #2**

**This is a nice piece of paper, well-structured with clear scoping and good delineation. It investigates the time evolution of extreme drought/flood events and the correlations of those extreme events with crop harvest in cold/warm epochs. In general, this paper provides very knowledgeable information on the drought/flood and harvest reconstruction method derived from documentary records, very comprehensive literature review in sections 1 and 2.1. However, there are still some points that I would suggest for authors to further improve the scientific quality and literacy of the paper.**

**Data used for analysis in this study is based on the previous analysis. The rationale for deciding the drought/flood (Zheng et al. 2006 and Hao et al. 2016) and harvest grades (Su et al, 2014; Yin et al, 2015) seemed promising however readers need to trace back those papers for more information on the source and profiles of data. There thus exists an ambiguity about how those records were collected for building the data sets, the characteristics and amounts of records, and data reliability evaluation. At this point, I can only assume that the data are reliable for the following analysis. It can be helpful if the authors provide some basic statistics (as appendix maybe, e.g. number of records per year, min, max etc for variables) of the data profile.**

Accepted and revised. Detailed statistics of the data profile had been given in revision (P3, L31-P4, L16; P6, L13-15). In addition, two tables on the methods for building the datasets and one figure on the statistics of data-missing status for extreme events are provided as supplementary materials (Table S1, S2; Fig. S1). These datasets have been commonly used in former studies and proved to be valid (P4, L22-25). Please refer to the supplementary material which is uploaded additionally as an independent file.

*Grades were classified using ideal frequency criteria of 10% (grade 1, severe drought), 20% (grade 2, drought), 40% (grade 3, normal), 20% (grade 4, flood), and 10% (grade 5, heavy flood) for the whole area and all time. These grades were calibrated based on descriptions of duration, intensity, and area of the drought/flood event during the wet season (usually May to September), and its impact (Table S1). Thus, the season of the drought/flood grade data overlaps with critical agricultural activities and phases of crop growth. The drought/flood grade data are unevenly spatially distributed across the 2000-year period. For example, drought/flood grade data for south China (south of 30 ̊N approximately) were limited for the period before CE 760, and there were even fewer data for south of the Huaihe River (approximately 34 ̊N) before CE 300 (Zhang, 1996). However, the coverage of this dataset has extended to south China since 760 CE and, therefore, covered the whole study area. There also existed missing data before 1470, as fewer historical*

*documents have survived from these earlier times (Zhang, 1996; Hao et al, 2016). Statistics show that the mean percentage of available data was 44.1% for 800–1469 and only 20% or lower for periods around 850 and for the 880s–920s, 1230s–1250s, 1360s and 1390s. During the period of 800–1469, the mean percentage of available data reporting "disasters or extremes" (i.e., grade 1, 2, 4, 5) was 41.8% and reporting "normal" (i.e., grade 3) was 2.3% (Fig. S1). Moreover, there was a period of 520 years when no "normal" record existed. This means that most of the available grade data recorded disasters and extremes following the principle of "recording the unusual rather than the normal" in the compilation of Chinese history. In consideration of ideal frequency criteria, in which 40% of all records were defined as "normal" and 60% defined as "disasters and extremes," it could be implied that approximately 70% of the "disasters and extremes" that actually happened in that period were recorded (41.8% in records compared with 60% in the ideal frequency criteria for the whole area and all time).*

*The criteria and methods for year-by-year grading of the documentary records (i.e., grain yield descriptions and related information) were presented by Su et al. (2014) and summarized by Yin et al. (2015). The classification of the yearly harvest grade and descriptions recorded in historical documents is shown in Table S2.*

*Therefore, this dataset provides a valuable proxy and has already been used to study characteristics of precipitation change in eastern China over the past 2000 years. For example, Zheng et al. (2006) used this dataset to reconstruct a 1500 year regional dry/wet index series for the North China Plain (approximately 34–40 °N), the Jiang-Huai area (approximately 31–34 °N) and the Jiang-Nan area (approximately 25–31 °N).*

**Based on the data, methods used for analysis is relatively simple. The authors used 50-years moving average (they term it as moving window) to smoothen extreme drought/flood trend, used Wilcoxon rank test to examine/compare median values of every intervals, and used contingency table to examine the effects between extreme drought/flood and harvest and between them and cold/warm periods. For this section, I suggest the authors to add a short paragraph giving readers some concepts about the method structure before going into details.**

Accepted and revised. A short paragraph has been added to introduce the methods used in this study in section 2.2. (P7, L4-5)

*Four kinds of data processing method were used in this study, including the moving average, the Wilcoxon rank sum test, the two-sampled t-test and the contingency table with the Chi-square test (χ2).*

**Also there are some unclear parts:**

**1).what do you mean by saying 'moving-window of 50 years and step of 10 years (line 24-25, page 5, and figure 3 caption, page 18)'? Please provide explanations.**

Accepted and revised. (P7, L8-9)

*For example, a smoothed series was made up of means of 801–850, 811–860, 821–970, and so on.*

**2). For the same figure 3 caption, please use real number to replace full confidence, high confidence, medium or low confidence. It is unclear what those mean.**

Accepted and revised. (P21, L6-9)

*The bars on the bottom row of each plate illustrate confidence levels (probability in being correct, PBC) for each reconstructions at 50 years intervals: extremely high confidence (PBC>99%): dark; very high confidence (PBC>90%): 50% shaded dark; high confidence (PBC>80%): 25% shaded dark; medium confidence (PBC>50%): 12.5% shaded dark; and low confidence (PBC>33.3%): blank.*

**3). Also, I don't quite understand the sentence in line 26-28 page 5 "This is because the mean of rank series in an interval was equivalent to the frequency of ……by labeling the extreme drought/flood years as 1 and non-extreme years as 0". Please provide more explanations.**

Accepted and revised. (P7, L10-13)

*By labelling extreme drought (or flood) years as 1 and non-extreme years as 0, the chronology of extreme drought and flood years could be transformed into a rank series, with the mean of this rank series equivalent to the frequency of drought (or flood) years. Therefore, those intervals with significantly more or fewer drought (or flood) years could be recognized through a Wilcoxon rank sum test performed on the rank series.*

**The research results are clear and straightforward. The drought, flood and harvest trends and their descriptions are clear. However some trends are inconsistent with previous studies. For example, the authors mentioned that "there was an evident jump around 1640s with increase of years of (harvest) grade 4......" (line 12-13 page 7). I admire the following sentences on the discussions of the records in Qing and previous dynasties to clarify the discrepancies of the**

**historical books. However, after removing grade 4 (average harvest), there still existed an obvious jump of grade 5&6 (bumper) after 1640 which was commonly recognized in previous literatures as having poor harvest and famine in this coldest interval of little ice age. There might be some reasons including suddenly increasing number of local chronicles in Qing dynasty that could dilute the drought magnitude based on your grading method or the 50-years moving average can further smoothen the trend. In a word, it will be extremely valuable if the authors can compare the present analysis with previous studies and provide explanations or new perspectives.**

Accepted and revised. Detailed discussion about the abrupt change in frequency of grade 5+6 in harvest has been applied. The jump around 1650s might be resulted from the uncertainty of source data, we added it in the discussion (P13, L4-7). However, the harvest in the whole Qing Dynasty was significantly high indeed, which could also be confirmed by other datasets from independent sources (P9, L19-L23; P15, L14-15).

*In addition, around the 1650–60s, there was a clear jump to grade 5+6 (bumper harvest), yet this period is commonly recognized in previous literatures as having poor harvests and famine in this coldest interval of the Little Ice Age. Also, the periods 1130－1150 and 1210–1270 (the early and later Southern Song Dynasty), 880–980 (later Tang Dynasty and Five Kingdoms) and around 1400, have remarkably more missing harvest grade data.*

*However, harvests were again higher in 1651–1840, corresponding to a cold climate. This can also be confirmed by other datasets from independent sources. For example, according to harvest reports in the Archives of the Qing Dynasty, the mean harvest percentage over eastern China for 1730–1820 was even greater than 70% (i.e. near bumper) (Ge and Wang, 1995). In Guangdong Province, southern China, this was over 75% for 1707–1800 (Marks, 1998).*

*Ge, Q., and Wang, W.-C.: Population Pressure, climate change and Taiping Rebellion, Geogr. Res., 14(4), 32–42, 1995. (in Chinese)*

**Overall, this paper provides new and important insights into the correlations among extreme event, harvest and cold/warm climate. Data statistic is suggested to provide, and since missing data especially for harvest is prominent (35%), it is suggested that authors are more careful to claim their conclusions. Some inconsistency is also found between text and tables, e.g. 49.4% (line 23) and 24.0% (line 24) on page 8 are different from those shown in table 4. Further English editing is strongly suggested to improve high quality writing style of this nice paper.**

Accepted and revised. The inconsistency between text and tables has been revised (P10, L9-10) and the revised and the revised manuscript has been edited for proper English language by LetPub. Certificate of English language editing provided by LetPub is attached below in this response file.

**Certificate of English Language Editing**

[Figure]

**Manuscript Title:**

Extreme droughts and their impacts on harvest derived from historical documents in Eastern China during 801-1910

**Date Issued:**

June 21, 2019

This document certifies that the manuscript listed above was copy edited for proper English language at LetPub. All of our language editors are native English speakers with long-term experience in editing scientific and technical manuscripts. We are committed to leveling the playing field for researchers whose native language is not English.

- Neither the research content nor the authors' intended meaning were altered in any way during the editing process.
- Documents receiving this certification should be considered ready for publication where language issues are concerned. *However, the authors may accept or reject LetPub's suggestions and changes at their own discretion.*
- If you have any questions or concerns about this edited document, please contact us at support@letpub.com

[Figure]

LetPub is an author service brand owned and operated by Accdon LLC. Headquartered in the Boston area, we are a full-spectrum author services company with a large team of US-based certified language and scientific editors, ISO 17001 accredited translators, and professional scientific illustrators and animators. We advocate ethical publication practices and are an official member of the Committee on Publication Ethics (COPE).

© 2018 Accdon, LLC. All Rights Reserved.    Tel: 1-781-202-9968    Email: info@accdon.com    Address: 204 2nd Ave, Waltham, MA 02451, United States

---

## Referee Report (RR1)

Decision: Major Revisions

In my review of the previous version, I identified three sets of problems with the manuscript: first, the author's use of their historical databases; second, the large temporal and spatial scale of the study; and third, the interpretation given to the pattern of correlations found. I believe the authors' responses and revisions have mostly addressed the first set of concerns, and have partly addressed the second set, but have not adequately addressed the third set. Therefore, I recommend further major revisions. The text at the bottom reproduces the original review and the authors' responses and adds my replies (in bold).

The manuscript raises a number of major methodological issues common to much Chinese historical climatology. I believe it essential to address some of these here first in order to help set a precedent for publication in *Climate of the Past*. If, on the one hand, *CP* is to maintain its high standards and, on the other, Chinese historical climatology is become accepted in the global community of (climate) historians, then studies such as this one need to be approached somewhat differently. The task is important because Chinese historical climatology holds considerable promise, including an unparalleled wealth of historical sources and an active community of scholars. Yet that promise remains unfulfilled, because most (climate) historians outside China do not -- indeed, *cannot* -- accept its present methods and results. I therefore apologize for what may seem an unusually long and rigorous review.

Ultimately, the central issue with much Chinese historical climatology -- this manuscript included -- is that it does not study history per se but rather patterns in the data from historical databases compiled by other scholars. The data in those databases and their internal correlations probably reflect actual historical events and causal relationships, but we cannot know to what extent or with what consistency and accuracy, particularly for sources before the late Ming period. Chinese historical climatology studies such as this one rarely include historians who can provide an expert first-hand judgement of the underlying historical sources. Moreover, these studies -- including the present manuscript -- often try to establish causal relations between climatic events and human impacts based on data all drawn from the same historical sources, without adequately considering the ways that biases in reporting and transmission of information might lead to spurious correlations.

What some studies -- including the present manuscript -- do instead, is to try to establish the objectivity of their sources. They assume that if the sources are demonstrably objective by some simple test then what they report can fairly be related as historical fact. Therefore, if patterns emerge in the supposedly factual reporting of two or types of events, then one can extract data about their occurrence and look for patterns to establish actual causal relationships.

This, however, is not a correct method of historical investigation for this purpose. Philosophers of historiography (e.g., Aviezer Tucker) have demonstrated that sound historical research, as a matter of both logic and practice, uses a Bayesian approach to inferring historical conditions and causes from available evidence. In other words, the prior probability of some plausible historical hypothesis about conditions or causation ($p(h)$) should be updated by weighing, on the one hand, the likelihood of the appearance of the available evidence given the hypothesis ($p(o|h)$) and, on the other hand, the likelihood that the same evidence would appear regardless of the truth or falsehood of the hypothesis ($p(o)$). Thus, when researchers make an argument for historical

causation based on a correlation or co-variation among reported incidences of two phenomena, they ought to establish three points: (1) a reasonable prior probability of a historical hypothesis; (2) a high probability of finding the correlation if the hypothesis were true; and (3) a low probability of finding the correlation for any other reason. In this way, researchers may arrive at a high posterior probability (p(h|o)) and thus make a valid historical inference.

The approach taken by Chinese historical climatology -- including the present manuscript -- has problems, first of all, with points (1) and (2) above. That is because it uses a basic inductive approach. It starts with few or no prior hypotheses to test. It simply looks for patterns in the data and then comes up with plausible explanations based only on the variables being considered (usually climatic factors rather than equally plausible factors such as political, economic, demographic or cultural change).

However, the real problem comes when meeting point (3). Normally, (3) might be established by running statistical tests in order to indicate that the correlation or co-variation between the two phenomena -- the proposed cause and proposed effect -- has a low probability of occurring by chance. However, this method is only valid in historical climatology when the reporting of the two phenomena is truly independent and reliable: that is, when they can't be confounded by some other factor, whether in how the phenomena occurred or how the occurrence is reported. If, on the contrary, our knowledge of the (co-)occurrence of the two phenomena depends on reporting and transmission from the same or related historical sources, then there could easily be confounding factors influencing any apparent correlations. Thus, the usual statistical tests no longer suffice to prove point (3) above.

Historical climatologists typically use several methods to confront these problems. They focus on specific historical hypotheses (e.g., the correlation of harvest dates and summer temperatures, or the impact of summer temperatures on grain prices). They create climate indices from documentary evidence but then calibrate and verify these indices with an independent source of information: typically early instrumental measurements or data derived from proxies in natural archives. They make an explicit assumption that any correlation found during the period of overlap between the documentary and instrumental records is more or less constant over time; yet this "principal of stationarity" is applied only to a limited geographical area and type of data (e.g., summer temperatures in France based on grape harvest dates). When making arguments about climate impacts on society, they typically compare their climate data with societal data drawn from historical sources independent of those used to reconstruct climate. All of these steps help to reduce the risk that the ways historical evidence has been recorded, transmitted, and analyzed will generate artefacts in datasets of climate and societal phenomena that lead to spurious correlations and thus false conclusions about causality.

Much current Chinese historical climatology (this manuscript included) does not take such precautions. The datasets on floods and droughts are not calibrated to an independent source of climate information. Both the climatic data, on the one hand, and the harvest data, on the other, come from the same Ming and Qing official sources (principally the *Twenty-Four Histories* and *Qing History Draft*) subject to the same processes of historical reporting and transmission. Therefore, the risk is very high that processes of recording, transmission, interpretation, and recompilation of those historical records has introduced artefacts into the datasets and thus

invalidates supposed causal connections between climate phenomena (in this case, extreme precipitation events) and harvests.

The problem is all the more acute because the basic inductive method described above asks so much of the consistency and independence of the datasets. The authors don't just want to show that drought or flood affected Chinese agriculture in some particular way stated as a prior hypotheses. They want to argue that any patterns they can identify in reported frequencies of floods and/or droughts and reported grades of harvests demonstrates real shifts in the influence of climate. This basic inductive approach not only risks mistaking random fluctuations in the relationships among the data for real shifts in climate impacts; more importantly, it also overlooks the obvious potential for other factors (administrative, geographic, demographic, political, technological) to create patterns in the data, too. Thus, this study, like much Chinese historical climatology, depends too heavily on the homogeneity and consistency of reporting and transmission of information across two millennia of history and millions of square kilometers of territory, across the reigns of scores of emperors and whole new dynasties, over the scope of China's far from homogenous or unified history.

In short, I do not dispute that the data have been properly collected nor do I dispute that the authors have found real patterns in the dataset. I find that the authors have drawn some valid conclusions *about the data*. However, I believe they have not proven their conclusions about the *actual (climate) history* of imperial China.

Nevertheless, I believe that it is worthwhile to publish studies specifically and explicitly about patterns in the data found in Chinese climate and historical databases. First, such studies can help us establish the reliability (or not) and usefulness (or not) of such databases. Second, such studies could form a useful starting point for further historical investigations, even if they cannot point to any firm historical conclusions by themselves.

I would therefore insist on the following further revisions for the publication of this manuscript, and I would strongly encourage their use as standards for similar Chinese historical climatology studies:

1) The title should reflect the fact that this is primarily a study about patterns found in data in historical and climate databases, and only secondarily a study from which we might make inferences about history. In this case, I would accept, for instance, "Patterns in the Reporting of Droughts/Floods and of Harvest Grades in Historical Documents in Eastern China, 801-1910" or "Patterns in Data on Precipitation Extremes and Harvests in Historical Databases for Eastern China 801-1910".

2) The article should clearly distinguish its identification and analysis of data and patterns in that data from any inferences about (climate) history that it makes based on those data and patterns. Those inferences should be more limited, in accordance with the uncertainties and methodological problems outlined above, and should probably appear in the discussion or conclusion of the article. That would also be a good place to discuss whether studies such as Hao et al. 2010 really enable researchers to infer actual historical events and causality from this data.

3) While discussing the data itself and patterns identified in that data, all mentions of historical floods, droughts, and harvest levels should be qualified as "reported" floods or droughts and "reconstructed" harvest levels. Instead of "co-occurrence" of hydrological extremes and poor harvests, the authors should refer to the "co-reporting" of hydrological extremes and poor harvests.

4) Please see the replies below for further specific points.

**Previous Review with Author Responses and Reviewer Replies:**

1) What kinds of droughts are recorded in the historical sources: meteorological drought? hydrological drought? agricultural drought? or some combination of these? Were observers more likely to report precisely those droughts that affected crops, or did they report all droughts equally?
*Accepted and revised. Most of droughts recorded in historical documents were events due to no or*
*less precipitation, and could be regarded as meteorological droughts. Therefore it's reasonable to*
*suggest that they reported all droughts equally rather than those affected crops. (P3, L23-27)*
*As this sample suggests, historical documents were usually focused on the events due to no or less*
*precipitation than usual and, thus could be regarded as meteorological drought rather than*
*hydrological or agricultural droughts, although some records also report impacts on the*
*hydrosphere (e.g., rivers drying up for river) or on agriculture (e.g. wilting for crops). Historical*
*documents, therefore, appear to report all droughts equally, rather than only those affecting*
*crops.*
**Reply: Revision accepted.**

2) What kinds of floods are recorded in the historical sources: heavy rains? tsunamis? rivers that burst their banks? Does the database control for ongoing problems related to river hydrology? How do major events such as course changes in the Yellow River figure into the measure of flood frequency: as one flood? as many?
*Accepted and revised. Similar to the records on drought, flood recorded in historical documents were mostly about more rains or heavy rains. River hydrology events were not taken into account unless it was resulted from more precipitation or heavy rain. (P3, L27-29)*
*Similarly, floods recorded in the historical documents could be regarded as more rain or heavier rain than usual, rather than in the context of rivers bursting their banks or tsunamis, although some*
*records also report the impacts of overflowing or bursting lakes and rivers due to more or heavier*
*precipitation.*
**Reply: Revision accepted.**

3) What is the seasonality of the meteorological events recorded in the historical sources? Does the seasonality of floods or droughts necessarily overlap with the seasonality of critical agricultural activities or phases of crop growth?

*Accepted and revised. The 63-stations annual drought/flood grades was reconstructed and calibrated*
*with descriptions on duration, intensity, and area of the drought/flood events in wet season, which*
*was usually May to September in the study area and overlaps with the critical agricultural activities*
*and phases of crop growth. (P3, L31-P4, L3)*
*Grades were classified using ideal frequency criteria of 10% (grade 1, severe drought), 20% (grade*
*2, drought), 40% (grade 3, normal), 20% (grade 4, flood), and 10% (grade 5, heavy flood) for the*
*whole area and all time. These grades were calibrated based on descriptions of duration, intensity,*
*and area of the drought/flood event during the wet season (usually May to September), and its*
*impact (Table S1). Thus, the season of the drought/flood grade data overlaps with critical*
*agricultural activities and phases of crop growth.*
**Reply: Revision accepted.**

4) What is being measured by "harvest"? Yield per seed? Total yield per hectare? Food availability?
*Accepted and revised. Harvest in records was a relative concept and represented the ratio of actual*
*yield compared to the possible maximum yield. (P6, L1-3)*
*In Chinese historical documents, the yearly harvest was usually recorded as a relative level*
*compared to an expected maximum yield, rather than crop yield per hectare, although some records*
*also report impacts of harvest fluctuation on food availability, tax remissions, livelihoods, and so*
*on.*
**Reply: If this is the case, then the authors need to be explicit throughout the text that what they have reconstructed is not absolute "harvest levels" but rather interannual variability in local production of staple crops.**

5) Are degrees of flood, drought, and harvest based entirely on narrative descriptions, or are there objective phenological or quantitative measures to help define them?
*Accepted and revised. The degrees of flood, drought, and harvest was not based entirely on narrative*
*descriptions. The duration, intensity, and area of these events and their impacts were adopted in*
*calibration, as well. We added two supplementary tables (Table S1, S2) to explain the detailed*
*criteria in grading drought/flood (P3, L30-P4, L2; Table S1) and harvest (P6, L12-15; Table S2).*
*Please refer to the supplementary material which is uploaded additionally as an independent file.*
*Based on these records, Zhang (1996) reconstructed a dataset of annual drought/flood grades at 63*
*stations from 137 BCE. Each station consisted of a local area of approximately 20 counties with the*
*same climate. Grades were classified using ideal frequency criteria of 10% (grade 1, severe*
*drought), 20% (grade 2, drought), 40% (grade 3, normal), 20% (grade 4, flood), and 10% (grade*

*5, heavy flood) for the whole area and all time. These grades were calibrated based on descriptions*
*of duration, intensity, and area of the drought/flood event during the wet season (usually May to September), and its impact (Table S1).*
*In the dataset, yearly harvest levels were classified into 6 grades: 1-Very poor, 2-Poor, 3-Slightly poor, 4-Average, 5-Near bumper, 6-Bumper. The criteria and methods for year-by-year grading of*
*the documentary records (i.e., grain yield descriptions and related information) were presented by*
*Su et al. (2014) and summarized by Yin et al. (2015). The classification of the yearly harvest grade*
*and descriptions recorded in historical documents is shown in Table S2.*
**Reply: Revision accepted.**

Regarding the temporal and spatial scale of the study, I am concerned that it relies on improbable assumptions of continuity and homogeneity in Chinese population, land use, and record keeping. In order to accept as valid any long-term correlations between reported drought or flood frequency and "Chinese" or even "regional" "harvests" I would need the authors to address the following issues:
1) How do the data control for the changing borders of Chinese empires? A priori, I would expect vastly different vulnerabilities and patterns of reporting between the Northern Song and Southern Song periods, simply based on the major geographical shifts in population and wealth between those two dynasties.
*Accepted and revised. For droughts and floods, the historical records was transformed into graded*
*data based on 63-stations, each of which was set as a local area consisted of about 20 counties and*
*does not change in different dynasties (P3, L30-31). Although the available graded data was unevenly distributed spatially for different dynasties, it had been proved in the paper of Hao et al. (2010a) that the extreme drought/flood years recognized were mostly robust despite of the percentage of data-missing stations (P5, L5-11). As for harvest, the impact of changing borders on*
*harvest grade should also be limited since the main grain product area had been relatively stable in*
*the study period and the records in documents was about relative harvest rather than absolute yield*
*as suggested by Yin et al. (2015) (P6, L28-31).*
*Based on these records, Zhang (1996) reconstructed a dataset of annual drought/flood grades at 63*
*stations from 137 BCE. Each station consisted of a local area of approximately 20 counties with the*
*same climate.*
**Reply: This revision is appropriate, but again only if the authors consistently make clear that they are reconstructing interannual variability in local production of staple crops and not an absolute "harvest level." The absolute level of harvests in different regions and eras on decadal or longer scales would still have depended more on changes in political economy, population density, crop strains, and technologies than on weather and climate.**

*To verify the rationality of this method and criteria, validation was conducted in Hao et al. (2010a),*
*based on 10 extreme events identified from a series of precipitation observations in each sub-region*
*according to a threshold of probabilities of 10% and 90% occurrence. In this validation, all or part*
*of grade 3 stations were deliberately omitted, and only 40% or 60% of stations with disaster or*
*extreme grade were reserved without changing the drought-to-flood ratio within the available data.*
*The results show that, with one exception, years of extreme drought and extreme flood, identified*
*according to this method and criteria, closely matched those extreme events identified by*
*precipitation data, demonstrating that the method and criteria were reasonable.*
*However, such social factors should have only limited influence on yearly harvest grade dataset,*
*since the harvest in the documents was reported as a relative level rather than the absolute yield,*
*also the main grain product area, the staple crop, and the cropping system have been relatively*
*stable throughout the study period (Yin et al., 2015).*

**Reply: It appears that Hao et al. 2010 helps establish that the historical sources underlying the databases were not biased toward reporting floods versus droughts or to reporting them in some regions and not others.  That study also appears to establish that these sources did not usually falsely report precipitation extremes.  However, Hao et al. 2010 does not establish that the reporting of precipitation extremes was independent of the reporting of poor food production throughout the long and diverse history of today's China.  This remains a major shortcoming of the study, which I will return to below.**

2) How do data on "harvests" control for changes in staple crops, introduction of New World crops including peanuts and sweet potatoes, changing cropping patterns, and the increasing commercial orientation of agriculture?

*Accepted and revised. The records for harvests were usually about relative percentage compared to*
*expected maximum yield rather than absolute yield, and thus it should not be influenced by these*
*factors. (P6, L1-5)*
*In Chinese historical documents, the yearly harvest was usually recorded as a relative level*
*compared to an expected maximum yield, rather than crop yield per hectare, although some records*
*also report impacts of harvest fluctuation on food availability, tax remissions, livelihoods, and so*
*on. Therefore these harvest records exclude differences in absolute yield between sub-regions with*
*different climates, soil fertility and types, crop varieties, etc., as well as difference between historical*
*periods with changing agricultural centres, farming technologies, staple crops, and so on. (Su et al.,*
*2014).*

**Reply: This revision is appropriate, but again only if the authors consistently make clear that they are reconstructing interannual variability in local production of staple crops and not an absolute "harvest level."  The absolute level of harvests in different regions and eras**

**on decadal or longer scales would still have depended more on changes in political economy, population density, crop strains, and technologies than on weather and climate.**

3) How do the data deal with the changing vulnerabilities to climate variability based on changing settlement patterns even within regions (e.g., uplands in the south and southwest colonized by Han settlers during the late Ming and Qing periods)?
*Accepted and revised. The graded harvest data represents a nationwide status and since the main grain product area had been relatively stable in the study period, the expansion of agricultural area*
*should have limited influence on the yearly harvest documents. (P6, L25-31)*
*During this study period, several social factors existed which could have influenced China's total yield, such as changing borders of empires, the expansion of agricultural area (e.g., uplands in the*
*south and southwest colonized by Han settlers during the late Ming and Qing periods), the updated*
*crop varieties introduced from the New World (e.g., peanuts and sweet potatoes), advanced agricultural management technology, and so on. However, such social factors should have only limited influence on yearly harvest grade dataset, since the harvest in the documents was reported*
*as a relative level rather than the absolute yield, also the main grain product area, the staple crop,*
*and the cropping system have been relatively stable throughout the study period (Yin et al., 2015).*
**Reply: Revision accepted.**

[revised manuscript text omitted]

**Reply: It appears that Hao et al. 2010 helps establish that the historical sources underlying the databases were not biased toward reporting floods versus droughts or to reporting them in some regions and not others. It also appears to establish that these sources did not usually falsely report precipitation extremes. However, Hao et al. 2010 does not establish that the reporting of disasters was independent of the reporting of poor food production throughout the long and diverse history of today's China. This remains a major shortcoming of the study, which I will return to below.**

Third, even if the correlations found in the study are valid, there is a problem with the authors' historical interpretation of them. The correlations discovered here are not between climate and harvests, but rather reports of floods and droughts and reported harvests. The authors assume that the correlations mean that floods and droughts reduced harvests. However, there are a number of potentially confounding variables, which indicate other potential pathways of causality and therefore other historical possibilities:

1. Drought and/or flood might have correlated with other climate variables (such as temperature) that caused harvest failure.

*Accepted. As elaborated in previous study, the relationship between temperature and harvest had been investigated by Yin et al. (2015, 2016), which suggested that there would be better harvest in*

*warm climate. And our study, in section 3.2 of the original manuscript, found that more occurrence*

*of extreme drought in eastern China could lead to significant increase of frequency of poor harvest*

*(grade 1+2) compared with non-extreme years. To further examine whether the drought and/or flood*

*are correlated with temperature change, and if so, how the drought and/or flood are correlated with*

*harvest failure under different temperature backgrounds, we presented a study in section 3.3, and found that there were slightly more extreme droughts in the warm period. However, the connection*

*between extreme droughts and poor harvest was not significantly close in the warm epoch, while it*

*was more significant in the cold epoch. These results suggested that warm period could weaken the*

*impact of extreme drought on poor harvest during historical times. (P11, L18-22; P11, L28-P12, L2)*

*As found in section 3.2, more occurrence of extreme drought in eastern China led to a significant*

*increase in the frequency of poor harvests (grade 1+2) when compared with non-extreme years. Since more extreme droughts occurred over eastern China in 920–1300 than in 1310–1880, the harvest in the warm epoch could be expected to be worse than in the cold epoch. However, as Yin et al. (2015, 2016) found, the harvest in warm epoch was better than that in cold epoch. This suggests that the effects of regional extreme drought on the grain harvest differed between warm and cold epochs.*

*The results show that, during the warm epoch of 920–1300, there was no significant connection between the occurrence of poor harvest and regional extreme drought, although the frequency of poor harvest in extreme drought years was slightly higher than in non-extreme years for each subregion.*

*In contrast, during the cold epoch of 1310–1880, the frequency of poor harvest in extreme drought years was significantly higher than in non-extreme years, which indicates that the connection between the occurrence of poor harvest and extreme drought was still significant. Moreover, similar characteristics were found for the latter half of the cold period from 1650 to 1880,*

*which indicates that the shift of harvest grade distribution did not affect the connection between poor harvest and extreme drought/flood during the cold epoch. These results suggest that the warm*

*period could weaken the impact of extreme drought on poor harvests in historical times.*

**Reply: I do not find that this approach adequately disambiguates the effects of extreme precipitation and temperature on food production. It's a basic inductive method that simply takes all the data and then comes up with an explanation after the fact based on any observed patterns – patterns that could have emerged entirely by chance, or by some confounding third variable (e.g., population movements or changes in political economy that influenced agricultural vulnerabilities), or due to some artefact in the way events were reported or records kept. I'd be much more satisfied if there were some way to model the effects of both temperature variations and extreme precipitation on food production levels so that the authors could weigh relative contributions of each.**

2. Drought and/or flood might have increased the likelihood that officials reported problems such as poor harvests and other disasters

*Accepted. As expressed in abovementioned revisions (P6, L1-5; Table S2), the records on harvests*
*in historical documents was a relative level and focused directly on cropping in most cases, therefore*
*it is reasonable to suggest that there was no tendency in harvest records.*
*In Chinese historical documents, the yearly harvest was usually recorded as a relative level compared to an expected maximum yield, rather than crop yield per hectare, although some records*
*also report impacts of harvest fluctuation on food availability, tax remissions, livelihoods, and so on. Therefore these harvest records exclude differences in absolute yield between sub-regions with*
*different climates, soil fertility and types, crop varieties, etc., as well as difference between historical*
*periods with changing agricultural centres, farming technologies, staple crops, and so on. (Su et al.,*
*2014).*

**Reply: The authors' response does not address my concern. Any causal argument about precipitation extremes and low harvests relies on the assumption that both of these phenomena were consistently reported independently of each other. It appears that Hao et al. 2010 helps establish that the historical sources underlying the databases were not biased toward reporting floods versus droughts or to reporting them in some regions and not others. It also appears to establish that these sources did not usually falsely report precipitation extremes. However, Hao et al. 2010 does not establish that the reporting of precipitation extremes was independent of the reporting of poor food production throughout China's long and diverse history. There remains the strong possibility that officials were more likely to report either the occurrence of such extremes when the harvests were poor (e.g., by way of explaining or justifying those poor harvests) or to mention poor harvests when there were meteorological extremes (e.g., in an effort to take advantage of the situation to secure state funds or tax remissions). I do not make this as merely a theoretical argument. Although I am not an expert on imperial China, this is exactly the pattern I have observed in the records of other early modern empires. Moreover, certain features of this study make this problem particularly troublesome for their conclusions. First, the information concerning both types of events (precipitation extremes and poor harvests) comes from the same set of historical records. Second, the records from earlier eras come primarily from Ming and Qing recompilations and not originals. Third, the older records are very incomplete and biased toward extreme events. Fourth, and most important, the authors' basic inductive method—taking all the data and then explaining any observed patterns on the assumption that they are causally related— sets a very high standard for the independence of the different datasets. In other words, if the authors were simply making the case that over the long run we should see an impact of floods and/or droughts on interannual food production, then small changes in political priorities or record-keeping and transmission practices shouldn't make a big difference. However, because the authors are identifying shifts and patterns over time—such as possible changes in the impact of precipitation extremes due to phases of warming and cooling—then any shifts in the independence of reporting the occurrence of extreme precipitation events and of poor harvests are likely show up in their presentation of the data and to be explained as real historical changes in the impact of floods and droughts on food production. In fact, that is exactly what I think has happened in this study, and that is why I am asking for major revisions.**

3. Harvest failures might have increased the likelihood that officials reported disasters such as droughts and/or floods.
*Accepted. As mentioned before, the droughts and floods records in historical documents were usually focused on abnormal precipitation, and appeared to report all extremes equally. (P3, L23-*
*29)*
*As this sample suggests, historical documents were usually focused on the events due to no or less*

*precipitation than usual and, thus could be regarded as meteorological drought rather than hydrological or agricultural droughts, although some records also report impacts on the hydrosphere (e.g., rivers drying up for river) or on agriculture (e.g. wilting for crops). Historical documents, therefore, appear to report all droughts equally, rather than only those affecting crops.*

*Similarly, floods recorded in the historical documents could be regarded as more rain or heavier rain than usual, rather than in the context of rivers bursting their banks or tsunamis, although some*
*records also report the impacts of overflowing or bursting lakes and rivers due to more or heavier*
*precipitation.*
**Reply: (see previous reply)**

4. Droughts and/or floods might have harmed human and animal health, reducing labor for harvests.
*Accepted and revised. We added these possible pathways for the connection between extreme events*
*and poor harvest in the revised manuscript. (P10, L25-28)*
*This relationship may have been caused by both reductions in water supply and other indirect*
*pathways. For example, droughts might harm human and domestic animal health or result in*
*migration, leading to a reduced agricultural labour force. In addition, extreme events might reduce*
*public revenue or increase public expenses, thus increasing political and economic instability, and*
*further affecting agricultural activities (Zheng et al., 2014a).*
**Reply: Revision accepted.**

5. Droughts and/or floods might have damaged infrastructure and transportation, leading to food availability decline.
*Accepted. Most of the yearly harvest records were direct wording on assessment of crop yield, which could not be influenced by damaged grain transportation. (Table S2)*
**Reply: Accepted.**

6. Droughts and/or floods might have driven migrations, creating regional shortages both where agricultural labor emigrated and where people arrived seeking food.
*Accepted and revised. This possible pathway has been addressed in revised manuscript along with*
*pathway 4. (P10, L25-28)*
*This relationship may have been caused by both reductions in water supply and other indirect*
*pathways. For example, droughts might harm human and domestic animal health or result in*
*migration, leading to a reduced agricultural labour force. In addition, extreme events might reduce*
*public revenue or increase public expenses, thus increasing political and economic instability, and*
*further affecting agricultural activities (Zheng et al., 2014a).*
**Reply: Revision accepted.**

7. Periods of drought and/or flood might have reduced public revenue and/or increased public expenses, thus increasing the political and economic instability and decreasing food availability. (For instance, it's not clear how much the figures overall are influenced by the very high frequency of disasters and widespread famine during the political turbulence and violence accompanying the collapse of the Ming dynasty.) I am not arguing that any of these

scenarios is necessarily the case. Nevertheless, each of these may be influencing the observed correlations.

*Accepted and revised. This possible pathway has also been addressed in revised manuscript along*

*with pathway 4 and 6. (P10, L25-28)*

*This relationship may have been caused by both reductions in water supply and other indirect pathways. For example, droughts might harm human and domestic animal health or result in migration, leading to a reduced agricultural labour force. In addition, extreme events might reduce*

*public revenue or increase public expenses, thus increasing political and economic instability, and*

*further affecting agricultural activities (Zheng et al., 2014a).*

**Reply: Revision accepted.**

---

## Referee Report (RR2)

**Minor revisions**

The authors have accepted the key changes requested in the previous reviews. However, they have not been entirely consistent in applying those changes. It is important that throughout the text the authors carefully distinguish (1) the analysis of data in these databases of historical weather and harvests from (2) their interpretation of the underlying climatic and human history. Therefore, I would request the following minor revisions:

The abstract has added terms such as "reported" but needs to rework the language more carefully to reflect the new approach of the article. The following changes would be appropriate in order to make the meaning more precise and avoid inaccurate claims:
- "reported extreme droughts [floods] occurred" should be: "extreme droughts [floods] were reported" (or a similar phrase). The key distinction here is a report of a drought [flood] is not the same as the historical occurrence of drought [flood].
- "reconstructed grain harvest was poor [medium, high] in" should be: "the grain harvest was reconstructed as poor [medium, high] for" (or a similar phrase).
- "occurrence of reported extreme drought in any sub-region of eastern China was significantly associated with reduced harvests in the long-term average" should be: "frequency of reporting of extreme droughts was significantly associated over the long term with lower reconstructed harvests" (or a similar phrase).
- "association between harvest and extreme floods" should be: "associated between the reported frequency of extreme floods and reconstructed low harvests" (or a similar phrase).
- "other social factors" should be "other historical factors" to include other historical environmental changes, both natural and anthropogenic

On page 2, lines 20-34 have not been reworked to reflect the new approach of the article -- that is, to first discuss patterns in the data derived from the historical documents, and only then to discuss the interpretations of those patterns as historical climate impacts. This paragraph makes the unwarranted assumption that the reported frequencies of events in the historical documents represent real frequencies and that associations between the frequencies of reported disasters and extremes and variations in reconstructed grain harvests represent causation (i.e., climate impacts). The studies cited by Su et al. and Yin et al. present the same problems of historical method and epistemology as did the previous draft of this manuscript, as discussed in my previous review. Therefore, their results need to be qualified in the same manner. The authors may refer the reader to the discussion section for their causal interpretation.

In the sentence at the bottom of page 4 and top of page 5, it is important that the authors clarify that their method is based *on an assumption* that the probabilities for omitting drought and flood events in reporting and transmission of historical records were random and unbiased, for the reasons they have stated. They haven't actually proven that omissions were random and unbiased. They have merely made a reasonable argument that it would be appropriate to proceed on this assumption.

Now that the authors have worked to distinguish their results (that is, the patterns and associations in their datasets) from their discussion (that is, the climate and historical interpretation of those associations), their use of a combined "Results and Discussion" section has become more confusing. For the sake of clarity, I would encourage the authors to rename

section 3 as simply "Results" and turn subsection 3.4 into a new section, "Discussion."  Their discussion of results in the bottom of page 9 to the top of page 10 as well as their discussion of results at the top of page 12 might then be moved into the new "Discussion" section, as part of the authors' climatic and historical interpretation of the patterns in the data.  The fact that these pattern makes sense from a meteorological perspective supports the case that droughts had a significant historical impact on grain harvests.  The authors may wish to state this point clearly in their discussion.

The revised sections, while mostly clear, should receive further review for correct English grammar and word use before publication.

---

## Author Response (AR2)

**Reply to the referee comments on "Extreme droughts/floods and their impacts on harvest derived from historical documents in Eastern China during 801–1910" by Zhixin Hao et al**

Dear editors and reviewers,

Thank you for your valuable comments and thoughtful suggestions on our manuscript. Following your comments on the manuscript, we made careful revisions, and the point-to-point response (in red) of the comments (in black) is listed below. We hope these revisions would make this manuscript more acceptable for publication. Please feel free to contact me if you have any questions.

Many thanks again. With best wishes.

Sincerely yours,

Jingyun Zheng

**To Anonymous Referee #1:**

After we carefully read the comments, as our understanding, the most concern from reviewer is that the dataset of droughts/floods is not independent of the dataset of harvest grades, since they were reconstructed from the same historical documents. We thought that the spurious correlation induced by historical documents' recording and circulation could be limited for three reasons. First, the two datasets were not from the same documents, in which the reconstructed harvest grades were derived from the records in "Twenty-four histories" and "Qing history Draft", while the reported droughts/floods were derived not only from "Twenty-four histories" and "Qing history Draft", but also chronicles, miscellaneous historical books, local gazettes and others. Secondly, as our reading experience in the historical documents, we did not find large number of records reporting both extreme events and harvests simultaneously. Thirdly, during the reconstruction of droughts/floods and harvest grades, the record collection, explanation and calibrated parameters were not the same. Although as the reviewer suggested, this problem could not be completely avoided indeed. So we accepted all the comments, and deleted the historical inferences such as "occurrence of extreme drought in eastern China could lead to significant increase of frequency of poor harvest". The possible inferences according to statistics are now presented in the *Discussion* section, along with discussions on the data independence problem and confounding factors. Also, the "reported" droughts/floods and "reconstructed" harvest grades are highlighted explicitly throughout the text to make clear that all the results are focused on the two datasets, instead of

actual historical events. The detailed point-to-point responses are listed below.

1) The title should reflect the fact that this is primarily a study about patterns found in data in historical and climate databases, and only secondarily a study from which we might make inferences about history. In this case, I would accept, for instance, "Patterns in the Reporting of Droughts/Floods and of Harvest Grades in Historical Documents in Eastern China, 801-1910" or "Patterns in Data on Precipitation Extremes and Harvests in Historical Databases for Eastern China 801-1910".

Accepted. The title has been changed to "Patterns in data of extreme droughts/floods and harvest grades derived from historical documents in Eastern China during 801–1910".

2) The article should clearly distinguish its identification and analysis of data and patterns in that data from any inferences about (climate) history that it makes based on those data and patterns. Those inferences should be more limited, in accordance with the uncertainties and methodological problems outlined above, and should probably appear in the discussion or conclusion of the article. That would also be a good place to discuss whether studies such as Hao et al. 2010 really enable researchers to infer actual historical events and causality from this data.

Accepted. The results (section 3.1-3.3) are now focused on analysis of datasets, while the possible inferences are presented in discussions. (P12, L22-P13, L3)

*The statistical results from these two datasets indicate that regional extreme droughts might be closely connected with poor harvest in Chinese history, and this connection seems to be weaker in the warm period and stronger in the cold period. However, these inferences are purely based on those two reconstructed datasets, and insufficient to reveal actual historical connections. One of the reasons is that both datasets used Twenty-Four Histories and Qing History Draft as their record resources in the reconstruction, which might induce artefact in databases and lead to spurious correlations between extreme drought/flood and harvest. Another reason is that there existed major shifts in many social factors (such as political, economic, demographic or cultural change) in the study period, which means that the different pattern in association between reported extreme drought and reconstructed poor harvest between warm and cold periods might be co-created by those confounding factors along with climate factor. Although the dataset for extreme droughts/floods used more historical documents and were not limited to the Twenty-Four Histories and Qing History Draft as resources, and both datasets have been validated by other proxy data in previous studies, the independence and consistency problems could not be eliminated entirely. The solution for these two problems requires further research using more*

*independent datasets from multi-proxies and many other study fields, and could be of great importance in improving the understanding of the climatic impacts on agriculture and adaptation to future global warming along with higher extreme climate probability.*

3) While discussing the data itself and patterns identified in that data, all mentions of historical floods, droughts, and harvest levels should be qualified as "reported" floods or droughts and "reconstructed" harvest levels. Instead of "co-occurrence" of hydrological extremes and poor harvests, the authors should refer to the "co-reporting" of hydrological extremes and poor harvests.

Accepted. In the revised manuscript, when the droughts, floods and harvests are mentioned, the word "reported" and "reconstructed" are added explicitly to make clear that they refer to reconstructed datasets rather than actual historic events. The use of "co-occurrence" are also revised as "co-reporting".

4) Please see the replies below for further specific points.

In the reviewer comments, all the comments and author responses from previous review are listed, along with reviewer's reply to each of them. Some of our previous responses are accepted, and some are still disputable to the reviewer. Here we only present the comments which need more revisions according to the reviewer's comment.

4.1.4) What is being measured by "harvest"? Yield per seed? Total yield per hectare? Food availability?

Accepted and revised. Harvest in records was a relative concept and represented the ratio of actual yield compared to the possible maximum yield. (P6, L1-3)

*In Chinese historical documents, the yearly harvest was usually recorded as a relative level compared to an expected maximum yield, rather than crop yield per hectare, although some records also report impacts of harvest fluctuation on food availability, tax remissions, livelihoods, and so on.*

**Reply: If this is the case, then the authors need to be explicit throughout the text that what they have reconstructed is not absolute "harvest levels" but rather interannual variability in local production of staple crops.**

Accepted. We replaced "harvest level" with "harvest grade" in the text, and emphasized the definition for "harvest grade" in the *Data* section. (P6, L13-14)

*It should be noted that these reconstructed grades do not represent absolute grain yield, but rather*

*the relative percentage in production of staple crops and reflect their inter-annual variability.*

4.2.1) How do the data control for the changing borders of Chinese empires? A priori, I would expect vastly different vulnerabilities and patterns of reporting between the Northern Song and Southern Song periods, simply based on the major geographical shifts in population and wealth between those two dynasties.

Accepted and revised. For droughts and floods, the historical records was transformed into graded data based on 63-stations, each of which was set as a local area consisted of about 20 counties and does not change in different dynasties (P3, L30-31). Although the available graded data was unevenly distributed spatially for different dynasties, it had been proved in the paper of Hao et al. (2010a) that the extreme drought/flood years recognized were mostly robust despite of the percentage of data-missing stations (P5, L5-11). As for harvest, the impact of changing borders on harvest grade should also be limited since the main grain product area had been relatively stable in the study period and the records in documents was about relative harvest rather than absolute yield as suggested by Yin et al. (2015) (P6, L28-31).

*Based on these records, Zhang (1996) reconstructed a dataset of annual drought/flood grades at 63 stations from 137 BCE. Each station consisted of a local area of approximately 20 counties with the same climate.*

**Reply: This revision is appropriate, but again only if the authors consistently make clear that they are reconstructing interannual variability in local production of staple crops and not an absolute "harvest level." The absolute level of harvests in different regions and eras on decadal or longer scales would still have depended more on changes in political economy, population density, crop strains, and technologies than on weather and climate.**

Accepted. It is the same with comment 4.1.4. We replaced "harvest level" with "harvest grade" in the text, and emphasized the definition for "harvest grade" in the *Data* section. (P6, L13-14)

*It should be noted that these reconstructed grades do not represent absolute grain yield, but rather the relative percentage in production of staple crops and reflect their inter-annual variability.*

*To verify the rationality of this method and criteria, validation was conducted in Hao et al. (2010a), based on 10 extreme events identified from a series of precipitation observations in each sub-region according to a threshold of probabilities of 10% and 90% occurrence. In this validation, all or part of grade 3 stations were deliberately omitted, and only 40% or 60% of stations with disaster or extreme grade were reserved without changing the drought-to-flood ratio within the available data. The results show that, with one exception, years of extreme drought and*

*extreme flood, identified according to this method and criteria, closely matched those extreme events identified by precipitation data, demonstrating that the method and criteria were reasonable.*

*However, such social factors should have only limited influence on yearly harvest grade dataset, since the harvest in the documents was reported as a relative level rather than the absolute yield, also the main grain product area, the staple crop, and the cropping system have been relatively stable throughout the study period (Yin et al., 2015).*

**Reply: It appears that Hao et al. 2010 helps establish that the historical sources underlying the databases were not biased toward reporting floods versus droughts or to reporting them in some regions and not others. That study also appears to establish that these sources did not usually falsely report precipitation extremes. However, Hao et al. 2010 does not establish that the reporting of precipitation extremes was independent of the reporting of poor food production throughout the long and diverse history of today's China. This remains a major shortcoming of the study, which I will return to below.**

Accepted. The paper of Hao et al. (2010) only introduced the reconstruction of droughts/floods, and did not write about harvest grades. As mentioned in the first paragraph of this response, we revised the expression in the results, and made discussion on the flaw of data reflecting the real facts. (P12, L22-L27)

*The statistical results from these two datasets indicate that regional extreme droughts might be closely connected with poor harvest in Chinese history, and this connection seems to be weaker in the warm period and stronger in the cold period. However, these inferences are purely based on those two reconstructed datasets, and insufficient to reveal actual historical connections. One of the reasons is that both datasets used Twenty-Four Histories and Qing History Draft as their record resources in the reconstruction, which might induce artefact in databases and lead to spurious correlations between extreme drought/flood and harvest.*

4.2.2) How do data on "harvests" control for changes in staple crops, introduction of New World crops including peanuts and sweet potatoes, changing cropping patterns, and the increasing commercial orientation of agriculture?

Accepted and revised. The records for harvests were usually about relative percentage compared to expected maximum yield rather than absolute yield, and thus it should not be influenced by these factors. (P6, L1-5)

*In Chinese historical documents, the yearly harvest was usually recorded as a relative level compared to an expected maximum yield, rather than crop yield per hectare, although some*

*records also report impacts of harvest fluctuation on food availability, tax remissions, livelihoods, and so on. Therefore these harvest records exclude differences in absolute yield between sub-regions with different climates, soil fertility and types, crop varieties, etc., as well as difference between historical periods with changing agricultural centres, farming technologies, staple crops, and so on. (Su et al., 2014).*

**Reply: This revision is appropriate, but again only if the authors consistently make clear that they are reconstructing interannual variability in local production of staple crops and not an absolute "harvest level." The absolute level of harvests in different regions and eras on decadal or longer scales would still have depended more on changes in political economy, population density, crop strains, and technologies than on weather and climate.**

Accepted. It is the same with previous comments 4.1.4 and 4.2.1. We replaced "harvest level" with "harvest grade" in the text, and emphasized the definition for "harvest grade" in the *Data* section (P6, L13-14).

*It should be noted that these reconstructed grades do not represent absolute grain yield, but rather the relative percentage in production of staple crops and reflect their inter-annual variability.*

4.2.5) Most importantly, how can we make up for the fact there are simply more records from the Qing period than earlier periods? I don't see that the methods used in this manuscript avoid the problem that more records will create a misimpression of a greater frequency of floods and droughts. The authors propose to ignore reports of "average" conditions in Qing records to make them more comparable to Song and Ming records. However, that would only work if the Song and Ming records still reliably reported all disasters and extremes and only left out "average" conditions. I don't see any reason to make that assumption. Perhaps the authors could experiment with methods of introducing "noise" into the data in order to reflect the events missing from the reports. Or else they could employ a Bayesian method to indicate that the presence or absence of certain descriptions in the records may be used to obtain updated posterior probabilities of actual conditions, without ever assuming that the records provide a complete account of events. In any case, the authors must come up with a way to handle these changes in the documentary record over time if they are to make a convincing case for stable long-term correlations between floods and droughts and harvests.

Accepted and revised. The method of ignoring "average" conditions is based on the hypothesis that the records on droughts and floods were omitted randomly and unbiased, which suggests that the relative drought-to-flood ratio in the available data would be close to that in actual history. As in the abovementioned revisions, the recognition for extreme drought and flood years was still

effective even if 40% or 60% of the available data with disasters was omitted deliberately. (P4, L34-P5, L13)

*Extreme drought or extreme flood years were defined in this way, as the probabilities for omitting drought and flood records were random and unbiased, despite the greater frequency of missing data in the older records. In other words, if one period had a large number of documents, it was expected to be rich in both drought and flood records, and vice versa. Therefore, the amount of missing data should not have a significant effect on the relative drought-to-flood ratio within the available data. To verify the rationality of this method and criteria, validation was conducted in Hao et al. (2010a), based on 10 extreme events identified from a series of precipitation observations in each sub-region according to a threshold of probabilities of 10% and 90% occurrence. In this validation, all or part of grade 3 stations were deliberately omitted, and only 40% or 60% of stations with disaster or extreme grade were reserved without changing the drought-to-flood ratio within the available data. The results show that, with one exception, years of extreme drought and extreme flood, identified according to this method and criteria, closely matched those extreme events identified by precipitation data, demonstrating that the method and criteria were reasonable. The reason for the close match is that precipitation variability in eastern China is dominated by the East Asian Summer Monsoon (EASM). Therefore, when extreme drought or flood events occur, the precipitation variation for stations within each sub-region usually share similar relative magnitudes.*

**Reply: It appears that Hao et al. 2010 helps establish that the historical sources underlying the databases were not biased toward reporting floods versus droughts or to reporting them in some regions and not others. That study also appears to establish that these sources did not usually falsely report precipitation extremes. However, Hao et al. 2010 does not establish that the reporting of precipitation extremes was independent of the reporting of poor food production throughout the long and diverse history of today's China. This remains a major shortcoming of the study, which I will return to below.**

Accepted. It is the same with second part of comment 4.2.1. The paper of Hao et al. (2010) only introduced the reconstruction of droughts/floods, and did not write about harvest grades. As mentioned in the first paragraph of this response, we revised the expression in the results, and made discussion on the flaw of data reflecting the real facts. (P12, L22-L27)

*The statistical results from these two datasets indicate that regional extreme droughts might be closely connected with poor harvest in Chinese history, and this connection seems to be weaker in the warm period and stronger in the cold period. However, these inferences are purely based on those two reconstructed datasets, and insufficient to reveal actual historical connections. One of*

*the reasons is that both datasets used Twenty-Four Histories and Qing History Draft as their record resources in the reconstruction, which might induce artefact in databases and lead to spurious correlations between extreme drought/flood and harvest.*

4.3.1) Drought and/or flood might have correlated with other climate variables (such as temperature) that caused harvest failure.

Accepted. As elaborated in previous study, the relationship between temperature and harvest had been investigated by Yin et al. (2015, 2016), which suggested that there would be better harvest in warm climate. And our study, in section 3.2 of the original manuscript, found that more occurrence of extreme drought in eastern China could lead to significant increase of frequency of poor harvest (grade 1+2) compared with non-extreme years. To further examine whether the drought and/or flood are correlated with temperature change, and if so, how the drought and/or flood are correlated with harvest failure under different temperature backgrounds, we presented a study in section 3.3, and found that there were slightly more extreme droughts in the warm period. However, the connection between extreme droughts and poor harvest was not significantly close in the warm epoch, while it was more significant in the cold epoch. These results suggested that warm period could weaken the impact of extreme drought on poor harvest during historical times. (P11, L18-22; P11, L28-P12, L2)

*As found in section 3.2, more occurrence of extreme drought in eastern China led to a significant increase in the frequency of poor harvests (grade 1+2) when compared with non-extreme years. Since more extreme droughts occurred over eastern China in 920–1300 than in 1310–1880, the harvest in the warm epoch could be expected to be worse than in the cold epoch. However, as Yin et al. (2015, 2016) found, the harvest in warm epoch was better than that in cold epoch. This suggests that the effects of regional extreme drought on the grain harvest differed between warm and cold epochs.*

*The results show that, during the warm epoch of 920–1300, there was no significant connection between the occurrence of poor harvest and regional extreme drought, although the frequency of poor harvest in extreme drought years was slightly higher than in non-extreme years for each sub-region. In contrast, during the cold epoch of 1310–1880, the frequency of poor harvest in extreme drought years was significantly higher than in non-extreme years, which indicates that the connection between the occurrence of poor harvest and extreme drought was still significant. Moreover, similar characteristics were found for the latter half of the cold period from 1650 to 1880, which indicates that the shift of harvest grade distribution did not affect the connection between poor harvest and extreme drought/flood during the cold epoch. These results suggest that*

*the warm period could weaken the impact of extreme drought on poor harvests in historical times.*

**Reply: I do not find that this approach adequately disambiguates the effects of extreme precipitation and temperature on food production. It's a basic inductive method that simply takes all the data and then comes up with an explanation after the fact based on any observed patterns – patterns that could have emerged entirely by chance, or by some confounding third variable (e.g., population movements or changes in political economy that influenced agricultural vulnerabilities), or due to some artefact in the way events were reported or records kept. I'd be much more satisfied if there were some way to model the effects of both temperature variations and extreme precipitation on food production levels so that the authors could weigh relative contributions of each.**

Accepted. In the revised manuscript, the idea of making simple historical inferences from patterns in the data has been revised. The inappropriate subjective expressions, e.g., "the impact of extreme floods/droughts on the harvest", "connections between climate and harvest", have been changed to more objective expressions, such as "association between reported droughts/floods and reconstructed harvest grades". The result sections focus only on the patterns from the datasets, and possible inferences and confounding factors have been put in the discussion section.

In addition, since the reconstructed climate datasets have different time resolutions (e.g., reconstructed temperature has a decadal resolution, while reported droughts/floods dataset has an annual resolution), and the response of food production to each climate factor has different sensitivity and time scale, so it is difficult to weigh the relative contributions of each climate factor. Even under the modern observation condition, it is not easy to answer this question, which needs large data sample size or model simulation method. Therefore the revised manuscript did not induce discussions on the separate effects of temperature variations and extreme precipitation on food production. This question might be addressed in our future studies.

4.3.2) Drought and/or flood might have increased the likelihood that officials reported problems such as poor harvests and other disasters

Accepted. As expressed in abovementioned revisions (P6, L1-5; Table S2), the records on harvests in historical documents was a relative level and focused directly on cropping in most cases, therefore it is reasonable to suggest that there was no tendency in harvest records.

*In Chinese historical documents, the yearly harvest was usually recorded as a relative level compared to an expected maximum yield, rather than crop yield per hectare, although some records also report impacts of harvest fluctuation on food availability, tax remissions, livelihoods, and so on. Therefore these harvest records exclude differences in absolute yield between*

*sub-regions with different climates, soil fertility and types, crop varieties, etc., as well as difference between historical periods with changing agricultural centres, farming technologies, staple crops, and so on. (Su et al., 2014).*

**Reply: The authors' response does not address my concern. Any causal argument about precipitation extremes and low harvests relies on the assumption that both of these phenomena were consistently reported independently of each other. It appears that Hao et al. 2010 helps establish that the historical sources underlying the databases were not biased toward reporting floods versus droughts or to reporting them in some regions and not others. It also appears to establish that these sources did not usually falsely report precipitation extremes. However, Hao et al. 2010 does not establish that the reporting of precipitation extremes was independent of the reporting of poor food production throughout China's long and diverse history. There remains the strong possibility that officials were more likely to report either the occurrence of such extremes when the harvests were poor (e.g., by way of explaining or justifying those poor harvests) or to mention poor harvests when there were meteorological extremes (e.g., in an effort to take advantage of the situation to secure state funds or tax remissions). I do not make this as merely a theoretical argument. Although I am not an expert on imperial China, this is exactly the pattern I have observed in the records of other early modern empires. Moreover, certain features of this study make this problem particularly troublesome for their conclusions. First, the information concerning both types of events (precipitation extremes and poor harvests) comes from the same set of historical records. Second, the records from earlier eras come primarily from Ming and Qing recompilations and not originals. Third, the older records are very incomplete and biased toward extreme events. Fourth, and most important, the authors' basic inductive method—taking all the data and then explaining any observed patterns on the assumption that they are causally related— sets a very high standard for the independence of the different datasets. In other words, if the authors were simply making the case that over the long run we should see an impact of floods and/or droughts on interannual food production, then small changes in political priorities or record-keeping and transmission practices shouldn't make a big difference. However, because the authors are identifying shifts and patterns over time—such as possible changes in the impact of precipitation extremes due to phases of warming and cooling—then any shifts in the independence of reporting the occurrence of extreme precipitation events and of poor harvests are likely show up in their presentation of the data and to be explained as real historical changes in the impact of floods and droughts on food production. In fact, that is exactly what I think has happened in this**

**study, and that is why I am asking for major revisions.**

Accepted. In the revised manuscript, the historical inferences with insufficient evidence has been revised, and the result sections focus only on the statistics and patterns extracted from the two datasets. Although several explanations should be made toward some aspects of this comment. For the first part of this comment, the two datasets are not from same documents, in which the reconstructed harvest grades were derived from the records in "Twenty-four histories" and "Qing history Draft", while the reported droughts/floods events were derived not only from "Twenty-four histories" and "Qing history Draft", but also chronicles, miscellaneous historical books, local gazettes and others. For the second part of this comment, the recompilation of the Ming and Qing histories was only a systematic documents reorganization, and did not add new subjective points of view or more contents related with droughts/floods or harvests. For the third part of this comment, the problem induced by incomplete and biased older records could be avoided to a large extent, because the statistics focus on the relative ratio between extreme droughts and extreme floods. This ratio should not be influenced by the percentage of missing data, since there is no tendency in omitting drought or flood records. In addition, we divided the whole eastern China into three sub-regions based on the movement of Asian Summer Monsoon, and the large number of sites with similar precipitation characteristics in each sub-regions could compensate the shortcoming of incomplete records in the early period to a certain degree.

**To Anonymous Referee #2:**

The author has made an effort to insert a paragraph describing data characteristics and basic statistics. This definitely helps to increase data transparency procedure. However their explanations about the nature of the written records, interpretation of the records, and the data statistics are not rigorous and likely to incur more problems at the epistemological level. In fact, from my perspective, the authors do not really face the real challenges that the reviewers proposed to them and just to repeat the problems in the paper or to ignore. Below are just some of the examples.

Accepted. Same as the comments given by reviewer #1, in the revised manuscript we discussed the flaw of the dataset, and especially emphasized that the results only reflect patterns extracted from two datasets instead of actual historical conditions. We weakened the description of "impact" or "connection", and only wrote about the existing phenomena about "reported" extreme events and "reconstructed" harvests.

In addition, the uncertainty of the historical documents indeed existed, which need more historical

document resources, more datasets from other proxies, and new methods such as model simulation. In order to keep the continuity, we expect to address this problem with more datasets and model simulation method in the future study.

On page 3 line 23-29, the authors strictly assert that drought records in the historical documents can be regarded as meteorological drought rather than hydrological or agricultural drought, and same declaration also appears for flood records that reflect more rain rather than river or lake overflowing etc. The declarations are hard to be accepted or digested literally because it is obvious that both drought and flood phenomena and the intensity can be largely influenced by the population size, biological and geographical conditions even today. Thus, when authors purely declare so without providing further justifications, this can lead to a non-negligible mistrust and suspicion from the readers to question the reliability of the record data and the analysis.

As showed in the data example (P3, L13-15), the documents at the 19th year of emperor Zhenyuan in Tang Dynasty (803 CE) record: "*no rain fell in the Guanzhong area (now Xi'an and surrounding areas) from the 1st month (January 27 to February 24) until now (In day 25 of the 6th month (of the Chinese lunar calendar, or July 17, 803 in the solar calendar). Hundreds of officials and many people are praying for rain. In day 26 of the 7th month (August 18), rain. In day 17 of the 8th month (September 6), heavy rain*". The record has explicit date, type of rain (intensity), duration for the rainfall process or for long-lasting drought. Many other records also recorded the amount of precipitation through the number of inches. These characteristics are not related with population, land cover and geographic condition.

Also it is not clear for me on page 4 line13-16, how the number of 70% of ….can be derived. It is like the authors' assertion without much logical reasonability to follow.

Accepted. We rewrote this sentence. (P4, L13-15)

*Even so, it is reasonable to infer that a considerable proportion of extreme events was recorded in 800-1469, comparing the percentage of records reporting "disasters and extremes" in that period and in the ideal frequency (41.8% compared with 60%, respectively).*

The added content for validating the grade method on page 4 line 5-10 is somehow not surprising since in this case only stations with extreme grade and disaster were kept (omit grade 3, 40% of data) for comparing with precipitation data (of what years?) that seem to show good match between two sets of data.

Yet, in the first referee report, a clear jump of bumper harvest records (Figure 4) was notified that

corresponds to the coldest interval of LIA and many famine records exist in other documents. All those are against the authors' findings in the study. But the authors only respond to this comment by repeating this 'jump point' in the paper(p13 in revision), without more in-depth reflections or discussions. Actually, from my viewpoint, the number of good harvest can simply be exaggerated by the dramatically increased number of official documents and local chronicle staring Qing dynasty (1644 CE), and the kind of the very basic record/data issues has not been seriously taken by the authors resulting in severe degradation of the research quality of the paper.

First, the reconstructed harvest grades do not represent absolute grain yield, but rather the relative percentage in crop production and reflect their inter-annual variability, which is not influenced by the number of documents. Secondly, as fig.4c shows, the "jump point" still exists when all "average" harvest (grade 3) and missing data are excluded. As addressed in *Data* section, the probabilities for omitting each side of extremes in records are supposed to be random and unbiased, despite the greater frequency of missing data in the older records. When the total amount of records significantly increased in Qing Dynasty, it is expected to find that amounts of records on poor harvest and bumper harvest both increased, and the relative ratio between poor and bumper harvest should not be influenced by the change in number of records. In addition, according to fig.4c, there's also a similar abundant period for poor harvest (grade 1+2) during 1260-1530, yet the number of records in that period did not decrease significantly, which also indicates that the jump in relative percentage of harvest grades is not associated with number of records.

---

## Author Response (AR3)

**Point by point response to the reviewer and editor's comments on "Patterns in data of extreme droughts/floods and harvest grades derived from historical documents in Eastern China during 801–1910"**

Dear editor and reviewer,

We appreciate the positive and constructive comments and suggestions from the reviewer and the editor. We have revised our manuscript and all the modifications are highlighted by red color. Our point-to-point response (in red) is listed below.

Thank you for your consideration.

Sincerely yours,

Jingyun Zheng

**To editor:**

The manuscript did improve in all major points previously addressed by the reviewers. Still there some smaller points that should be solved: please, follow the advices of Reviewer 1. Additionally, I have two further comments that, I think, needs further consideration:

1) While drought is often a large-scale phenomenon, the impacts of floods are mostly restricted to the areas directly affected by floods. As the authors only discuss floods seemingly as a counterpoint of droughts (and it is not exactly that), in the latter case, the authors could as well mention that much precipitation / wet periods (with or without extraordinary floods) could as well be responsible for (larger-scale) bad harvest and harvest failures.

*Accepted. We added the text about much precipitation could as well be responsible for bad harvest. (P12, L25-27)*

*Although it should be noted that these meteorological interpretations only apply for short-term extreme flood events. When it comes to long-term wet periods, it could as well be responsible for bad harvest, due to the low-temperature and short sunlight.*

2) There is a clear and well-visible improvement in the 2.1 Data subchapter with examples from original source references and also with listing major advantages of the compiled dataset and indices they apply. However, the authors could also address any weaknesses of their databases, as their data are mainly available in compilations, and usually compilations are based on all kinds of contemporary and non-contemporary data mixed, with the (sometimes high) probability of misdating, doubling (tripling) events etc. Consequently, majority of the Chinese database, the paper is based on, is only available in a form that retracing the sources and the reconstruction steps ate not possible. Naturally, the authors cannot solve this problem all within this article; nevertheless, in one sentence (e.g. on page 4, btw. lines 20-25, after listing all advantages of the dataset) they should refer to this general problem or weakness.

Accepted. The weakness about this database has been added. (P4, L30-32)

*However, this dataset also has weaknesses. For example, the reconstruction derived from the historical documents relies highly on the accuracy of the compilation, and is only available in a form that retracing of the original sources and reconstruction steps are not same as reconstructions from natural evidences.*

**To Anonymous Referee #1:**

The authors have accepted the key changes requested in the previous reviews. However, they have not been entirely consistent in applying those changes. It is important that throughout the text the authors carefully distinguish (1) the analysis of data in these databases of historical weather and harvests from (2) their interpretation of the underlying climatic and human history. Therefore, I would request the following minor revisions:

The abstract has added terms such as "reported" but needs to rework the language more carefully to reflect the new approach of the article. The following changes would be appropriate in order to make the meaning more precise and avoid inaccurate claims:

- "reported extreme droughts [floods] occurred" should be: "extreme droughts [floods] were reported" (or a similar phrase). The key distinction here is a report of a drought [flood] is not the same as the historical occurrence of drought [flood].

- "reconstructed grain harvest was poor [medium, high] in" should be: "the grain harvest was reconstructed as poor [medium, high] for" (or a similar phrase).

- "occurrence of reported extreme drought in any sub-region of eastern China was significantly associated with reduced harvests in the long-term average" should be: "frequency of reporting of extreme droughts was significantly associated over the long term with lower reconstructed harvests" (or a similar phrase).

- "association between harvest and extreme floods" should be: "associated between the reported frequency of extreme floods and reconstructed low harvests" (or a similar phrase).

- "other social factors" should be "other historical factors" to include other historical environmental changes, both natural and anthropogenic

Accepted. All related expressions have been revised throughout the text to distinguish "the analysis of data in these databases of historical weather and harvests" from "their interpretation of the underlying climatic and human history".

On page 2, lines 20-34 have not been reworked to reflect the new approach of the article -- that is, to first discuss patterns in the data derived from the historical documents, and only then to discuss the interpretations of those patterns as historical climate impacts. This paragraph makes the unwarranted assumption that the reported frequencies of events in the historical documents represent real frequencies and that associations between the frequencies of reported disasters and extremes and variations in reconstructed grain harvests represent causation (i.e., climate impacts). The studies cited by Su et al. and Yin et al. present the same problems of historical method and epistemology as did the previous draft of this manuscript, as discussed in my previous review. Therefore, their results need to be qualified in the same manner. The authors may refer the reader to the discussion section for their causal interpretation.

Accepted. We revised the expression on the reconstructions by Su *et al.* and Yin *et al.*,

rephrased our objective, and highlighted that we worked with data first, and then discussed possible interpretations. (P2, L29-P3, L3)

*It should be noticed that these conclusions are based on reconstructed datasets, and do not necessarily reflect actual historical connections due to several reasons as discussed in the Discussion section. These studies focused on the connection between agriculture and long-term climate change, while the effect on harvest induced by short-term extreme events (such as extreme droughts/floods) might be different. Therefore, this study aims to explore the patterns in the data of extreme droughts/floods and harvests in eastern China from 801 to 1910, using reconstructions of regional grain harvest grades and extreme drought/flood events derived from Chinese historical documents. The results from historical datasets could provide implications to improve understanding in the relationship between poor harvests and extreme drought/flood, and how cold and warm periods, such as the Medieval Climate Anomaly (MCA, 950–1250) and the Little Ice Age (LIA, 1450–1850) (IPCC, 2013), contributed to difference in that relationship.*

In the sentence at the bottom of page 4 and top of page 5, it is important that the authors clarify that their method is based on an assumption that the probabilities for omitting drought and flood events in reporting and transmission of historical records were random and unbiased, for the reasons they have stated. They haven't actually proven that omissions were random and unbiased. They have merely made a reasonable argument that it would be appropriate to proceed on this assumption.

Accepted. (P5, L6-8)

*Extreme drought or extreme flood years were defined in this way under the assumption that the probabilities for omitting drought and flood events in reporting and transmission of historical records were random and unbiased, despite the greater frequency of missing data in the earlier records.*

Now that the authors have worked to distinguish their results (that is, the patterns and associations in their datasets) from their discussion (that is, the climate and historical

interpretation of those associations), their use of a combined "Results and Discussion" section has become more confusing. For the sake of clarity, I would encourage the authors to rename section 3 as simply "Results" and turn subsection 3.4 into a new section, "Discussion." Their discussion of results in the bottom of page 9 to the top of page 10 as well as their discussion of results at the top of page 12 might then be moved into the new "Discussion" section, as part of the authors' climatic and historical interpretation of the patterns in the data. The fact that these pattern makes sense from a meteorological perspective supports the case that droughts had a significant historical impact on grain harvests. The authors may wish to state this point clearly in their discussion.

Accepted. Interpretations have been moved to *Discussion* section. (P12, L17-P13, L8)

*The statistical results from section 3.2 indicate that the reported frequency of regional extreme droughts is closely associated with poor harvests in reconstruction, while no significant pattern is found between the reported frequency of extreme floods and reconstructed poor harvests. This phenomenon might be explained by the fact that extreme droughts usually cover an immense area dominated by a single large-scale air mass, leading to significant and extensive impacts on agriculture, economics, and many other social factors. On the other hand, extreme floods were mostly caused by rainstorms induced by the confrontation of air masses, which usually occur across a relatively narrow belt. Meanwhile, rainstorms could irrigate agricultural land in areas surrounding the extreme floods and, thus, improve grain yields, leading to limited impacts of extreme floods on harvests over an immense area (Zhang, 1982). Although it should be noted that these meteorological interpretations only apply for short-term extreme flood events. When it comes to long-term wet periods, it could as well be responsible for bad harvest, due to the low-temperature and short sunlight.*

*Section 3.3 showed the statistical results for the different pattern in association between reported frequency of regional extreme droughts and reconstructed grain harvests between warm and cold periods. These results suggest that the simultaneously occurrence of reported extreme droughts and reconstructed poor harvests tends to be weakened in a warm climate background. Possible cause might*

*be that, there was low and limited adaptability during that period, and the warm climate provided more thermal resources and extended the growing season, thus increasing the multiple cropping index and providing more thermal-limited lands for growing crops. This gave people more options to adapt to climatic variation, and mitigated the impacts of extreme droughts on harvest yield. As assessed by Zhang (1982), the harvest may change by approximately 10% if the temperature changed by 1℃ on national scale based on the data from 1909 to 1979, in which the harvest increased significantly in 7 out of 8 warm years. However, a cold climate could limit multiple cropping and shrink the area of arable land, leading to the harvest becoming more vulnerable to extreme drought. Moreover, as reported by Zhang et al. (2007), limited resources could also cause social turbulence, such as famine, peasant uprising, the outbreak of war, and population decline, all of which may further increase agricultural vulnerability. Therefore, even though the frequency of reporting of extreme droughts was slightly higher in the warm period of 920–1300, the frequency of reconstructed poor harvests did not increase significantly.*

The revised sections, while mostly clear, should receive further review for correct English grammar and word use before publication.

Accepted and revised.